

# Dynamic upscaling of decomposition kinetics for carbon cycling models

Arjun Chakrawal[1,2], Anke M. Herrmann [3], Johannes Koestel [3], Jerker Jarsjö [1,2], Naoise Nunan [4], Thomas Kätterer [5], and Stefano Manzoni[1,2]

[1]Department of Physical Geography, Stockholm University, Svante Arrhenius väg 8C, Frescati, SE-106 91 Stockholm, Sweden
[2]Bolin Centre for Climate Research, Stockholm University, Stockholm, Sweden
[3]Department of Soil & Environment, Swedish University of Agricultural Sciences, P. O. Box 7014, 75007 Uppsala, Sweden
[4]Institute of Ecology and Environmental Sciences - Paris, Sorbonne Universités, CNRS-IRD-INRA-P7-UPEC, 4 place Jussieu, 75005 Paris, France
[5]Department of Ecology, Swedish University of Agricultural Sciences, P. O. Box 7014, 75007 Uppsala, Sweden

**Correspondence:** Arjun Chakrawal (arjun.chakrawal@natgeo.su.se)

**Abstract.** The distribution of organic substrates and microorganism in soils is spatially heterogeneous at the micro-scale. Most soil carbon cycling models do not account for this micro-scale heterogeneity, which may affect predictions of carbon (C) fluxes and stocks. In this study, we hypothesize that the mean respiration rate $\overline{R}$ at the soil-core scale (i) is affected by the micro-scale spatial heterogeneity of substrate and microbes and (ii) depends upon the degree of this heterogeneity. To assess

theoretically the effect of spatial heterogeneities on $\overline{R}$, we contrast highly heterogeneous conditions with isolated patches of substrate and microbes versus spatially homogeneous conditions equivalent to those assumed in most soil C models. Moreover, we distinguish between biophysical heterogeneity, defined as the non-uniform spatial distribution of substrate and microbes, and full heterogeneity, defined as the non-uniform spatial distribution of substrate quality (or accessibility) in addition to biophysical heterogeneity.

Three commonly used formulations for decomposition kinetics (linear, multiplicative and Michaelis-Menten) are considered in a coupled substrate-microbial biomass model valid at the micro-scale. We start with a 2D domain characterized by a heterogeneous substrate distribution and numerically simulate organic matter dynamics at each cell in the domain. To interpret the mean behavior of this spatially-explicit system, we propose an analytical scale transition approach in which micro-scale heterogeneities affect $\overline{R}$ through the second order spatial moments (spatial variances and covariances).

It was not possible to capture the mean behavior of the heterogeneous system when the model assumed spatial homogeneity, because the second order moments cause the heterogeneous system to deviate from the behavior attained under homogeneous conditions. Consequently, $\overline{R}$ in the heterogeneous system can be higher or lower than the respiration of the homogeneous system, depending on the sign of the second order spatial moments. This effect of the spatial heterogeneities appears in the





upscaled nonlinear decomposition formulations, whereas the upscaled linear decomposition model deviates from homogeneous conditions only when substrate quality is heterogeneous. Thus, this study highlights the inadequacy of applying at the macro-scale the same decomposition formulations valid at the micro-scale, and proposes a scale transition approach as a way forward to capture micro-scale dynamics in core-scale models.

## 1  Introduction

Soil organic substrates and microorganisms are heterogeneously distributed in the soil medium (Nunan et al., 2002; Peth et al., 2014; Raynaud and Nunan, 2014; Rawlins et al., 2016). The importance of this heterogeneous distribution in soil organic matter (SOM) dynamics has been shown both experimentally and in modeling studies. Early experimental results show that the mineralization of SOM is affected by the non-uniform distribution of the substrates within macro and micro pores (Killham et al., 1993). The recognition that spatial location of substrates and microorganisms constrains decomposition and thus C persistence is causing a paradigm shift from the previous emphasis on chemical composition of organic substrates to a focus on the biophysical environment in which decomposition occurs (Schmidt et al., 2011). Soil pore structure is emerging as a fundamental property that integrates these biophysical constraints on decomposition (Dungait et al., 2012; Falconer et al., 2015; Fraser et al., 2016). The biophysical and biochemical properties of the pore structure such as pore connectivity, tortuosity of water and air diffusion pathways, and adsorption/desorption, limit the accessibility of organic substrates to decomposers. As a result, these micro-scale constraints create a spatially heterogeneous landscape with highly variable distributions of substrate and microbial C. In the following, we refer to this type of variability as micro-scale heterogeneity.

Despite the importance of micro-scale heterogeneities, most SOM decomposition models are based on reaction kinetics that are valid for reactions in well-mixed media, including C cycling schemes implemented in ecosystem and Earth system models. In well-mixed systems, the mean concentrations of substrate and microbial C, and the rates defined using these mean values are assumed to be representative of the system. Most existing SOM models embrace this assumption regardless of whether they are microbial implicit (i.e., based on first order kinetics) or microbial explicit (i.e., based on multiplicative and enzyme kinetics) (Manzoni and Porporato, 2009). This approach is often referred to as mean-field approximation and is meant to describe spatially averaged SOM dynamics at soil core- to plot-scales. There is an underlying, but untested, assumption that the kinetics that are valid under well mixed conditions at fine scales also hold at larger scales, where conditions are often far from well-mixed. For this assumption to hold, a spatially averaged C flux should be equal to the average flux when organic C is uniformly distributed throughout the system. This is not the case when C concentrations are heterogeneously distributed and the kinetics are nonlinear (Chesson, 1998; Melbourne and Chesson, 2006; Morozov and Poggiale, 2012; Van Oijen et al., 2017).



For example, even in the simple case of only two soil patches, the overall C fluxes follow more complex behaviors than within an individual homogeneous patch, requiring the use of kinetics that differ from those applied at the micro-scale (Manzoni et al., 2008). The use of the same decomposition kinetics across a wide range of spatial scales is therefore questionable in systems that are spatially heterogeneous and regulated by nonlinear kinetics.

To understand at which scale a model developed for well-mixed conditions is expected to work, both the spatial scale at which heterogeneities become important and the scale at which homogeneity can be assumed must be identified. The average inter-cell distance in soil is in the order of 10 µm (Raynaud and Nunan, 2014) and the median length of spatial correlation of SOM varies between approximately 40 and 175 µm (Rawlins et al., 2016). Furthermore, the pore class 30 to 150 µm has been argued to be the most important for microbial activity (Kravchenko and Guber, 2017). This heterogeneity occurring at

scales from ∼10 to 200 µm, is generally neglected in C cycling models. Below the ∼50 µm scale, diffusion time scales can be assumed to be faster than advection and reaction time scales. Thus, it can be argued that below ∼50 µm the assumption of homogeneity is likely to hold, while it is no longer valid above this threshold (see Section 2.1.1 for details). If homogeneity cannot be assumed, how should decomposition kinetics be described in soil cores or at larger scales that include strong spatial heterogeneity?

Including micro-scale heterogeneities in the kinetics of SOM models is recognized as a much needed advancement in the field (Manzoni and Porporato, 2009; Sierra and Muller, 2015; Wieder et al., 2015), though only a few attempts have been made in this direction (Ebrahimi and Or, 2016; Van Oijen et al., 2017). The challenge is therefore to develop spatially up-scaled models that describe SOM decomposition at the macro-scale while taking into account the micro-scale heterogeneities. Mathematically, this upscaling problem is equivalent to spatial averaging of the mass balance equations based on the well-

mixed assumption at the micro-scale.

Three types of upscaling approaches are often used for dynamical systems such as those used to describe soil biogeochemical processes: (i) spatial averaging of known numerically simulated C flux fields, (ii) definition of effective parameters to capture fine-scale heterogeneity, and (iii) scale transition theory or volume averaging of the equations at the micro-scale. Spatial averaging of simulated dynamics at the micro-scale is common (Allison, 2012; Kaiser et al., 2014; Yan et al., 2016), but this

approach does not lend itself to analytical solutions that would offer insights into the effects of heterogeneity on macroscopic properties. The effective parameter approach, more common in sub-surface hydrology (e.g. Dagan (1987)), has been used to relate the macroscopic decomposition rate to the characteristic parameters of micro-scale heterogeneity, but only in a minimal 'lumped' model (Manzoni et al., 2008). The estimated effective parameters tend to be specific to studied scenarios and difficult to generalize. Here, we focus mainly on the third method based on scale transition theory, because this approach provides a



dynamic link between micro- and macro-scale using spatial moment approximations (SMA). Using scale transition theory, it is possible to obtain an analytical, but approximate representation of dynamics at the macro-scale by accounting for the nonlinear dynamics at micro-scale.

Scale transition theory is based on spatial averaging of the dynamical equations themselves (as opposed to averaging known
fluxes as in point (i)). This approach has been used to study predator-prey population dynamics at the patch and regional scales (Bergström et al., 2006; Englund and Leonardsson, 2008; Barraquand and Murrell, 2013). The macroscopic (regional) population dynamics is controlled by the mean population densities of predator and prey, which in turn relate to the spatial statistics of population densities at the micro-scale (patch). Similar approaches are also used in hydrology to calculate average hydrologic fluxes when soil and micro-climatic conditions are spatially heterogeneous (Albertson and Montaldo, 2003; Fatichi
et al., 2015), and in groundwater hydrology to derive transport equations at the Darcy- or field-scale (in this field the approach is called 'volume averaging' (Dentz et al., 2011)). We are aware of only one study using similar techniques to scale up C and N fluxes in soils from plot to regional scale (Van Oijen et al., 2017). Specifically, an empirical nonlinear function was used to link methane and nitrous oxide fluxes to soil moisture and temperature at each grid cell (corresponding to the micro-scale model) and the scale transition theory was applied to calculate the mean fluxes at the regional scale. However, an explicit
expression linking fluxes to C pools at any time point is not always available. In most C cycle models, the fluxes are calculated by solving first the mass balance equations for the C pools (i.e., a system of differential equations). Therefore, to proceed, these differential equations at micro-scale must be scaled-up. This upscaling exercise is expected to yield a set of differential equations describing the mass balances of the spatially averaged C compartments, including kinetics for the macro-scale C fluxes that depend on the degree of micro-scale heterogeneity.

Using scale transition theory, here we develop a general theoretical approach to link micro- and macro-scales in SOM decomposition models. With this approach, we demonstrate the effect of heterogeneity and nonlinearity at the micro-scale on macroscopic decomposition rates. Two types of micro-scale heterogeneity are identified and accounted for: biophysical and biochemical. Biophysical heterogeneity is caused by the non-uniform spatial distribution of substrate and microbes (i.e., heterogeneous distribution of the state variables), and biochemical heterogeneity is a result of spatial variations in substrate
quality and thus turnover rates (i.e., heterogeneous distribution of the values of kinetic constants). With the proposed upscaling approach, we test the hypotheses that the rate of decomposition (i) is affected by the micro-scale spatial heterogeneity of substrate and microbial C and (ii) depends upon the degree of heterogeneity. Scale transition theory is applied to three types of micro-scale decomposition kinetics commonly employed in C cycling models: conventional linear, multiplicative,





and Michaelis-Menten (MM). Considering these three kinetic laws allow us to assess the consequences of neglecting spatial heterogeneities in the most common C cycling models. Our specific objectives are

1. To develop an analytical upscaling solution for a two pool C model

2. To quantify the impact of different spatial structures of substrate $C_s$, microbial biomass $C_b$ and kinetic parameters $\boldsymbol{k}$ on the C dynamics

3. To compare the results of a spatially-explicit heterogeneous model with the homogeneous equivalent as a function of the degree of heterogeneity

While the proposed upscaling approach is general, we apply it in this contribution to scale up pore-scale processes to the scale of a small soil core or laboratory soil sample. These theoretical developments can be applied to SOM models employed to study respiration and microbial responses to perturbations at this relatively small spatial scale, or in models describing dynamics at a larger scale over relatively uniform spatial domains.

## 2 Methods

### 2.1 Theory

We distinguish between 'micro-scale' equations valid at the small scale where the well-mixed assumption holds, from 'macro-scale' equations valid at a larger scale of interest, which result from spatial averaging of the microscopic equations. While our derivations are general, in the presented model setup and results, we interpret 'macro-scale' as the scale of a small soil core. The goal of spatial upscaling is to derive the macro-scale soil C dynamics by spatial averaging of the micro-scale dynamics. We employ two approaches: (i) a numerical approach based on grid-scale simulations followed by spatial averaging (upper panel Fig. 1) and (ii) an analytical approach based on scale transition theory (lower panel Fig. 1). The first, computationally demanding approach requires solving the micro-scale equations at each cell of the domain grid. The mean behavior is then estimated by first order spatial moment approximation, which corresponds to the spatial averaging of the micro-scale state variables $C_s$ and $C_b$ (and associated C fluxes) at each time step. This numerical approach is equivalent to running a distributed model where the biophysical laws describing the system are known at the micro-scale and applied throughout the entire domain. We define the spatial average operator for our 2D domain as

$$\overline{\chi(t)} = \frac{\iint \chi(x,y,t)dxdy}{\iint dxdy} \approx \frac{1}{N_x N_y} \sum_{i=1}^{N_x} \sum_{j=1}^{N_y} \chi_{i,j}(t), \tag{1}$$





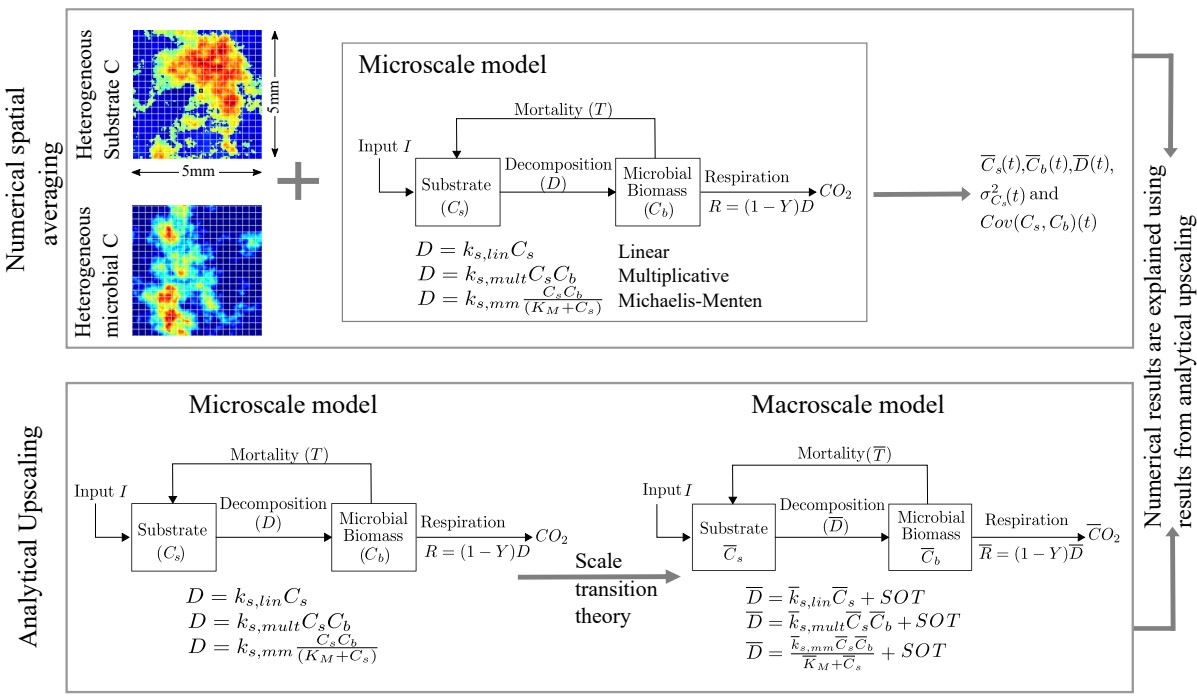

**Figure 1.** Schematic of the two upscaling approaches used to study the C dynamics at the macro-scale. Numerical spatial averaging (top panel): the micro-scale model is applied at each grid cell of the heterogeneous domain; and the mean C pools (substrate and microbial biomass), their mean fluxes, and second order spatial moments $\left(\sigma^2_{C_s}, \overline{C'_s C'_b}\right)$ are estimated by Eq. (21)–(25) at each time step. This approach is referred as 'distributed model'. Analytical upscaling (bottom panel): the micro-scale model is dynamically scaled up using scale transition theory, which provides the mean C fluxes as a function of mean C concentrations (mean-field approximation) and second order spatial moments representing the degree of heterogeneity. The deviations from the mean-field approximation are denoted as 'second order terms' (SOT) in the expressions for the mean decomposition fluxes ($\overline{D}$, where overbar represents mean quantities). The numerical results obtained from the distributed model are explained using the mathematical expression derived from analytical upscaling. This upscaling scheme is applied to three types of decomposition kinetics (linear, multiplicative and Michaelis-Menten), shown at the bottom of the lower panel.

where the double integral extends to the whole 2D domain, $\chi$ is a generic variable ($C_s$ or $C_b$) or C flux, and $N_x$ and $N_y$ are the number of grid cells in the x and y direction. The second equality allows estimating $\chi$ using the simulated time series of variable of interest in each grid cell (denoted by $\chi_{i,j}$). In contrast to numerically solving the problem at each grid cell, the second approach derives the dynamics of the macro-scale variables and fluxes using scale transition theory, discussed in the

5 following sections.





### 2.1.1 Micro-scale model of soil carbon dynamics

The dynamics of soil organic C in a homogeneous medium are characterized by specific reaction kinetics that define organic C fluxes, and the number and arrangement of soil C pools. For simplicity, we use a two pool model that subdivides organic carbon into two pools: (i) soil organic carbon substrate ($C_s$) and (ii) microbial biomass carbon ($C_b$) (Manzoni and Porporato, 2007;

German et al., 2012). This simple structure was selected because it is at the core of most microbial explicit models (Zelenev et al., 2000; Schimel and Weintraub, 2003). The typical time scale of diffusive fluxes is given by $\tau_{diff} = x^2/D$ where $x$ is the length scale of space discretization and $D$ is the diffusion coefficient (Hunt and Manzoni, 2015) and the typical time scale of reactive fluxes is given by the turnover time of the substrate; i.e., $\tau_{react}$. The ratio of the two time scales defines the Damköhler number, $Da = \tau_{diff}/\tau_{react}$, which provides the relative importance of mass transport of substrate via diffusion vs. reaction

(Dentz et al., 2011). For a relevant substrate such as glucose, $D$ is the order of $10^{-11}\,\mathrm{m}^2/\mathrm{s}$ (Watt et al., 2006), the turnover time is in the order $\sim 1$ day and the length scale of the order of $\sim 50$ µm. With these values, $Da << 1$, which characterizes a reaction-limited system in well-mixed conditions. The result of this approximated calculation would not change with reaction time scales in the order of a few hours. Thus, the well-mixed assumption is valid at the scale of a pore $\sim 50$ µm and we refer to this model as a 'micro-scale model' (Fig. 1). The general mathematical description of the microscale model is given by

$$\frac{\mathrm{d}C_s}{\mathrm{d}t} = I - D + T, \tag{2}$$

$$\frac{\mathrm{d}C_b}{\mathrm{d}t} = YD - T, \tag{3}$$

where $I$ is the rate of external input of organic C, $D$ is the rate of decomposition, $T$ is the microbial mortality, and $Y$ is the microbial carbon use efficiency. The substrate $C_s$ and microbial carbon $C_b$ are the state variables of the micro-scale model, and their mass balances, Eq. (2) and (3), describe their temporal evolution in absence of a spatial dimension. The rate of

20 decomposition is described by three commonly used formulations: linear (Eq.4), multiplicative (Eq.5), and MM (Eq.6) (Wutzler and Reichstein, 2008; Manzoni and Porporato, 2009),

$$D = k_{s,lin}C_s, \tag{4}$$

$$D = k_{s,mult}C_sC_b, \tag{5}$$

$$D = k_{s,mm}\frac{C_sC_b}{(K_M + C_s)}, \tag{6}$$





**Table 1.** Summary of the microscopic decomposition functions and steady state solutions

|  | Conventional (subscript lin) | Multiplicative (subscript mult) | Michaelis-Menten (subscript mm) |
|---|---|---|---|
| $D$ | $k_{s,lin}C_s$ | $k_{s,mult}C_sC_b$ | $\frac{k_{s,mm}C_sC_b}{C_s+K_m}$ |
| $T$ | $k_BC_b$ | $k_BC_b$ | $k_BC_b$ |
| Steady state $C_s^*$ | $\frac{I}{(1-Y)k_{s,lin}}$ | $\frac{k_B}{Yk_{s,mult}}$ | $\frac{K_Mk_B}{Yk_{s,mm}-k_B}$ |
| Steady state $C_b^*$ | $\frac{YI}{(1-Y)k_B}$ | $\frac{YI}{(1-Y)k_B}$ | $\frac{YI}{(1-Y)k_B}$ |

where $k_{s,lin}$, $k_{s,mult}$ and $k_{s,mm}$ are the decomposition rate constants for linear, multiplicative, and MM kinetics respectively; and $K_M$ is the half saturation constant for the MM kinetics. Table 1 summarizes the functional form of $D$ and corresponding steady state solutions for each case. Microbial mortality is typically assumed to follow first order kinetics ($T = k_BC_b$). We assume constant temperature and soil moisture conditions so that $D$ is only a function of $C_s$ and $C_b$. This assumption facilitates

assessing the role of spatial heterogeneity of C substrates and microbial biomass in our idealized system.

### 2.1.2 Spatial upscaling of soil carbon dynamics: scale transition theory

The scale transition theory links the dynamics at two different scales (Melbourne and Chesson, 2006; Morozov and Poggiale, 2012). Here we applied it to study the C dynamics at micro- and macro-scale, and derive the changes in the structure of the equations describing the C pools and their fluxes at macro-scale. To upscale the micro-scale model, the spatial averaging

operator given by Eq. (1) is applied to Eq. (2) and (3), leading to the governing equations at the macro-scale,

$$\frac{\overline{\mathrm{d}C_s}}{\mathrm{d}t} = \overline{I} - \overline{D} + \overline{T}, \tag{7}$$

$$\frac{\overline{\mathrm{d}C_b}}{\mathrm{d}t} = \overline{YD} - \overline{T}, \tag{8}$$

where the overbars denote the spatially averaged micro-scale quantities, so that $\overline{D}$ and $\overline{T}$ are the macro-scale rates of decomposition and microbial mortality. Since the order of averaging and differentiation can be exchanged, the right hand side of Eq.

(7) and (8) can be written as $\frac{\mathrm{d}\overline{C_s}}{\mathrm{d}t}$ and $\frac{\mathrm{d}\overline{C_b}}{\mathrm{d}t}$. Moreover, we assume that $Y$ and $I$ are spatially invariant, so that averaging does not alter their values. The final mass balance equations for substrate and microbial C at macro-scale are thus given by

$$\frac{\mathrm{d}\overline{C_s}}{\mathrm{d}t} = \overline{I} - \overline{D} + \overline{T}, \tag{9}$$

$$\frac{\mathrm{d}\overline{C_b}}{\mathrm{d}t} = Y\overline{D} - \overline{T}, \tag{10}$$

$$\overline{R} = \frac{\mathrm{d}\overline{CO2}}{\mathrm{d}t} = (1 - Y)\overline{D}. \tag{11}$$





where $\overline{R}$ is the mean respiration at macro-scale. In Eq.(9)-(11), the macro-scale variables $\overline{C}_s$ and $\overline{C}_b$ can be obtained once the average fluxes $\overline{D}$ and $\overline{T}$ are known. The next step is therefore to express $\overline{D}$ and $\overline{T}$ as a function of macro- and micro-scale state variables: $\overline{C}_s, \overline{C}_b$, and $C_s, C_b$ respectively.

We can generalize the problem and consider a generic microscopic C flux (i.e. $D$ or $T$) as a nonlinear (and smooth) function $F$ of state variables $C_s, C_b$ and a parameter vector $\boldsymbol{k}$ ($[k_1, k_2, ..., k_n]$), where $n$ is the number of parameters. The spatial averages of $C_s, C_b$ and $k$ are denoted as $\overline{C}_s, \overline{C}_b$ and $\overline{\boldsymbol{k}}$ ($[\overline{k}_1, \overline{k}_2, ..., \overline{k}_n]$). Applying the averaging operator given by Eq. (1) to a multivariate Taylor's series expansion of $F(C_s, C_b, k)$ around the spatial average value of $C_s, C_b$ and $\overline{\boldsymbol{k}}$ and truncating the series to second order gives the macroscopic C flux (detailed derivation is provided in Appendix),

$$\overline{F}(C_s, C_b, \boldsymbol{k}) = F(\overline{C}_s, \overline{C}_b, \overline{\boldsymbol{k}}) + \frac{1}{2}\frac{\partial^2 F}{\partial C_s^2}\bigg|_{\overline{C}_s, \overline{C}_b, \overline{\boldsymbol{k}}}\sigma^2_{C_s} + \frac{1}{2}\frac{\partial^2 F}{\partial C_b^2}\bigg|_{\overline{C}_s, \overline{C}_b, \overline{\boldsymbol{k}}}\sigma^2_{C_b} + \sum_{i=1}^{n}\sum_{j=1}^{n}\frac{\partial^2 F}{\partial k_i \partial k_j}\bigg|_{\overline{C}_s, \overline{C}_b, \overline{\boldsymbol{k}}}\overline{k'_i k'_j} +$$
$$\frac{\partial^2 F}{\partial C_s \partial C_b}\bigg|_{\overline{C}_s, \overline{C}_b, \overline{\boldsymbol{k}}}\overline{C'_s C'_b} + \sum_{i=1}^{n}\frac{\partial^2 F}{\partial k_i \partial C_s}\bigg|_{\overline{C}_s, \overline{C}_b, \overline{\boldsymbol{k}}}\overline{k'_i C'_s} + \sum_{i=1}^{n}\frac{\partial^2 F}{\partial k_i \partial C_b}\bigg|_{\overline{C}_s, \overline{C}_b, \overline{\boldsymbol{k}}}\overline{k'_i C'_b}, \tag{12}$$

where $\overline{F}(C_s, C_b, \boldsymbol{k})$ is the macroscopic C flux, $\sigma^2_{C_s}$ and $\sigma^2_{C_b}$ are the spatial variances of substrate and microbial C respectively; $\overline{k'_i k'_j}$ is the spatial variance (if $i = j$) or spatial covariance (if $i \neq j$) between the micro-scale parameters; $\overline{C'_s C'_b}$, $\overline{k'_i C'_s}$ and $\overline{k'_i C'_b}$ are the spatial covariances between micro-scale substrate and microbial C, substrate and parameters, and microbial biomass and parameters, respectively.

In Eq. (12), the first term on the right hand side, $F(\overline{C}_s, \overline{C}_b, \overline{\boldsymbol{k}})$, represents the first order approximation of $\overline{F}(C_s, C_b, \boldsymbol{k})$ also known as 'mean-field' approximation (MFA). For multiplicative kinetics, MFA is given by $k_{s,mult}\overline{C}_s\overline{C}_b$ and for MM kinetics it is $k_{s,mm}\overline{C}_s\overline{C}_b/(K_M + \overline{C}_s)$. Most C cycling models neglect all the other terms in Eq. (12). The remaining six spatial variance and covariance terms in Eq. (12) are collectively referred to as 'second order terms' (SOT). When the system is well-mixed, all variances and covariance terms vanish, leaving only the MFA. Therefore, only considering the MFA is equivalent to assuming well-mixed conditions at the macro-scale (i.e., Eq. (2) and (3) are mathematically equivalent to Eq. (9) and (10)).

Equation (12) provides a proof that the 'mean-field' approximation is a specific case of the more general expression for a macroscopic C flux that also depends on spatial heterogeneity through the SOT. The MFA is valid only when either of the following two conditions are met. First, the micro-scale decomposition rate is assumed to follow first order kinetics, because when $F$ is a linear function of substrate and microbial C, the second order partial derivatives in Eq. (12) are zero. Second, $C_s, C_b$ and kinetic parameters are spatially homogeneous, because in this case all the second order moments (spatial variances and covariances) are zero. However, if $F$ is nonlinear, the second order partial derivatives are non-zero; similarly, if any type of





micro-scale biophysical or biochemical heterogeneity is present, the SOT in Eq. (12) play a role in determining the macroscopic C dynamics.

Equation (12) illustrates the advantage of using scale transition theory as it provides an approximate analytical relation between the micro- and macro-scale quantities, which allows an immediate assessment of the role of both nonlinearities in the
C flux formulations and spatial heterogeneities. Importantly, in some cases, Eq. (12) yields an exact (rather than approximated) equation for macro-scale quantities, as shown in the following section.

## 2.2 Effect of micro-scale heterogeneities on macro-scale dynamics

Depending upon the kinetics of the micro-scale decomposition model (Table 1), the macro-scale $\overline{D}$ is expected to take different forms. Using different kinetic models, we now discuss some specific cases of micro-scale heterogeneities based on their
biophysical or biochemical nature. Biophysical heterogeneity is characterized by the spatially heterogeneous distribution of substrate and microbial C, whereas biochemical heterogeneity is characterized by the spatially heterogeneous distribution of substrate quality and microbial properties, captured by the kinetic parameters. The inaccessibility of SOM can result in C persistence. Therefore, inaccessibility can be modelled (at least at a conceptual level) through kinetic rate constants, similar to biochemical properties. In the simple model used here, accessibility to substrates or chemical recalcitrance are not mechanisti-
cally distinguished, so variations in substrate 'quality' in the broadest sense can be interpreted as spatial heterogeneity in either chemical characteristics or accessibility at the microscale.

First, we focus on systems with only biophysical heterogeneity of substrate and microbial C. For the first order kinetics model, the rate of decomposition is given by $D = k_{s,lin}C_s$, and using Eq. (12) and substituting $F = D = k_{s,lin}C_s$, we obtain

$$\overline{D} = k_{s,lin}\overline{C}_s. \tag{13}$$

In Eq. (13), $\overline{D}$ has the same form as $D$, indicating that microbial-implicit first order kinetic models do not show any sensitivity to spatial heterogeneities because of the linearity of the decomposition function. For the multiplicative model, the rate of decomposition at the micro-scale is given by

$$D = k_{s,mult}C_sC_b, \tag{14}$$





where $k_{s,mult}$ is spatially constant and the initial values of the state variables $C_s$ and $C_b$ are spatially variable. Inserting Eq. (14) into Eq. (12) gives

$$\overline{D} = k_{s,mult}\overline{C}_s\overline{C}_b + k_{s,mult}\overline{C'_sC'_b}. \tag{15}$$

In Eq. (15), the biophysical heterogeneities play a role through the covariance term $\overline{C'_sC'_b}$. Note that Eq. (15) is an exact

equation because all the spatial moments of order higher than two are zero. Thus, only the mean state variables and the spatial covariance are needed to fully characterize the macro-scale dynamics for this case. Furthermore, a positive spatial covariance (i.e. co-location of substrates and microorganisms) would increase the mean decomposition rate ($\overline{D}$), whereas a negative spatial covariance (i.e. spatial separation between substrates and microorganisms) would decrease it.

Similar to the multiplicative decomposition model, also in models based on MM kinetics the rate of decomposition at the

macro-scale depends on the covariance term $\overline{C'_sC'_b}$ and an additional term representing the spatial variance of the substrate (Table 2). The spatial variance of the substrate is always negative because the variance is a positive quantity and the partial derivative multiplying the variance is negative in all decomposition functions that saturate at high substrate concentration. In contrast, the spatial covariance term is positive or negative based on the sign of $\overline{C'_sC'_b}$. Therefore, when using the MM kinetics, $\overline{D}$ can be the approximated by the MFA only if variance and covariance balance each other or are both negligible.

Second, we consider only biochemical heterogeneity. In this case, model parameters $\overline{C}_b$ and $\overline{\boldsymbol{k}}$ ($[\overline{k}_1, \overline{k}_2, ..., \overline{k}_n]$) vary spatially but the initial value of state variables $C_s$ and $C_b$ are constant everywhere in the domain. With linear decomposition, substituting $D = k_{s,lin}C_s$ into Eq. (12) yields

$$\overline{D} = \overline{k}_{s,lin}\,\overline{C}_s + \overline{k'_{s,lin}C'_s}. \tag{16}$$

Equation (16) shows that for a biochemical heterogeneous system, even the simplest linear model requires an additional covari-

ance term to describe the governing equations at the macro-scale. This covariance term might change the linear microscopic model into a nonlinear macroscopic model. For the multiplicative model (Eq. (14)), Eq. (12) yields

$$\overline{D} = \overline{k}_{s,mult}\overline{C}_s\overline{C}_b + \overline{C}_b\,\overline{k'_{s,mult}C'_s} + \overline{C}_s\,\overline{k'_{s,mult}C'_b}, \tag{17}$$





where $\overline{k'_{s,mult}C'_s}$ and $\overline{k'_{s,mult}C'_b}$ are respectively the spatial covariances between the state variables $C_s$, $C_b$ and the rate constant parameter $k_{s,mult}$. These two additional spatial covariance terms capture the effects of biochemical heterogeneity caused by the spatial variation in the rate constants of decomposition.

Lastly, we consider a heterogeneous system with combined biophysical and biochemical heterogeneities, denoted as 'fully heterogeneous'. Again, we use the multiplicative model to illustrate the relation between the dynamics at the micro- and macro-scale. Now, all the state variables and parameters in $D$ at the micro-scale are spatially variable. For the multiplicative kinetics, $k_{s,mult}$, $C_s$ and $C_b$ are spatially variable, so that inserting Eq. (14) into Eq. (12) gives

$$\overline{D} = \overline{k}_{s,mult}\overline{C}_s\overline{C}_b + \overline{C}_b\,\overline{k'_{s,mult}C'_s} + \overline{C}_s\,\overline{k'_{s,mult}C'_b} + \overline{k}_{s,mult}\,\overline{C'_sC'_b}. \tag{18}$$

This generalized case includes all the spatial covariances between parameters and the state variables, thereby capturing bio-physical and biochemical heterogeneities simultaneously. Moreover, Eq. (17) and (18) are second order approximations, but an exact equation can be obtained by including a third order term $\overline{k'_{s,mult}C'_sC'_b}$. A similar derivation is described for MM kinetics in the Appendix; however, an exact expression for macro-scale MM kinetics cannot be found and we only use the second order approximation. Table 2 provides a summary of the theoretical results for the discussed heterogeneous cases and for all three types of decomposition kinetics.

Similar derivations can be done for the microbial mortality rate ($F = T$). The Taylor expansion of microbial mortality is simpler because we assume $T$ to follow first order kinetics. This implies that all the second order terms are equal to zero, and $(\overline{T})$ is obtained as

$$\overline{T} = k_B\overline{C}_b. \tag{19}$$

The rate of decomposition at macro-scale can be used to calculate the specific growth rate ($SGR$) of the microorganisms - i.e., the respiration rate divided by the mean microbial C,

$$SGR = \frac{\overline{R}}{\overline{C}_b} = (1-Y)\frac{\overline{D}}{\overline{C}_b}. \tag{20}$$

To summarize the proposed upscaling approach, we started with the spatial averaging of the SOM dynamics equations at micro-scale and applied scale transition theory to derive relations between the micro- and macro-scale C fluxes, which depend on both mean state variables and their spatial statistics (Table 2). Thus, to solve the macro-scale Eq. (9) and (10), we still need





**Table 2.** Summary of macro-scale equations for the decomposition rate

|  | Biophysical heterogeneity | Biochemical heterogeneity | Full-heterogeneity |
|---|---|---|---|
| Linear | $k_{s,lin}\overline{C}_s$ | $\overline{k}_{s,lin}\,\overline{C}_s + \overline{k'_{s,lin}C'_s}$ | $\overline{k}_{s,lin}\,\overline{C}_s + \overline{k'_{s,lin}C'_s}$ |
| Multiplicative[1] | $k_{s,mult}\overline{C}_s\overline{C}_b$ $+ k_{s,mult}\overline{C'_sC'_b}$ | $k_{s,mult}\overline{C}_s\overline{C}_b$ $+ \overline{C}_s\,\overline{k'_{s,mult}C'_b}$ $+ \overline{C}_b\,\overline{k'_{s,mult}C'_s}$ $+ \overline{k'_{s,mult}C'_sC'_b}$ | $k_{s,mult}\overline{C}_s\overline{C}_b$ $+ \overline{C}_s\,\overline{k'_{s,mult}C'_b}$ $+ \overline{C}_b\,\overline{k'_{s,mult}C'_s}$ $+ k_{s,mult}\overline{C'_sC'_b}$ $+ \overline{k'_{s,mult}C'_sC'_b}$ |
| Michaelis-Menten | $\dfrac{k_{s,mm}\overline{C}_s\overline{C}_b}{K_M+\overline{C}_s}+$ $\dfrac{1}{2}\left[\dfrac{-2k_{s,mm}K_M\overline{C}_b}{K_M+\overline{C}_s^{\,2}}\right]\sigma^2_{C_s}+$ $\left[\dfrac{k_{s,mm}K_M}{(\overline{C}_s+K_M)^2}\right]\overline{C'_sC'_b}$ | Eq.(A8) | Eq.(A9) |

[1] The expression of $\overline{D}$ for multiplicative kinetics in each heterogeneity case is exact.

information regarding the second order moments i.e., $\sigma^2_{C_s}$ and $\overline{C'_sC'_b}$. To close the problem mathematically, $\sigma^2_{C_s}$ and $\overline{C'_sC'_b}$ can be regarded as extra state variables requiring additional differential equations describing their dynamics (Keeling et al., 2002; Murrell et al., 2004; Barraquand and Murrell, 2013). Alternatively, the second order terms can be parameterized as empirical functions of first order terms $\overline{C}_s$ , $\overline{C}_b$ and $\overline{k}$ (Bergström et al., 2006). Here, our goal is to quantify how heterogeneities alter C

5   fluxes in idealized systems, so we leave the closure problem for a future contribution and use instead the numerically simulated dynamics at the micro-scale to calculate the spatial moments $\sigma^2_{C_s}$ and $\overline{C'_sC'_b}$.

## 2.3   Model setup

To investigate the effect of micro-scale spatial heterogeneity of SOM on the decomposition dynamics, we use a synthetic approach, which allows considering a wide range of heterogeneous SOM fields. As in other spatially explicit models (Ginovart

10   and Valls, 1996; Allison, 2005; Kaiser et al., 2014), we start with a 2D domain characterized by an initial heterogeneous field of the substrate and numerically simulate the dynamics of SOM with the micro-scale two pool model in Eq. (2) and (3) at each cell in the domain. The 2D domain has $100 \times 100$ square grid cells with an edge length of 50 μm, and we populate it with randomly generated initial substrate fields. This numerical model is referred to as 'distributed model' (see, Fig. 1). From the solution of the distributed model, the mean behavior of the system ($\overline{C}_s, \overline{C}_b, \overline{D}, \sigma^2_{C_s}$ and $\overline{C'_sC'_b}$) can be calculated at each time





step by using sample statistics of $C_s$ and $C_b$

$$\overline{C_s}(t) \approx \frac{1}{N_x N_y} \sum_{i=1}^{N_x} \sum_{j=1}^{N_y} C_{s_{i,j}}(t),$$ (21)

$$\overline{C_b}(t) \approx \frac{1}{N_x N_y} \sum_{i=1}^{N_x} \sum_{j=1}^{N_y} C_{b_{i,j}}(t),$$ (22)

$$\overline{D}(t) \approx \frac{1}{N_x N_y} \sum_{i=1}^{N_x} \sum_{j=1}^{N_y} k_{s,mult_{i,j}} C_{s_{i,j}}(t) C_{b_{i,j}}(t),$$ (23)

$$\sigma_{C_s}^2(t) \approx \frac{1}{N_x N_y} \sum_{i=1}^{N_x} \sum_{j=1}^{N_y} \left[ C_{s_{i,j}}(t) - \overline{C}_s(t) \right]^2,$$ (24)

$$\overline{C_s' C_b'}(t) \approx \frac{1}{N_x N_y} \sum_{i=1}^{N_x} \sum_{j=1}^{N_y} \left[ C_{s_{i,j}}(t) - \overline{C}_s(t) \right] \left[ C_{b_{i,j}}(t) - \overline{C}_b(t) \right],$$ (25)

where $\overline{D}$ is specified for multiplicative kinetics; a similar approach was applied for MM kinetics. Table A1 in the appendix lists all the parameters related to different kinetic models used in simulations. We performed the simulation in mass units fg (fg= $10^{-15}$g) and later converted the state variables to concentration units i.e. mgC/g of soil.

## 2.4 Initial 2D random fields of SOM and kinetic parameters

Two-dimensional spatially correlated random fields of substrates and microbial C were generated to run the distributed model. For the heterogeneous field of microbial C, we used a random field generator that provides $100 \times 100$ spatially correlated random numbers between -1 and 1 (Lennon, 2000). These values were then re-scaled by an appropriate mean and standard deviation of microbial C. To simulate the dead zones in the heterogeneous system some grid cells were forced to have no microbial C (the obtained field is denoted $y_{i,j}$). Moreover, to allow comparison among simulations, the microbial C field was re-normalized to have a specified value of total initial microbial C,

$$C_{b_{i,j}} = \frac{y_{i,j}}{\sum_{i,j} y_{i,j}} C_{b,total} \text{ (fg)}$$ (26)

It is assumed that the $C_{b,total}$ is equal to 1% of the total amount of substrate in the domain (Witter, 1996), which is in turn calculated as $C_{s,total} = \overline{C}_{s,0} \times N_x \times N_y$ where $\overline{C}_{s,0}$ is the initial mean substrate C in a single grid cell (fg). The amount of substrate C in any grid cell is limited by the maximum amount of C that the cell can accommodate, according to the density of organic matter ($\rho_{SOM}$). We assume that each cell can contain between zero and 50% of this maximum amount of organic





C because approximately only 50% of organic matter on a mass basis is composed of organic C. The maximum amount of substrate C that one cell can contain is thus given by

$$C_{max} = 0.5 \rho_{SOM} \, cell_{volume} \text{ (fg)} \tag{27}$$

where $cell_{volume}$ is the volume of a cell in grid. The value of $\overline{C}_{s,0}$ was chosen so that the maximum C amount at a micro-
site does not exceed $C_{max}$. To summarize, the obtained spatially heterogeneous random fields of microbial C and substrate C satisfy the following constraints: i) the total amount of organic C is set, ii) the total amount of microbial C is 1% of total organic C, iii) the maximum amount of C in a cell is set (Eq. (27)), and iv) some grid cells have no microbial biomass.

To study the effects of degree of heterogeneity on C decomposition, we generated random fields of substrate C with different degrees of correlation with microbial C. We created three cases where substrate and microbial C were initially positively
correlated, negatively correlated or uncorrelated. The three cases were obtained by applying a linear operator on the microbial C fields with positive and negative slope to obtain positively and negatively correlated substrate C fields, respectively. The uncorrelated substrate C field was generated independently from the microbial C field. The case of positive initial correlation between substrate and microbial C would result in a heterogeneous system with spatial co-occurrence of substrate and microbial C, whereas initial negative correlation would result in isolated patches of substrate and microbial C. The uncorrelated
initial field of substrate and microbial C can be interpreted as the result of external disturbances disrupting preexisting spatial correlations. In all scenarios, the substrate distributions are normalized to have the same amount of $C_{s,total}$, thereby allowing for comparisons among different degrees of heterogeneity. Example of the heterogeneous fields of C substrate are shown in Figure 2.

To generate a heterogeneous random field for kinetic parameters, we considered a uniform distribution for $K_M$ and a log-
uniform distribution for $k_{s,mm}$ and $k_{s,mult}$ (Forney and Rothman, 2012; Manzoni et al., 2012). The log-uniform distributions were defined so that the mean kinetic constants were equal to those of the homogeneous system (Table A1) and their variances were tuned to characterize different degrees of heterogeneity. To generate the random fields, $N_x \times N_y$ random numbers were extracted from the chosen distributions and placed into a 2D matrix of size $N_x \times N_y$. Figure A5 in the appendix shows the probability densities for two different standard deviations for $k_{s,mm}$ and $k_{s,mult}$ (the parameters of the distributions are listed
in Tables A2 and A3).



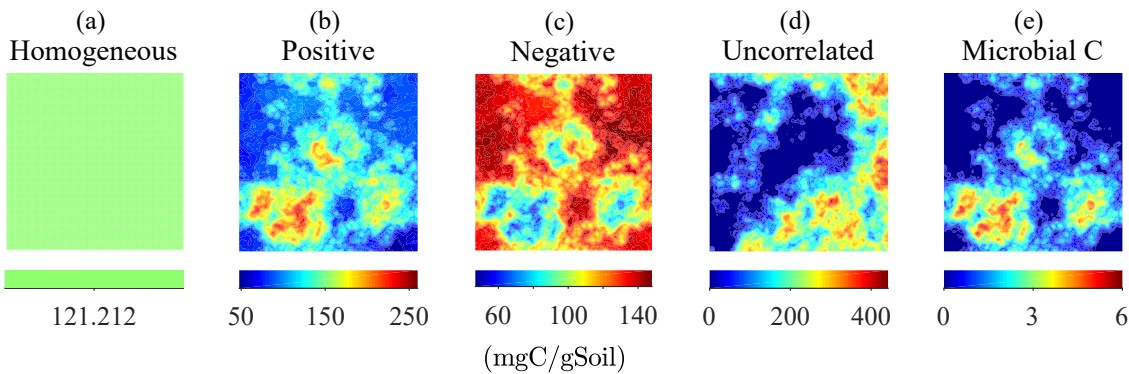

**Figure 2.** Examples of the homogeneous (a) and the heterogeneous distributions of substrate C constrained to have the same amount of total C substrate (b–d). The fields in b–d were obtained by imposing different degrees of correlation of substrate C with the initial heterogeneous distributions of microbial C, as indicated above each panel (e).

## 2.5 Estimation of kinetic parameters

To choose parameter values for the linear and multiplicative kinetics that allow comparisons with the MM model, we first simulated the substrate C dynamics at micro-scale for a given initial condition and using MM kinetics. Second, we fit the linear and multiplicative kinetics models to the time series obtained using MM kinetics(using the optimization toolbox in MATLAB). In this procedure, we assumed that the microbial mortality constant ($k_B$) was the same for all choices of the decomposition model.

## 2.6 Simulation scenarios

Two scenarios (Fig. 3), based on varying initial conditions (IC) were implemented to investigate the effects of micro-scale heterogeneities on macroscopic decomposition.

Scenario 1 (steady state initial condition): In this scenario, the initial heterogeneous field of substrate and microbial C was generated as described in section 2.4. The spatial mean of the initial substrate and microbial C match their steady state values given by the micro-scale equations (Eq. (2) and (3)) forced with a constant substrate input. Additionally, in the dead zones a small amount (i.e., order of magnitude ranging from one to two less than the steady state) of microbial C was imposed to ensure that OM could be decomposed, even if much slower than elsewhere.

Scenario 2 (transient initial condition): In the scenario, the initial heterogeneous field of substrate C was perturbed around a value much larger than the steady state as described in section 2.4.





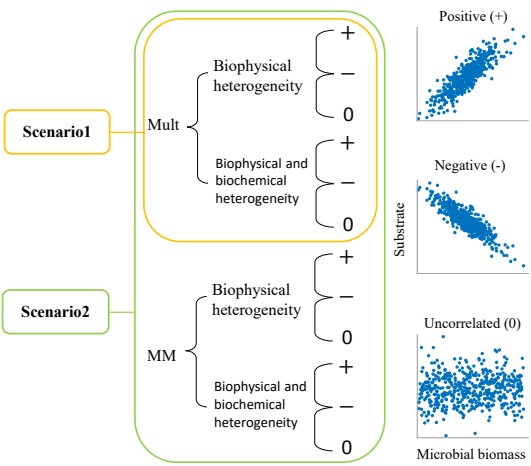

**Figure 3.** Two scenarios were implemented based on initial spatial distribution of substrate and microbial C. In scenario 1, substrate and microbial C are perturbed around the steady state of the micro-scale differential equations and simulations are only carried out with the multiplicative kinetics. In scenario 2, substrate and microbial C are perturbed to values larger than the steady state, and simulations are conducted for both multiplicative and Michaelis-Menten kinetics. For each scenario and type of heterogeneity, three different initial distributions of substrate and microbial biomass are considered as representative of micro-scale heterogeneities (positively correlated, negatively correlated, and uncorrelated fields of substrate and microbial C).

In the transient IC scenario, simulations were based on two nonlinear decomposition models (multiplicative and MM kinetic). However, in the steady state IC scenario, we present results only for multiplicative kinetics because MM kinetics can be approximated by multiplicative kinetics when the substrate is much smaller than the half-saturation constant ($K_M$), as is the case with the chosen initial heterogeneous substrate field and the parameter $K_M$. Results using the linear decomposition

5   model are not shown because with this model the spatially-averaged fluxes are equal to the macro-scale flux calculated at the mean C concentration.

In both scenarios, we explore the effects of biophysical and full heterogeneity on the temporal evolution of the mean state variables (substrate and microbial C) and their fluxes. We used the distributed modeling approach to estimate the mean quantities and second order spatial moments for three degrees of biophysical heterogeneity. A homogeneous system in which

10   the initial substrate and microbial C, as well as kinetic parameters, are spatially uniform was always used as a control. The combined effect of biophysical and biochemical heterogeneity was simulated by imposing the spatially heterogeneous kinetic parameters along with the heterogeneous initial substrate and microbial C.





## 3   Results

### 3.1   Scenario 1: Steady state initial condition

Figure 4 illustrates the temporal evolution of the macroscopic decomposition dynamics for the three different heterogeneous cases with varying degrees of initial correlation between substrate and microbial C in comparison to the homogeneous system. In figure 4, the left and right panels respectively show the effects of biophysical heterogeneity and full heterogeneity on the mean C pools and fluxes ($k_{s,mult}$ is based on the case 'biochemical heterogeneity 1' in Table A3).

Since the mean initial condition corresponds to the steady state of the micro-scale system, in a homogeneous soil no changes occur in substrate C (solid line in Fig. 4a and 4e) and microbial C (solid line in Fig. 4b and 4f), and the mean respiration rate is equal to the constant rate of input of external C (solid line in Fig. 4c and 4g). In contrast, for the system with biophysical heterogeneity, the mean C pools and respiration ($\overline{R}$) fluctuate towards the steady state of the micro-scale system as a result of the heterogeneous initial placement of C substrates. Similarly, for the fully heterogeneous system the mean microbial C pool (Fig.4f) and fluxes (Fig.4g) fluctuate near their steady state values, but the mean substrate C pool (Fig.4e) reaches a new steady state. The value of the new steady state for the $\overline{C}_s$ depends upon the parameters of the log-uniform distribution of $k_{s,mult}$ and is given by $\left[k_B(10^{-a} - 10^{-b})/Y(b-a)\ln(10)\right]$ (see details in the Appendix). In all heterogeneity scenarios, $\overline{R}$ is initially higher than in the homogeneous system when substrate and microbial C are initially correlated, whereas it is lower when substrate and microbial C are negatively correlated. When substrate and microbial C are uncorrelated, the system exhibits a behavior similar to that of the positively correlated fields (Fig. 4c), but with higher respiration peaks. This is caused by the high initial spatial variance of substrate C that resulted in hot spots richer in substrate C than in the positively correlated case (Fig.A1). Furthermore, in the multiplicative kinetics, the respiration flux is proportional to the amount of substrate C, so that larger variations in substrate cause larger fluctuations in the mean respiration flux. In the fully heterogeneous system, fluxes show similar dynamics as those in the biophysically heterogeneous system.

Figure 4d and 4h show the sum of all higher order terms ($\sum HOT$), including the third order term $\overline{k'_{s,mult}C'_sC'_b}$ in addition to the SOT. For a biophysically heterogeneous system, the $\sum HOT$ only includes the spatial covariance term, but for a fully heterogeneous system it includes the last three terms of the Eq. (18). The $\sum HOT$ is initially positive, zero and negative, respectively for positively correlated, uncorrelated, and negatively correlated substrate and microbial C. All components of HOT show a dynamic behavior (Appendix, Fig. A2). A positive $\sum HOT$ value enhances $\overline{R}$, whereas a negative value decreases it in all three heterogeneous cases compared to homogeneous $\overline{R}$. This result is aligned with our expectation from the analytical expression of the macro-scale multiplicative model (Eq.15).





Figure 5 compares the specific growth rate (SGR) of heterogeneous and homogeneous systems as a function the mean substrate C. In the case of a homogeneous system, SGR is equal to the time-invariant ratio $\frac{R}{C_b^*}$ (shown as a black dot), whereas in the heterogeneous system SGR is given by the time-dependent ratio $\frac{\overline{R}}{\overline{C_b}}$. As a result, the oscillations in $(\overline{R})$ caused by the HOT give rise to a non-unique relation between SGR and mean substrate C.

## 3.2   Scenario 2: Transient initial condition

Figure 6 illustrates the temporal evolution of the mean properties of the macroscopic decomposition dynamics for multiplicative kinetics, for systems with either biophysical (left panel) or full (right panel) heterogeneity, when the initial condition is perturbed from the steady state. In this scenario, both homogeneous and heterogeneous systems exhibit transient dynamics because the initial conditions are set far from the steady state. Figure 6c and 6f show the mean respiration rate $\overline{R}$ for the biophysically and the fully heterogeneous system, respectively. The results in Fig. 6b indicate that, during the microbial growth phase, the production of microbial C is faster when substrate and microbial C are positively correlated or uncorrelated, compared to the case of negative correlation. Consequently, at the beginning of the simulation, the mean substrate $\overline{C}_s$ (Fig. 6a) is decomposed faster due to the higher respiration (Fig. 6c) for the uncorrelated and the positively correlated substrate and microbial C, and slower for the negatively correlated substrate and microbial C, when compared to the homogeneous system. By the end of the simulation period, in all heterogeneous scenarios, biomass production and substrate consumption are lower than in the homogeneous system. As in scenario 1, the initial mean respiration for the uncorrelated case is higher than that in the positively correlated case.

The fully heterogeneous system (Fig. 6 right panel) shows a similar behavior compared to the biophysically heterogeneous system, but the peaks of $\overline{R}$ appear earlier for all degrees of correlation between substrate and microbial C.

Figure 7 shows similar results as Fig. 6, but for Michaelis-Menten kinetics. The transient dynamics of the mean C pools and fluxes differ from those for multiplicative kinetics. During the initial growth period, the decomposition of mean substrate C in heterogeneous systems (Fig. 7a and 7d) occurs at a rate comparable to that in the homogeneous system, but afterward the $\overline{R}$ becomes higher for positive, similar for uncorrelated, and lower for negatively correlated substrate and microbial C (Fig. 7c and 7f). Interestingly, for MM kinetics the uncorrelated case does not show higher $\overline{R}$ compared to the positively correlated case (Fig. 7c), because now the respiration flux is limited by the maximum rate of decomposition and not only by substrate availability.

The fully heterogeneous system (Fig. 6 and 7 right panel) shows different behavior compared to the biophysically heterogeneous system. The peaks of $\overline{R}$ appear much earlier than in the biophysically heterogeneous system. Additionally, the values of





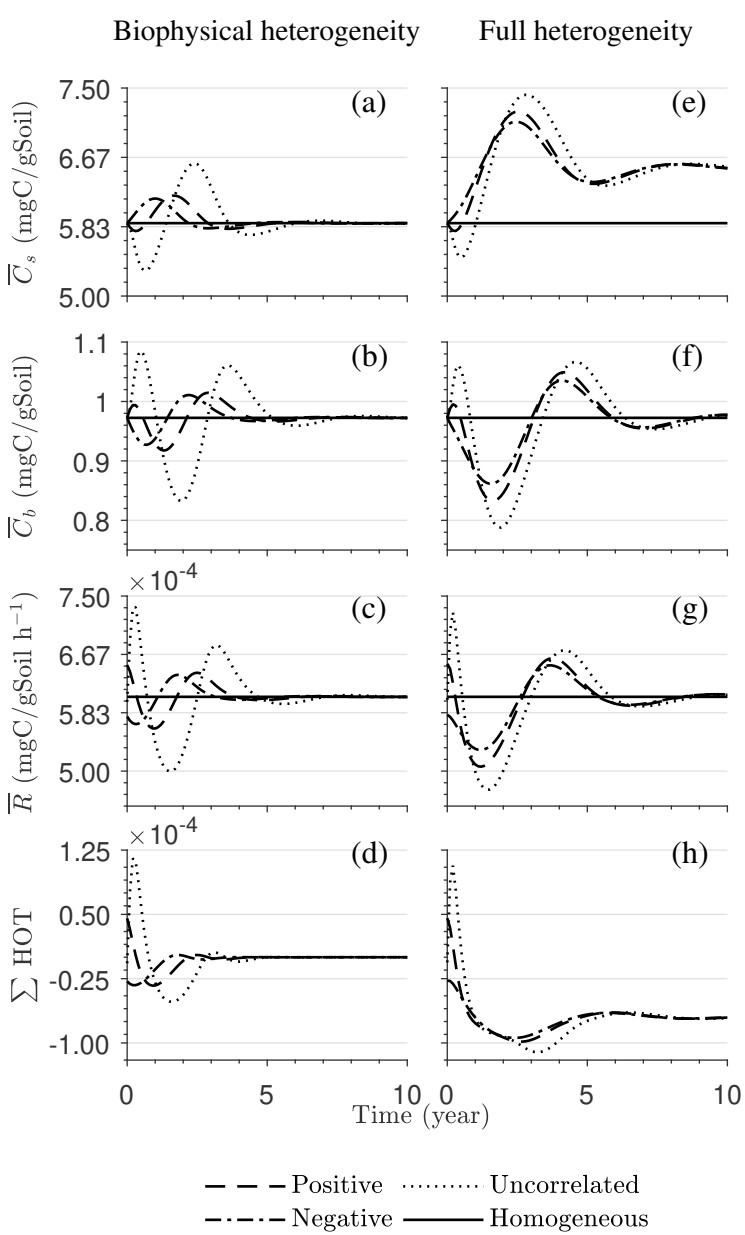

**Figure 4.** Scenario 1 (initial condition at steady state): effect of biophysical (left panel) and full heterogeneity (right panel) on the macroscopic decomposition dynamics when the substrate is distributed randomly around the steady state: (a,e) mean substrate C ($\overline{C}_s$), (b,f) mean microbial C ($\overline{C}_b$), (c,g) mean respiration rate ($\overline{R}$), and (d,h) sum of second and third order terms ($\sum HOT$).



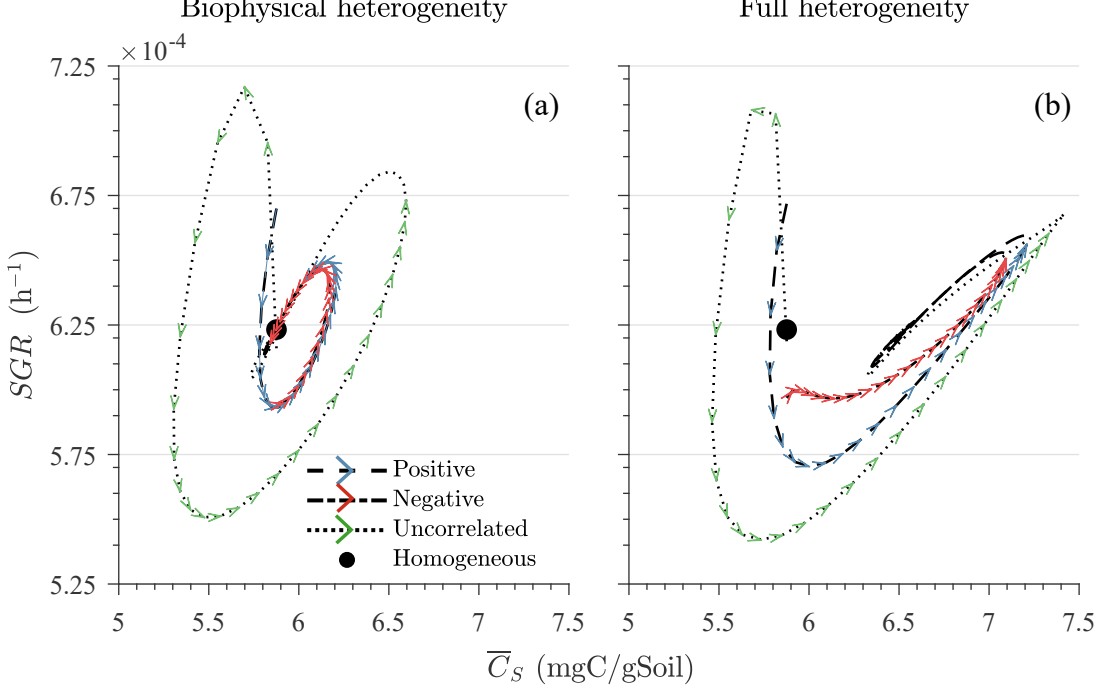

**Figure 5.** Scenario 1 (initial condition at steady state): effects of biophysical and full heterogeneity on the mean specific growth rate (SGR) as a function of mean substrate C for the three heterogeneous cases and the black dot showing the value of $\frac{R}{C_b^*}$. The direction of the arrows represent the increasing time.

mean fluxes and C pools after the peak are smaller than in the homogeneous system as well as in the system with only biophysical heterogeneity. The smaller mean fluxes are due to the left skewed probability distribution of the kinetic parameters ($k_{s,mult}$ and $k_{s,mm}$), which causes slower decay despite the mean values of the kinetic parameters being the same. Mathematically, this behavior is caused by the additional covariances in the fully heterogeneous system as explained in the following paragraph.

5   Figure 8 presents all the higher order spatial moments in the analytical expression of the macroscopic mean respiration rate, for the multiplicative decomposition model. The left (respectively right) vertical column shows the results for biophysically (fully) heterogeneous system with horizontal rows corresponding to the cases of positive (i.e. Fig. 8a and d), negative (i.e. Fig. 8b and e) and uncorrelated (i.e. Fig. 8c and f) substrate and biomass C. For both heterogeneous systems, $\overline{R}$ is higher than the MFA when the $\sum HOT$ is positive, whereas $\overline{R}$ is lower than the MFA when the contribution of these moments

10   is negative. This result agrees with the analytical expression and are valid for all types of biophysical heterogeneities. All the higher order spatial moments show a dynamic behavior. The $\sum HOT$ for biophysical heterogeneity is initially positive for the positively and the uncorrelated substrate and microbial C, but later becoming negative while it is always negative





for the negatively correlated substrate and microbial C. Spatial covariances among kinetic parameter and state variables (i.e., $\overline{k'_{s,mult}C'_s}, \overline{k'_{s,mult}C'_b}, \overline{k'_{s,mult}C'_sC'_b}$) also contribute to the $\sum HOT$ in the fully heterogeneous system in addition to $\overline{C'_sC'_b}$. Specifically, it is the spatial covariance between $k_{s,mult}$ and $C_b$ that gives rise to early peaks of $\overline{R}$.

The $\sum SOT$ (same as $\sum HOT$ but now limiting the HOTs to second order) for MM kinetics for the biophysical hetero-

geneity is given by the sum of the last two terms of Eq.(A7) and for the fully heterogeneous system it is given by the last eight terms of Eq. (A9). For the biophysically heterogeneous system, the values of $\sum SOT$ (Appendix, Fig. A3) are initially positive (very small in magnitude) for the positively correlated substrate and microbial C and later become negative, while for other heterogeneous cases $\sum SOT$ is always negative. Furthermore, the balance between variance and covariance terms makes the MFA a good approximation of $\overline{R}$ only when the combined second order terms are negligible, which is not the case

in this example. $\sum SOT$ of the fully heterogeneous system for MM kinetics is shown in appendix Fig.A4, and values were positive for the first 100 days of simulation and then negative onward, even though the heterogeneous $\overline{R}$ is smaller than the homogeneous $\overline{R}$ (Fig. 7). This result suggests that for a fully heterogeneous systems the SOT approximation of the macro-scale differential equation is not sufficient and additional higher order term can not be ignored. Individual components of $\sum SOT$ are not presented in Fig. A4 because of the large number of SOTs involved.

Figure 9 highlights the effect of these additional SOTs (HOTs for multiplicative) on the mean specific growth rate (SGR) of the heterogeneous system for multiplicative (top) and MM (bottom) kinetics. For both kinetics, the functional relation between the mean SGR and $\overline{C}_s$ for the heterogeneous system depends upon the initial degree of heterogeneity. In contrast, in the homogeneous system the mean SGR is a linear and saturating function of $\overline{C}_s$ for the multiplicative and MM kinetics, respectively. The effect of biophysical heterogeneity in both kinetic models is shown in Fig. 9a and 9c. The negative correlation between

substrate and microbial C leads to lower SGR than in the homogeneous system, even if both heterogeneous and homogeneous systems have exactly the same amount of total initial substrate and microbial C. In the case of positive correlation, the initial mean SGR is higher than in the homogeneous system, but in the later phase of decomposition, mean SGR becomes lower. Thus, when the substrate is physically accessible to the microorganisms, the mean SGR is initially higher but it decreases at a faster rate as the substrate is decomposed when compared to the homogeneous system. If substrate and microbial C are uncorrelated,

the SGR functional response remains between the negative and the positive correlation cases. In the fully heterogeneous system (Fig. 9b and 9d), the nonlinear character of the relation between mean SGR and $\overline{C}_s$ increases compared to the biophysically heterogeneous system. Interestingly, the mean SGR in the case of negative correlation for both kinetic models is higher (for high $\overline{C}_s$) than for the homogeneous system, despite being the co-location of substrates and microorganisms is less likely. This behavior is caused by the occurrence of patches with high turnover rate that control the mean SGR (Fig. 9a and 9c).





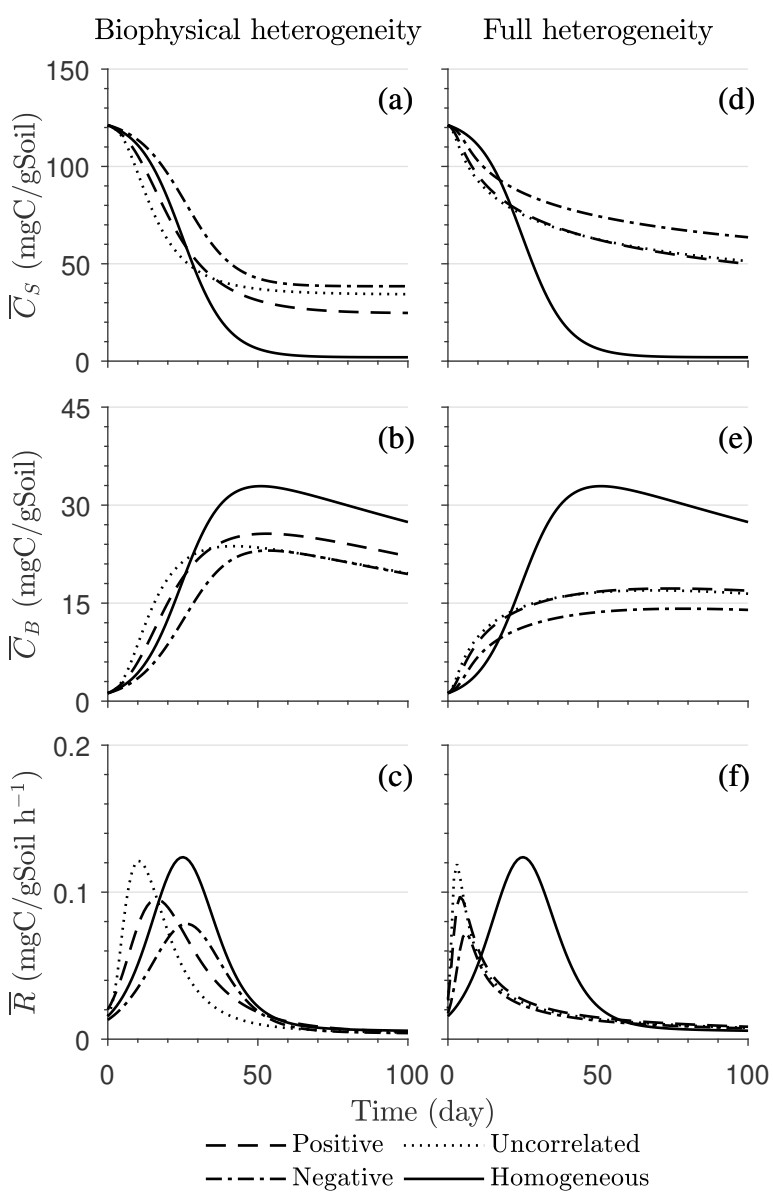

**Figure 6.** Scenario 2 (transient dynamics with multiplicative kinetics): effect of biophysical heterogeneity (left panel) and full heterogeneity (right panel) on the macroscopic decomposition dynamics when the substrate is distributed around a value higher than the steady state of the homogeneous system: (a,e) mean substrate C ($\overline{C}_s$), (b,f) mean microbial C ($\overline{C}_b$), and (c,g) mean respiration rate ($\overline{R}$).




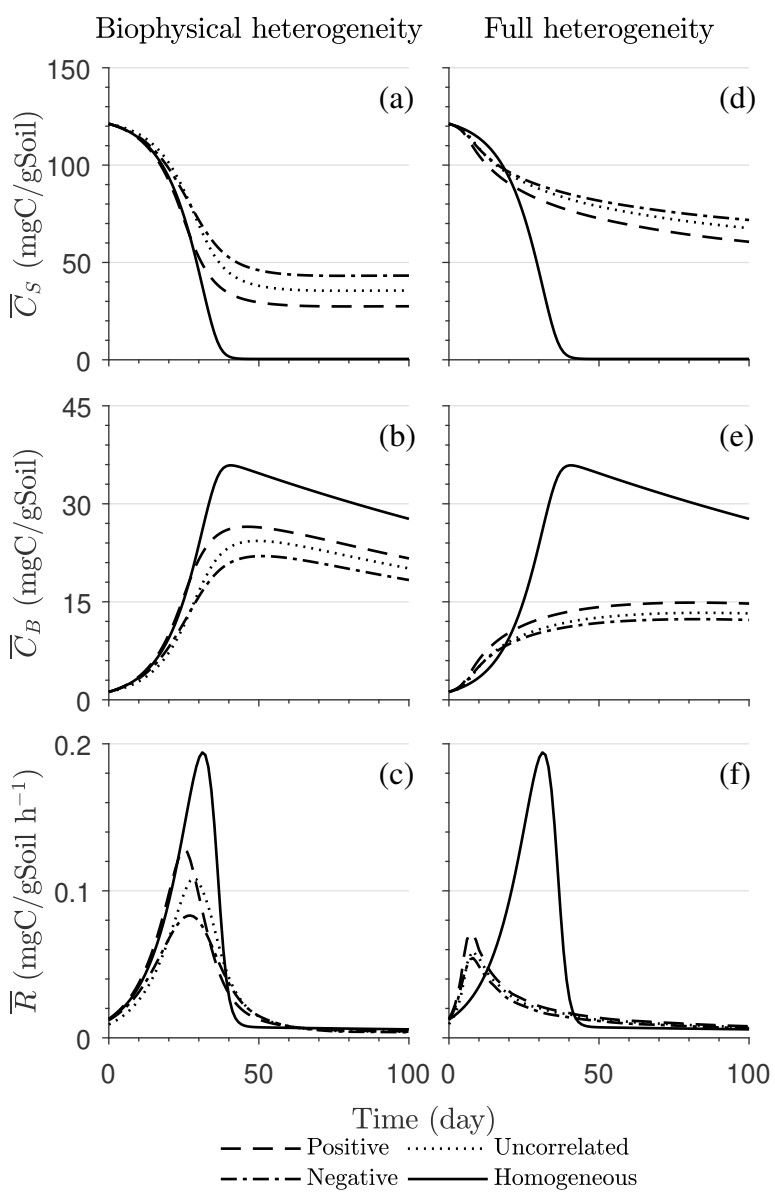

**Figure 7.** Scenario 2 (transient dynamics with Michaelis-Menten kinetics): effect of biophysical heterogeneity (left panel) and full heterogeneity (right panel) on the macroscopic decomposition dynamics when the substrate is distributed around a value higher than the steady state of the homogeneous system: (a,e) mean substrate C ($\overline{C}_s$), (b,f) mean microbial C ($\overline{C}_b$), and (c,g) mean respiration rate ($\overline{R}$).







**Figure 8.** Temporal evolution of mean respiration rate in the heterogeneous system ($\overline{R}_{het}$), which includes the mean-field approximation (MFA) and second order terms, and the respiration rate in the homogeneous system ($\overline{R}_{hom}$), for (a–c) biophysically and (d–f) fully heterogeneous system with positive (a–b), negative (c–d) and un-correlated (e–f) substrate and microbial C for multiplicative kinetics.



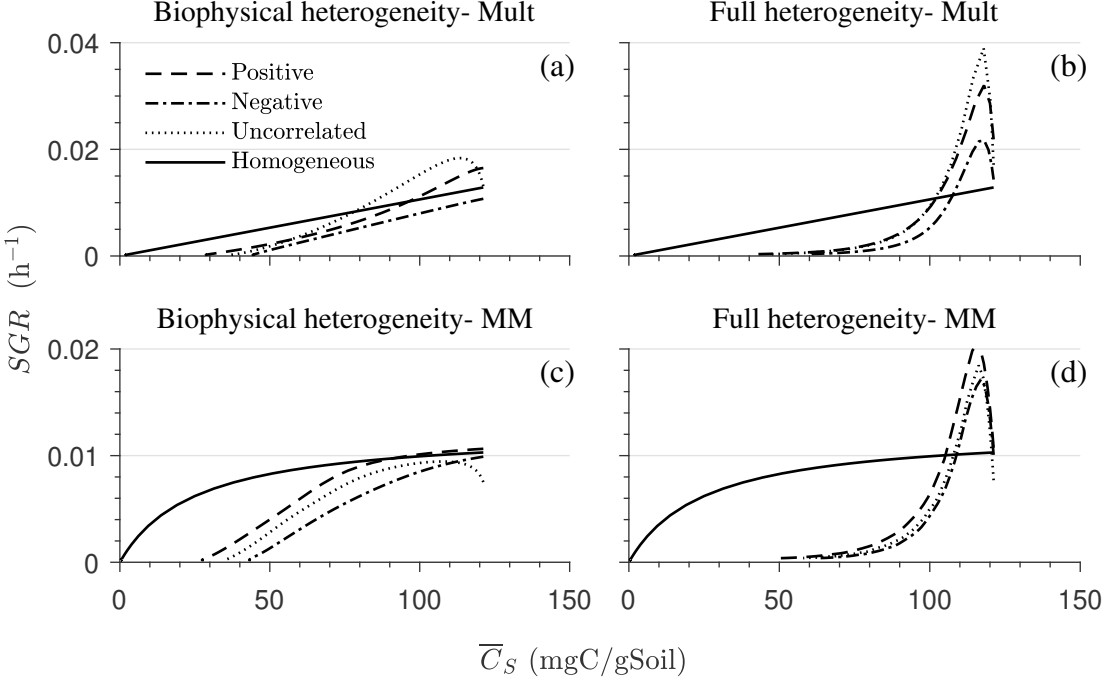

**Figure 9.** Effect of biophysical (left column) and full (right column) heterogeneity on the mean specific growth rate (SGR) as a function of mean substrate C ( $\overline{C}_s$ ) for the heterogeneous system for (a,b) multiplicative and (c,d) Michaelis-Menten kinetics. Time progresses from right to left, as $\overline{C}_s$ is depleted.

## 4   Discussion

### 4.1   Predicted effects of spatial heterogeneity on decomposition

The heterogeneous spatial distribution of organic matter in soils is a result of complex physical, chemical, and biological processes. Both the experimental quantification of the effects of heterogeneity on SOM dynamics (Kravchenko and Guber,

5   2017), and capturing such effects in mathematical models (Wieder et al., 2015) are challenging. Here, we used scale transition theory, applied to a two pool model, as a simple approach to analytically account for spatial heterogeneities. This approach was then used to upscale SOM dynamics in a range of idealized scenarios that cover different types of spatial heterogeneity. Even with the simplest scenarios, the macroscopic decomposition dynamics of a heterogeneous system differ from those predicted from the mean-field approximation (equivalent to assuming well-mixed conditions). This difference in the dynamics at two

10   spatial scales arises because spatial averaging of the nonlinear kinetics at the micro-scale create additional terms in the macro-scale equations that depend on the spatial distribution of organic matter and microorganisms (i.e., the second order spatial moments, SOT).



These second order moments capture fluxes that occur at the macro-scale as a result of nonlinear interactions in heterogeneous environments, but that do not occur in a homogeneous environment. Our numerical results showed that the second order spatial moments have their own dynamics that drive the heterogeneous system away from the mean-field approximation. Notably, while it is recognized that spatial distributions at the micro-scale affect macro-scale dynamics (Falconer et al., 2015), none of the current spatially-lumped SOM models include second or higher order terms that depend on micro-scale heterogeneity (see Sect. 4.3).

The simplicity of the model and the derived analytical expressions are such that specific insights on how heterogeneity shapes micro-scale decomposition patterns can be gleaned and hypotheses generated. The four main predictions of this model are

1. When only biophysical heterogeneity occurs, in the early microbial growth phase, macroscopic C fluxes are enhanced by co-location of substrates and microorganisms, and suppressed when they are isolated (Fig. 9a–c).

2. Combined biophysical and biochemical heterogeneity enhance C fluxes in the early stage of decomposition and suppress it in the later stages, compared to a homogeneous system (Fig. 9b–d).

3. Both biophysically and fully heterogeneous systems result in a transient persistence of SOM (Fig. 6a,d and 7a,d). This persistence vanishes for the biophysically heterogeneous in steady state (i.e. all the carbon is decomposed provided enough time; Fig. 4); however, the fully heterogeneous system retains this persistence even at steady state (i.e. not all C is decomposed as the substrate pool reaches a new equilibrium).

4. For a successive reduction in the degree of heterogeneity (i.e. systematically moving from a heterogeneous to a homogeneous system), macro-scale dynamics converge to the mean-field approximation; i.e., the same kinetics can be used at all scales (Sect. 2.1.2).

## 4.2 Linking theory and observations

We studied three initial heterogeneous distributions of substrate and microbial C; positive, negative or no correlation between these two variables. These heterogeneities may correspond to spatial aggregation, isolation or random occurrence of substrate and microorganisms, respectively. Spatial aggregation is expected in litter and in the surface soil where substrate is abundant and microbial colonies are formed around hot spots (Nunan et al., 2003). Spatial isolation is more likely to occur in the subsoil because of lower substrate and microorganism density as well as poor pore connectivity (Ekschmitt et al., 2008; Salomé et al., 2010), and C-rich patches occur around roots that are separated by large (in a relative sense) volumes of soil that





only receive diluted resources via percolation, diffusion, and bioturbation (Kuzyakov and Blagodatskaya, 2015). Uncorrelated spatial fields of substrate and microorganisms may correspond to spatial distributions between these two extremes. There are other examples of contrasting homogeneous vs. heterogeneous conditions. Disturbed, sieved or dispersed samples can be considered as homogeneous whereas intact soil samples retain their natural heterogeneity. Despite the correspondence of

our idealized heterogeneity scenarios with actual conditions in natural soils or soil samples, linking our model predictions to observations is challenging, mostly because the effects of heterogeneity can not be easily isolated in experiments or field observations.

Many studies have focused on understanding the effects of soil structure on the dynamics of SOM mineralization, either by changing the pore network by controlling the water potential (Killham et al., 1993; Ruamps et al., 2011), or by artificially

altering the soil structure through mechanical treatments (Stenger et al., 2002; Juarez et al., 2013; Negassa et al., 2015; Herbst et al., 2016). Changes in soil structure affect spatial heterogeneity of substrate and microbial C (Ruamps et al., 2011; Sleutel et al., 2012; Vos et al., 2013), but also introduce other types of heterogeneities that are not dealt with here. For example, samples with different pore networks (Ruamps et al., 2011) likely exhibit different water and air diffusive pathways, which in turn affect microbial respiration (Moyano et al., 2013; Manzoni and Katul, 2014; Herbst et al., 2016; Koestel and Schlüter,

2019). Furthermore, the same spatial distribution of decomposers and substrates in different pore sizes may result in different C fluxes because the size of the pores where decomposers reside affects the rate at which substrate can become available. Therefore, these studies may be ideal for studying the effect of soil structure, but not for isolating the spatial heterogeneity effects. When targeting experiments to test our up-scaled equations, using data from previous studies is challenging because (i) spatial distribution of substrate and/or microbial C are not changed in a controlled manner and (ii) the scale at which physical

treatments are performed is probably larger than the scale at which heterogeneity affects C dynamics.

We also found that SOT are highly dynamic, suggesting that the role of heterogeneities varies through time. However, in most long-term incubation experiments measurement frequency is low. Fraser et al. (2016) observed significant differences in the respiration dynamics using a very high frequency (6 minutes interval) measurements that is generally lost in daily or sub daily observations. In a recent review Baveye et al. (2018) suggested to improve the temporal resolution of observations of

the soil physical environment. Extending this idea to biochemical processes is important because a complete description of macro-scale C fluxes requires spatial and dynamic measurements of the second order moments (see Sect. 4.4).

Thus, experiments in which soil structure was manipulated do not allow direct testing of the predicted links between heterogeneity and decomposition kinetics at the macro-scale. Therefore, while the proposed mathematical framework is conceptually useful, it is challenging to test.





### 4.3 Developing soil carbon cycling models that account for micro-scale heterogeneity

Historically, the linear microbial implicit models were developed to explain long-term loss of C from agricultural soils or regional-scale variations in SOM (Jenny et al., 1949; Olson, 1963; Jenkinson and Rayner, 1977; Parton et al., 1987). However, when applied at fine spatial or temporal scale, these models fail to describe the dynamics of SOM (Zelenev et al., 2000; Manzoni

and Porporato, 2007). To fill this gap and describe microbial processes at the macro-scale, nonlinear microbial explicit models have been proposed (Schimel and Weintraub, 2003; Manzoni and Porporato, 2009; Xie, 2013). In contrast to these approaches that impose nonlinear kinetics at the macro-scale, here we started from the assumption that SOM kinetics are either linear or nonlinear at the micro-scale, and let scale transition theory determine the type of kinetics at the macro-scale.

When assuming linear kinetics at the micro-scale, we showed analytically that the kinetics at the macro-scale remain linear

and independent of soil biophysical heterogeneity (Eq. (13) and (19)). This result has implications for experimental studies linking soil architecture to SOM mineralization. In some of these studies, first order microbial implicit kinetics are used to describe the data (Bouckaert et al., 2013; Juarez et al., 2013). This means that if a linear model captures well the SOM dynamics in a heterogeneous system, then either the underlying micro-scale dynamics are indeed linear, or the averaging of underlying nonlinear equations leads to linearity at macro-scale.

Conversely, we demonstrate that nonlinear kinetics at micro-scale do not remain the same when scaling up. The macro-scale dynamics retain a clear signature of nonlinearities at the micro-scale in the MFA term, but the second order terms could be even more important than the MFA. Thus, nonlinear kinetics might improve SOM predictions because microbial activity is accounted for (Wieder et al., 2013) (at the cost of increased uncertainty, Wieder et al. (2018)), but the question remains: which kinetics formulation should be used at the macro-scale that captures both microbial activity and spatial heterogeneities?

We offer a framework to advance this area by using appropriately upscaled nonlinear kinetics including SOT at macro-scale. This upscaling framework can be extended to account for the role of other micro-scale interactions such as among substrates, microorganisms, and minerals, or even temporally varying connectivity due to water movement. These improvements, however, would come at the expense of increased number of nonlinear second order spatial moments.

To summarize, the proposed theoretical developments allow integration of spatial heterogeneity into decomposition kinetics.

Assuming that the second order spatial moments are known, this integration can be achieved by using the equations listed in Table 2 instead of standard linear or nonlinear kinetics equations used in the current models (Wieder et al., 2018; Abramoff et al., 2018). However, the second order moments and their dynamics are not known in general, as discussed at the end of the following section.





## 4.4 Limitations of the upscaling approach

To illustrate the effects of spatial heterogeneities alone, we simulated idealized laboratory conditions in which the environmental conditions are constant so that the decomposition rate is not affected by soil moisture and temperature changes through time and space. Moreover, we did not account for the spatial redistribution of organic matter and microbial biomass caused

by diffusion/dispersion/advection or bioturbation in the distributed model. Finally, the simulated domain is small compared to an actual soil sample, but we regard the number of simulated grid cells ($10^4$) as representative of the range of variation occurring in larger, similarly idealized samples. In other studies, more complex micro-scale models based on nonlinear reactive and diffusive fluxes have been implemented (Monga et al., 2008; Nguyen-Ngoc et al., 2013; Monga et al., 2014); however, their spatial upscaling would require volume averaging of the coupled transport and reaction equations, making the problem

mathematically intractable when aiming for analytical solutions (Whitaker, 1999; Valdés-Parada et al., 2009; Porter et al., 2011; Lugo-Méndez et al., 2015). The two pool micro-scale model with initial heterogeneous distributions of substrate and microorganisms as described in this study offers a simplified way of simulating reaction-diffusion systems. The two end-member cases of homogeneous and fully heterogeneous systems where grid cells are independent are representative of conditions in which diffusivities are high compared to reaction kinetics in the former and negligible in the latter. In more realistic settings,

conditions are likely to be intermediate between these two cases.

The upscaling mechanism described in this work assumes that microbial mortality is first order in microbial C, so that this term remains structurally similar in the macroscopic Eq. (9) and (10). A nonlinear mortality generalized by $T = k_B C_b^{\beta}$ (Georgiou et al., 2017) would create an additional term in the macro-scale equations. The mean microbial mortality can be calculated by inserting the nonlinear $T$ into Eq. (12), resulting in $\overline{T} = k_B \overline{C}_b^{\beta} + \beta \overline{C}_b^{\beta-1} \sigma_{C_s}^2$, where $\sigma_{C_s}^2$ is the spatial variance

of microbial C (for the biophysically heterogeneous system; i.e., $k_B$ is spatially invariant). For $\beta = 1$, the first order mortality is recovered, (Eq. (19)); for $\beta \neq 1, \overline{T}$ has an additional positive variance term that increases mortality at the macro-scale.

Finally, the upscaled macro-scale equations still require a closure scheme for integration; i.e., a set of equations linking the spatial moments to the mean state variables. With such a set of additional equations, the problem becomes mathematically 'closed', as the only remaining unknowns are the mean state variables. Examples of closure from other fields are mentioned in

the introduction (e.g., Bergström et al. (2006)) but finding a robust closure scheme remains challenging and will be the subject of future work.

Moreover, our derivations are general, but how these closure equations are formulated and parameterized will likely depend on the scale transition under consideration - soil pore to core (as in this work), soil core to field, or even field to landscape. It



is possible that a whole hierarchy of scale transitions is required to determine macro-scale equations suitable for regional or global-scale applications.

## 5  Conclusions and perspective

Most carbon cycling models implicitly assume a spatially homogeneous distribution of SOM in different C pools and are based
on the mean-field approximation of the rate of decomposition. However, assuming homogeneity is adequate only at the micro-scale in soils, due to the homogenizing effect of diffusion, which brings carbon sources and decomposers into direct contact with each other at such scales. Therefore, the mean-field approximation is valid only at the micro-scale, creating a challenge when developing SOM models at macro-scale that also account for environment heterogeneity. In this contribution, we used scale transition theory to link an idealized (but realistic) heterogeneous system and a homogeneous system by establishing an
analytical expression for the macroscopic mean decomposition rate that accounts for the micro-scale heterogeneities. Unlike the mean-field approximation adopted in most C cycling models, the upscaled governing equations we derived include second order spatial moments; i.e., spatial variances and/or covariances between micro-scale state variable and model parameters. The dynamical behavior of the second order terms drives the heterogeneous system away from the mean-field approximation. For a heterogeneous system, initially near steady state, micro-scale heterogeneities led to oscillations in the macro-scale respiration
flux and higher SOM persistence in a fully heterogeneous system. For a heterogeneous system perturbed from its equilibrium, the co-location of substrate and microorganisms increased macroscopic C fluxes compared to a case in which they were isolated.

In conclusion, this work provides a methodology to explicitly include micro-scale heterogeneity in soil C cycling models. Our upscaled kinetic equations could be used in lieu of current formulations, but additional equations describing the dynamics
of spatial moments should be further developed to mathematically close the problem. These upscaled equations show that, (i) heterogeneities alter the form of the carbon flux equations at the macro-scale and as a result (ii) co-location (respectively isolation) of microorganisms and their substrates promote (suppress) carbon fluxes in soils.

*Code availability.* The codes used to construct the heterogeneous soil maps and to solve the mass balance equations in heterogeneous domains are publicly available via DOI https://doi.org/10.5281/zenodo.3253880.





*Data availability.* The article does not use any relevant data.

**Appendix A**

**A1   Derivation of the macro-scale rate of decomposition**

Here we describe the derivation of the spatially averaged C flux for a generic microscopic C flux $F(C_s, C_b, k)$ using scale transition theory. As a first step, we calculate the multi-variate Taylor's series expansion of $F(C_s, C_b, k)$ around the spatial average value of $C_s$, $C_b$ and $\boldsymbol{k}$,

$$
\begin{aligned}
F(C_s, C_b, \boldsymbol{k}) = &\ F(\overline{C}_s, \overline{C}_b, \overline{\boldsymbol{k}}) + \frac{\partial F}{\partial C_s}\bigg|_{\overline{C}_s, \overline{C}_b, \overline{\boldsymbol{k}}}(C_s - \overline{C}_s) + \frac{\partial F}{\partial C_b}\bigg|_{\overline{C}_s, \overline{C}_b, \overline{\boldsymbol{k}}}(C_b - \overline{C}_b) + \frac{\partial F}{\partial C_b}\bigg|_{\overline{C}_s, \overline{C}_b, \overline{\boldsymbol{k}}}(C_b - \overline{C}_b) + \\
&\ \sum_{i=1}^{n} \frac{\partial F}{\partial k_i}\bigg|_{\overline{C}_s, \overline{C}_b, \overline{\boldsymbol{k}}}(k_i - \overline{k}_i) + \frac{1}{2}\frac{\partial^2 F}{\partial C_s^2}\bigg|_{\overline{C}_s, \overline{C}_b, \overline{\boldsymbol{k}}}(C_s - \overline{C}_s)^2 + \frac{1}{2}\frac{\partial^2 F}{\partial C_b^2}\bigg|_{\overline{C}_s, \overline{C}_b, \overline{\boldsymbol{k}}}(C_b - \overline{C}_b)^2 + \\
&\ \frac{1}{2}\sum_{i=1}^{n}\sum_{j=1}^{n} \frac{\partial^2 F}{\partial k_i \partial k_j}\bigg|_{\overline{C}_s, \overline{C}_b, \overline{\boldsymbol{k}}}(k_i - \overline{k}_i)(k_j - \overline{k}_j) + \frac{\partial^2 F}{\partial C_s \partial C_b}\bigg|_{\overline{C}_s, \overline{C}_b, \overline{\boldsymbol{k}}}(C_s - \overline{C}_s)(C_b - \overline{C}_b) + \\
&\ \sum_{i=1}^{n} \frac{\partial^2 F}{\partial k_i \partial C_s}\bigg|_{\overline{C}_s, \overline{C}_b, \overline{\boldsymbol{k}}}(C_s - \overline{C}_s)(k_i - \overline{k}_i) + \sum_{i=1}^{n} \frac{\partial^2 F}{\partial k_i \partial C_b}\bigg|_{\overline{C}_s, \overline{C}_b, \overline{\boldsymbol{k}}}(C_b - \overline{C}_b)(k_i - \overline{k}_i) + O(C_s^3 C_b^3 k_i^3),
\end{aligned}
\tag{A1}
$$

where $O(C_s^3, C_b^3, k_i^3)$ represents the higher order terms and the overbars denote the spatially averaged micro-scale quantities.

Second, the averaging operator given by Eq. (1) is applied in Eq. (A1). Truncating terms above the second order terms, Eq. (A1) becomes

$$
\begin{aligned}
\frac{1}{\int\int dxdy} & \int\int F(C_s, C_b, \boldsymbol{k})\, dxdy = \\
& \frac{1}{\int\int dxdy}\Bigg[ F(\overline{C}_s, \overline{C}_b, \overline{\boldsymbol{k}}) + \frac{\partial F}{\partial C_s}\bigg|_{\overline{C}_s, \overline{C}_b, \overline{\boldsymbol{k}}}(C_s - \overline{C}_s) + \frac{\partial F}{\partial C_b}\bigg|_{\overline{C}_s, \overline{C}_b, \overline{\boldsymbol{k}}}(C_b - \overline{C}_b) + \frac{\partial F}{\partial C_b}\bigg|_{\overline{C}_s, \overline{C}_b, \overline{\boldsymbol{k}}}(C_b - \overline{C}_b) + \\
& \sum_{i=1}^{n} \frac{\partial F}{\partial k_i}\bigg|_{\overline{C}_s, \overline{C}_b, \overline{\boldsymbol{k}}}(k_i - \overline{k}_i) + \frac{1}{2}\frac{\partial^2 F}{\partial C_s^2}\bigg|_{\overline{C}_s, \overline{C}_b, \overline{\boldsymbol{k}}}(C_s - \overline{C}_s)^2 + \frac{1}{2}\frac{\partial^2 F}{\partial C_b^2}\bigg|_{\overline{C}_s, \overline{C}_b, \overline{\boldsymbol{k}}}(C_b - \overline{C}_b)^2 + \\
& \frac{1}{2}\sum_{i=1}^{n}\sum_{j=1}^{n} \frac{\partial^2 F}{\partial k_i \partial k_j}\bigg|_{\overline{C}_s, \overline{C}_b, \overline{\boldsymbol{k}}}(k_i - \overline{k}_i)(k_j - \overline{k}_j) + \frac{\partial^2 F}{\partial C_s \partial C_b}\bigg|_{\overline{C}_s, \overline{C}_b, \overline{\boldsymbol{k}}}(C_s - \overline{C}_s)(C_b - \overline{C}_b) + \\
& \sum_{i=1}^{n} \frac{\partial^2 F}{\partial k_i \partial C_s}\bigg|_{\overline{C}_s, \overline{C}_b, \overline{\boldsymbol{k}}}(C_s - \overline{C}_s)(k_i - \overline{k}_i) + \sum_{i=1}^{n} \frac{\partial^2 F}{\partial k_i \partial C_b}\bigg|_{\overline{C}_s, \overline{C}_b, \overline{\boldsymbol{k}}}(C_b - \overline{C}_b)(k_i - \overline{k}_i) \Bigg].
\end{aligned}
\tag{A2}
$$





In Eq. (A2), the first order partial derivatives (second, third and fourth term between the square brackets) disappear, because the partial derivatives evaluated at the mean state variables are constants that are multiplied by the expectation of the deviation of a quantity, which is zero $\left( \frac{\partial F}{\partial \chi} \Big|_{\overline{\chi}} \int \int (\chi - \overline{\chi}) \, dx dy = 0, \text{ where } \chi \text{ is } C_S, C_b \text{ or } k \right)$.

Finally, after applying the averaging operator, deviations multiplying the second order partial derivatives become spatial

variances and covariances. As a result, the macro-scale C flux $\overline{F}(C_s, C_b, \boldsymbol{k})$ is obtained

$$
\overline{F}(C_s, C_b, \boldsymbol{k}) = F(\overline{C}_s, \overline{C}_b, \overline{\boldsymbol{k}}) + \frac{1}{2} \frac{\partial^2 F}{\partial C_s^2} \Big|_{\overline{C}_s, \overline{C}_b, \overline{\boldsymbol{k}}} \sigma_{C_s}^2 + \frac{1}{2} \frac{\partial^2 F}{\partial C_b^2} \Big|_{\overline{C}_s, \overline{C}_b, \overline{\boldsymbol{k}}} \sigma_{C_b}^2 + \sum_{i=1}^{n} \sum_{j=1}^{n} \frac{\partial^2 F}{\partial k_i \partial k_j} \Big|_{\overline{C}_s, \overline{C}_b, \overline{\boldsymbol{k}}} \overline{k_i' k_j'}
$$

$$
\frac{\partial^2 F}{\partial C_s \partial C_b} \Big|_{\overline{C}_s, \overline{C}_b, \overline{\boldsymbol{k}}} \overline{C_s' C_b'} + \sum_{i=1}^{n} \frac{\partial^2 F}{\partial k_i \partial C_s} \Big|_{\overline{C}_s, \overline{C}_b, \overline{\boldsymbol{k}}} \overline{k_i' C_s'} + \sum_{i=1}^{n} \frac{\partial^2 F}{\partial k_i \partial C_b} \Big|_{\overline{C}_s, \overline{C}_b, \overline{\boldsymbol{k}}} \overline{k_i' C_s'}, \tag{A3}
$$

Equation (A3) can be used to obtain the macro-scale C flux given the decomposition function $D$ at micro-scale. Explicit solutions for the multiplicative kinetics are reported in the main text (Eq. (15), (17), and (18)).

For illustration, here we report the derivation of the spatially averaged rate of decomposition for Michaelis-Menten (MM) kinetics. The micro-scale rate of decomposition for MM kinetics is given by

$$
F = \frac{k_{s,mm} C_s C_b}{K_M + C_s} \tag{A4}
$$

where both the parameters $k_{s,mm}$ and $K_m$ and the state variables $C_s$ and $C_b$ are considered spatially variable quantities. Inserting Eq. (A4) into Eq. (12) gives the macro-scale rate of decomposition

$\overline{F}(C_s, C_b, [k_{s,mm}, K_M]) =$

$$
F(\overline{C}_s, \overline{C}_b, \overline{k}_{s,mm}, \overline{K}_M) + \frac{1}{2} \frac{\partial^2 F}{\partial C_s^2} \Big|_{\overline{C}_s, \overline{C}_b, \overline{k}_{s,mm}, \overline{K}_M} \sigma_{C_s}^2 + \frac{1}{2} \frac{\partial^2 F}{\partial k_{s,mm}^2} \Big|_{\overline{C}_s, \overline{C}_b, \overline{k}_{s,mm}, \overline{K}_M} \sigma_{k_{s,mm}}^2 +
$$

$$
\frac{1}{2} \frac{\partial^2 F}{\partial k_M^2} \Big|_{\overline{C}_s, \overline{C}_b, \overline{k}_{s,mm}, \overline{K}_M} \sigma_{k_M}^2 + \frac{\partial^2 F}{\partial k_{s,mm} \partial k_M} \Big|_{\overline{C}_s, \overline{C}_b, \overline{k}_{s,mm}, \overline{K}_M} \overline{k_{s,mm}' K_M'} +
$$

$$
\frac{\partial^2 F}{\partial C_s \partial C_b} \Big|_{\overline{C}_s, \overline{C}_b, \overline{k}_{s,mm}, \overline{K}_M} \overline{C_s' C_b'} + \frac{\partial^2 F}{\partial k_{s,mm} \partial C_s} \Big|_{\overline{C}_s, \overline{C}_b, \overline{k}_{s,mm}, \overline{K}_M} \overline{C_s' k_{s,mm}'} +
$$

$$
\frac{\partial^2 F}{\partial k_{s,mm} \partial C_b} \Big|_{\overline{C}_s, \overline{C}_b, \overline{k}_{s,mm}, \overline{K}_M} \overline{C_b' k_{s,mm}'} + \frac{\partial^2 F}{\partial K_M \partial C_s} \Big|_{\overline{C}_s, \overline{C}_b, \overline{k}_{s,mm}, \overline{K}_M} \overline{C_s' K_M'} +
$$

$\displaystyle \frac{\partial^2 F}{\partial K_M \partial C_b} \Big|_{\overline{C}_s, \overline{C}_b, \overline{k}_{s,mm}, \overline{K}_M} \overline{C_b' K_M'}. \tag{A5}$





The partial derivative of $F$ with respect to $k_{s,mm}$ is zero because $F$ is a linear function of $k_{s,mm}$. Now, for biophysical heterogeneous and biochemical homogeneous system, covariances and variances related to parameters are zeros so that we are left with

$$\overline{F}(C_s, C_b, [k_{s,mm}, K_M]) = F(\overline{C}_s, \overline{C}_b, \overline{k}_{s,mm}, \overline{K}_M) + \frac{1}{2}\frac{\partial^2 F}{\partial C_s^2}\bigg|_{\overline{C}_s,\overline{C}_b,\overline{k}_{s,mm},\overline{K}_M} \sigma_{C_s}^2 + \frac{\partial^2 F}{\partial C_s \partial C_b}\bigg|_{\overline{C}_s,\overline{C}_b,\overline{k}_{s,mm},\overline{K}_M} \overline{C_s' C_b'}. \quad (A6)$$

5  Calculating the derivatives gives

$$\overline{F}(C_s, C_b, [k_{s,mm}, K_M]) = \frac{k_{s,mm}\overline{C}_s\overline{C}_b}{K_M + \overline{C}_s} + \frac{1}{2}\left[\frac{-2k_{s,mm}K_M\overline{C}_b}{K_M + \overline{C}_s^2}\right]\sigma_{C_s}^2 + \left[\frac{k_{s,mm}K_M}{(\overline{C}_s + K_M)^2}\right]\overline{C_s' C_b'}. \quad (A7)$$

For biophysical homogeneous and biochemical heterogeneous system, covariances and variances of state variables ($C_s$ and $C_b$) are zeros so that we are left with

$$\overline{F}(C_s, C_b, [k_{s,mm}, K_M]) =$$

$$F(\overline{C}_s, \overline{C}_b, \overline{k}_{s,mm}, \overline{K}_M) + \frac{1}{2}\frac{\partial^2 F}{\partial k_M^2}\bigg|_{\overline{C}_s,\overline{C}_b,\overline{k}_{s,mm},\overline{K}_M}\sigma_{k_M}^2 +$$

$$\frac{\partial^2 F}{\partial k_{s,mm}\partial k_M}\bigg|_{\overline{C}_s,\overline{C}_b,\overline{k}_{s,mm},\overline{K}_M}\overline{k_{s,mm}'K_M'} + \frac{\partial^2 F}{\partial k_{s,mm}\partial C_s}\bigg|_{\overline{C}_s,\overline{C}_b,\overline{k}_{s,mm},\overline{K}_M}\overline{C_s'k_{s,mm}'} +$$

$$\frac{\partial^2 F}{\partial k_{s,mm}\partial C_b}\bigg|_{\overline{C}_s,\overline{C}_b,\overline{k}_{s,mm},\overline{K}_M}\overline{C_b'k_{s,mm}'} + \frac{\partial^2 F}{\partial K_M\partial C_s}\bigg|_{\overline{C}_s,\overline{C}_b,\overline{k}_{s,mm},\overline{K}_M}\overline{C_s'K_M'} +$$

$$\frac{\partial^2 F}{\partial K_M\partial C_b}\bigg|_{\overline{C}_s,\overline{C}_b,\overline{k}_{s,mm},\overline{K}_M}\overline{C_b'K_M'}. \quad (A8)$$





For a completely heterogeneous system with biophysical and biochemical heterogeneity, the mean rate of decomposition at macro-scale is given by Eq. (A9).

$$\overline{F}(C_s, C_b, [k_{s,mm}, K_M]) =$$

$$F(\overline{C}_s, \overline{C}_b, \overline{k}_{s,mm}, \overline{K}_M) + \frac{1}{2} \frac{\partial^2 F}{\partial C_s^2}\bigg|_{\overline{C}_s, \overline{C}_b, \overline{k}_{s,mm}, \overline{K}_M} \sigma^2_{C_s} +$$

$$\frac{1}{2} \frac{\partial^2 F}{\partial k_M^2}\bigg|_{\overline{C}_s, \overline{C}_b, \overline{k}_{s,mm}, \overline{K}_M} \sigma^2_{k_M} + \frac{\partial^2 F}{\partial k_{s,mm} \partial k_M}\bigg|_{\overline{C}_s, \overline{C}_b, \overline{k}_{s,mm}, \overline{K}_M} \overline{k'_{s,mm} K'_M} +$$

$$\frac{\partial^2 F}{\partial C_s \partial C_b}\bigg|_{\overline{C}_s, \overline{C}_b, \overline{k}_{s,mm}, \overline{K}_M} \overline{C'_s C'_b} + \frac{\partial^2 F}{\partial k_{s,mm} \partial C_s}\bigg|_{\overline{C}_s, \overline{C}_b, \overline{k}_{s,mm}, \overline{K}_M} \overline{C'_s k'_{s,mm}} +$$

$$\frac{\partial^2 F}{\partial k_{s,mm} \partial C_b}\bigg|_{\overline{C}_s, \overline{C}_b, \overline{k}_{s,mm}, \overline{K}_M} \overline{C'_b k'_{s,mm}} + \frac{\partial^2 F}{\partial K_M \partial C_s}\bigg|_{\overline{C}_s, \overline{C}_b, \overline{k}_{s,mm}, \overline{K}_M} \overline{C'_s K'_M} +$$

$$\frac{\partial^2 F}{\partial K_M \partial C_b}\bigg|_{\overline{C}_s, \overline{C}_b, \overline{k}_{s,mm}, \overline{K}_M} \overline{C'_b K'_M}. \tag{A9}$$

## A2 An analytical expression of the substrate C at steady state for the multiplicative kinetics in the fully heterogeneous system

The substrate C at steady state for multiplicative and MM kinetics in a homogeneous system are given in Table 1 and restated here for convenience. These expressions are also valid for biophysically heterogeneous systems.

$$C^*_{s,mult} = \frac{k_B}{Y k_{s,mult}} \tag{A10}$$

$$C^*_{s,mm} = \frac{K_M k_B}{Y k_{s,mm} - k_B} \tag{A11}$$

where * represents the steady state. Eq. (A10) and (A11) show that the steady state substrate C depends only on the kinetics parameters. Thus, if the kinetic parameters are spatially variable (i.e., fully heterogeneous system) then $C^*_{s,mult}$ and $C^*_{s,mm}$ are also spatially variable and different from the steady state values in biophysically heterogeneous and homogeneous systems. Knowing the probability distributions of the kinetic parameters, the mean steady state substrate C in the fully heterogeneous system can be calculated as the mean value of $C^*_{s,mult}$ or $C^*_{s,mm}$.

The mean value of a generic function, $g(x)$ is given by $\overline{g(x)} = \int\limits_{-\infty}^{\infty} g(x) f_X(x) dx$, where $f_X(x)$ is the probability density function of $x$. For the multiplicative kinetics and assumed a $logUniform(a, b)$ distribution for $k_{s,mult}$, the mean value of





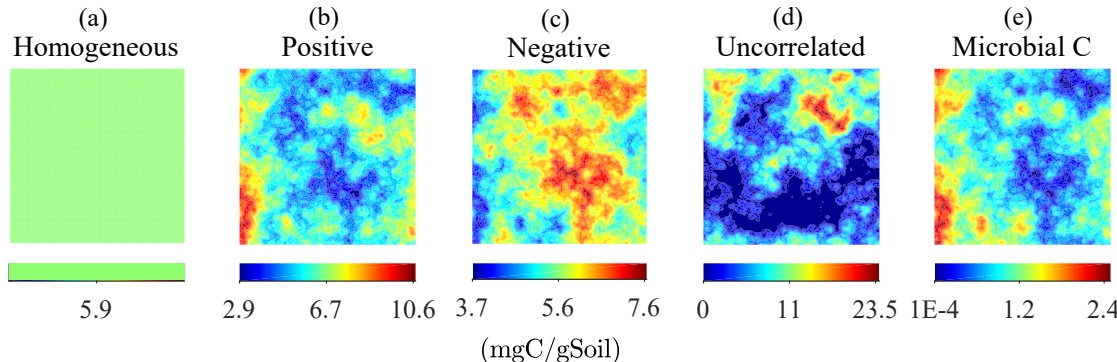

**Figure A1.** Steady state initial condition: Examples of the homogeneous (a) and the heterogeneous distributions of substrate C constrained to have the same amount of total substrate (b–d). The fields in b–d were obtained by imposing different degrees of correlation with the initial heterogeneous distributions of microbial C (e).

**Table A1.** List of parameters. Values in the brackets correspond to the units reported in the brackets.

| Parameter | Value | Unit | Description |
|---|---|---|---|
| $I$ | $6.06 \times 10^{-4}$ | $\mathrm{mgCg^{-1}soil\,h^{-1}}$ | rate of input of external carbon |
| $K_m$ | 5156250 (25) | fgC/grid cell ($\mathrm{mgCg^{-1}soil}$) | half saturation constant |
| $k_{s,mm}$ | 0.018 | $\mathrm{h^{-1}}$ | decomposition rate constant for the Michaelis-Menten model |
| $K_{s,mult}$ | $7.45 \times 10^{-10}$ ($1.53 \times 10^{-4}$) | $\mathrm{h^{-1}\,(fgC/grid\,cell)^{-1}}$ ($\mathrm{h^{-1}\,(mgCg^{-1}soil)^{-1}}$) | decomposition rate constant for the Multiplicative model |
| $k_B$ | 0.00028 | $\mathrm{h^{-1}}$ | decomposition rate constant for the Michaelis-Menten model |
| $Y$ | 0.31 | $\mathrm{h^{-1}}$ | $\mathrm{g/cm^3}$ |
| $\rho_{BD}$ | 1.65 | $\mathrm{g/cm^3}$ | Soil bulk density |
| $\rho_{OM}$ | 1.1 | $\mathrm{g/cm^3}$ | Organic matter density |

$C^*_{s,mult}$ is given by,

$$\overline{C}^*_{s,mult} = \int_a^b \frac{k_B}{Y k_{s,mult}}\, f(k_{s,mult}) dk_{s,mult} \tag{A12}$$

where $f(k_{s,mult})$ is the probability density function of $k_{s,mult}$ and given by $f(k_{s,mult}) = \dfrac{1}{(b-a)k_{s,mult}\log_e 10}$. Inserting the expression of $f(k_{s,mult})$ in Eq. (A12) gives,

5    $$\overline{C}^*_{s,mult} = \frac{k_B \left[10^{-a} - 10^{-b}\right]}{Y(b-a)\ln(10)} \tag{A13}$$





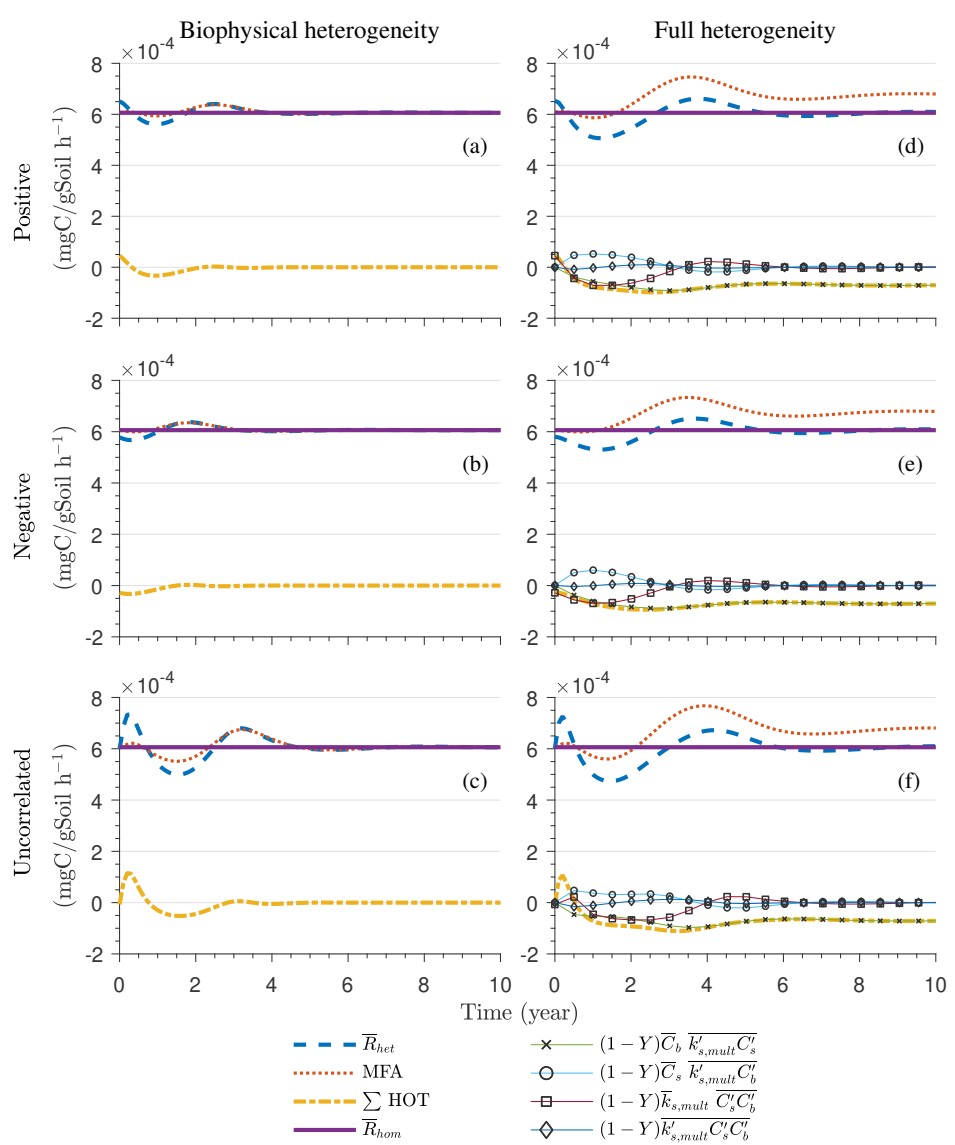

**Figure A2.** Scenario 1 (initial condition at steady state): temporal evolution of mean respiration rate in the heterogeneous system ($\overline{R}_{het}$, including the mean-field approximation (MFA) and second order terms), and the respiration rate in the homogeneous system ($\overline{R}_{hom}$),for multiplicative kinetics and for (a–c) biophysical and (d–f) fully heterogeneous system with (a–d) positively and (b–e) negatively correlated, or (c–f) uncorrelated initial substrate and microbial C.



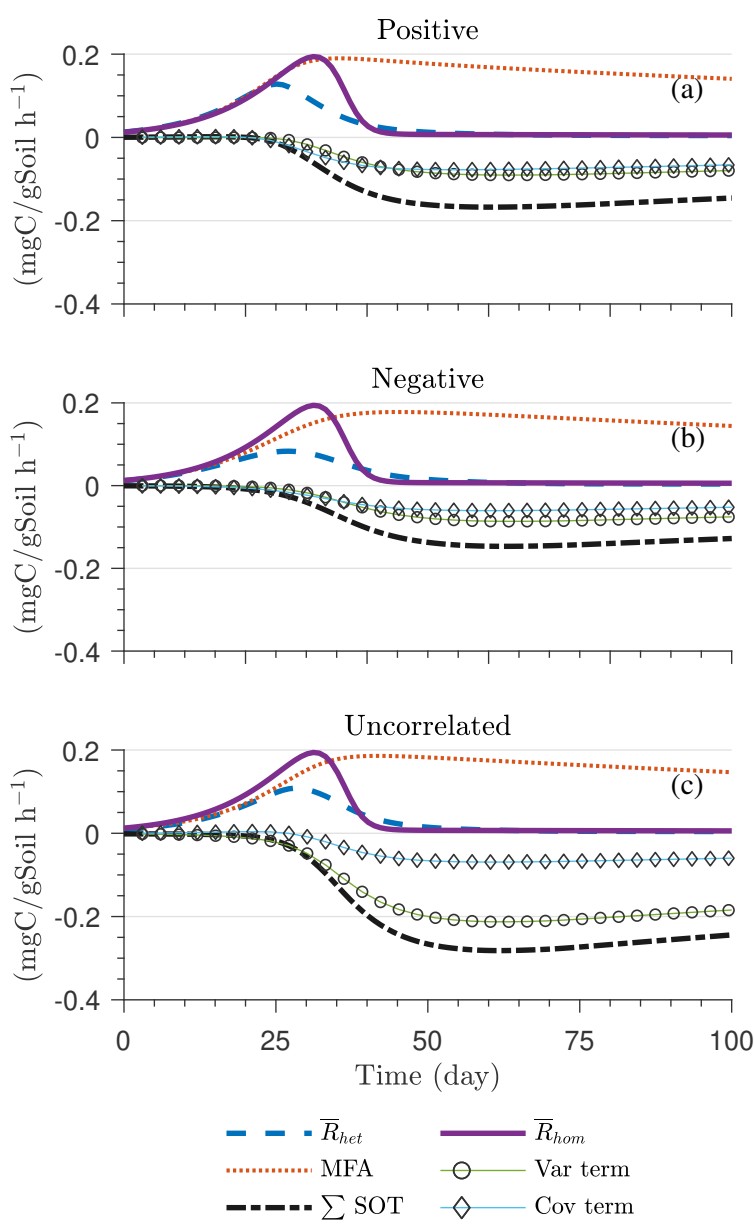

**Figure A3.** Scenario 2 (transient dynamics with Michaelis-Menten kinetics): temporal evolution of mean respiration rate in the biophysically heterogeneous system ($\overline{R}_{het}$, including the mean-field approximation (MFA), and second order terms), and the respiration rate in the homogeneous system ($\overline{R}_{hom}$), for (a) positively and (b) negatively correlated, or (c) uncorrelated substrate and microbial C.



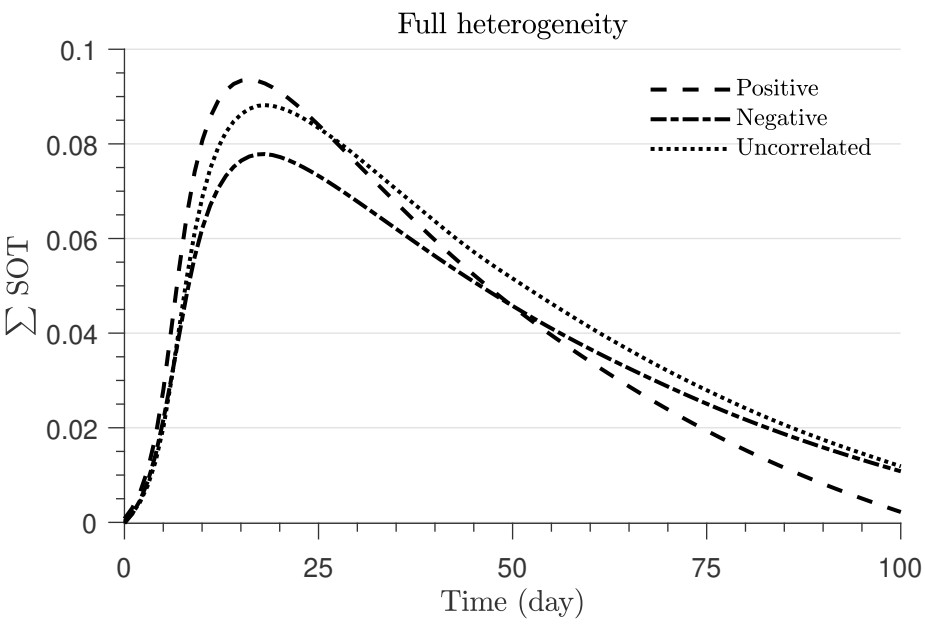

**Figure A4.** Scenario 2 (transient dynamics with Michaelis-Menten kinetics): temporal evolution of the sum of second order terms for the fully heterogeneous system for different degrees of correlation between the initial substrate and microbial C. Only the sum of second order terms ($\sum SOT$) is shown because of large number of spatial moments involved.

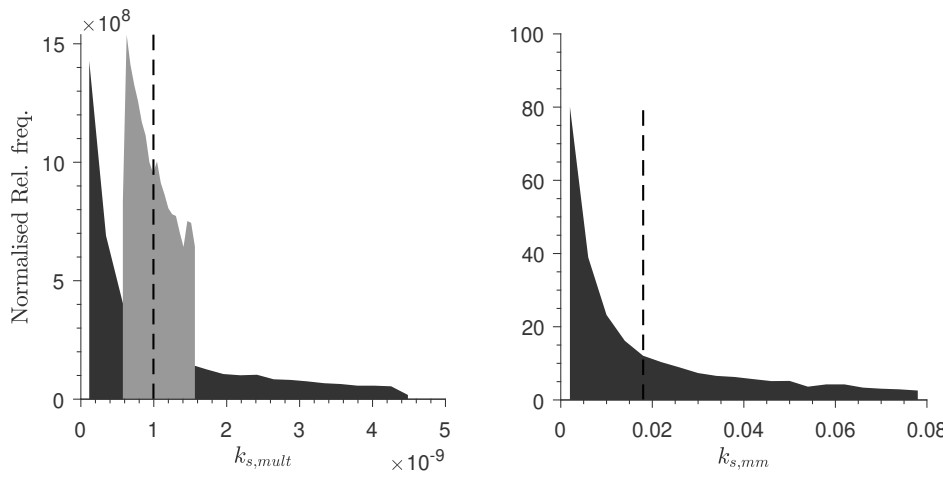

**Figure A5.** Distribution of the decomposition rate constant for different degrees of biochemical heterogeneity, and for (a) multiplicative and (b) Michaelis-Menten kinetics. Black and grey shadings represent higher and lower degree of biochemical heterogeneity respectively, and the dashed line represents the mean rate constant for the homogeneous system. The half saturation constant $K_M$ is uniformly distributed, not shown in figure.



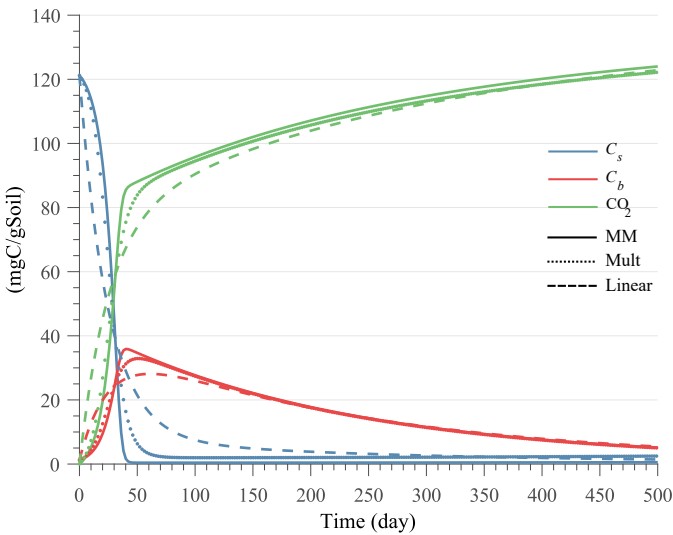

**Figure A6.** Comparison of soil organic matter decomposition dynamics of linear, multiplicative and Michaelis-Menten kinetics. The rate constant of decomposition for multiplicative and linear kinetics was estimated by fitting the of the dynamic equations for the two models to the $CO_2$ production rate simulated by MM kinetics.

**Table A2.** Initial mean substrate and microbial C in scenarios one and two (in fgC/grid cell); values in brackets are expressed in $mgCg^{-1}soil$.

| # | Scenario | Initial $\overline{C}_s$ | Initial $\overline{C}_b$ |
|---|----------|--------------------------|--------------------------|
| | | fgC/grid cell($mgCg^{-1}soil$) | |
| 1 | Steady state IC | $1.212 \times 10^5 (5.9)$ | $2.005 \times 10^5 (0.9725)$ |
| 2 | Transient IC | $2.5 \times 10^7 (121.21)$ | $2.5 \times 10^5 (1.21)$ |

**Table A3.** Probability distributions of the parameters for the multiplicative and MM kinetics models. Values in brackets indicate the minimum and maximum parameter values.

| | | Multiplicative | MM | |
|---|------|------------------------------------------------|---------------------------|---------------------------|
| # | Case | $k_{s,mult}$ (h$^{-1}$) (fgC/grid cell)$^{-1}$ | $k_{s,mm}$ (h$^{-1}$) | $K_M$ (fgC/grid cell) |
| 1 | Biochemical heterogeneity 1 | $log10Uniform(-10.1, -8.56)$ | $log10Uniform(-1.098, -3)$ | $Uniform(0.25, 49.75)$ |
| 2 | Biochemical heterogeneity 2 | $log10Uniform(-9.4, -8.9)$ | | |





*Author contributions.* All authors conceived the initial conceptualization. Arjun Chakrawal and Stefano Manzoni developed the theoretical

formalism, performed the analytic calculations and the numerical simulations. Arjun Chakrawal took the lead in writing the manuscript. All

authors discussed the results, provided critical feedback, and revised the manuscript.

*Acknowledgements.* This research was supported by the Swedish Research Councils Vetenskapsrådet (grant 2016-04146) and Formas (grant

5  2015-468).





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
