# Peer review of "Dynamic upscaling of decomposition kinetics for carbon cycling models"

_Geoscientific Model Development, 2019_

## Referee Comment (RC1) · Thomas Wutzler (Referee) · 6 Jul 2019

The authors study one very basic assumption of soil organic matter (SOM) modelling at theoretical level. They falsify the assumption that ynamic equations that work for the pore-scale can be used a the soil corescale. This is because soil is heterogeneous and properties vary much at a scale longer than diffusion mixing length. Moreover, they present a framework that is able to derive equations at the soil core scale, albeit currently only for simple models and without the required information about the dynamics of the heterogeneity. This work is very interesting to the SOM modelling community, because it shows basic shortcomings in current model development and interpretation of simulation results. It could be a start for the development of a new class of SOM models. I enjoyed reading the manuscript and looking at the differences in results be-

tween homogeneous and heterogeneous simulation. The length is quite long and it was difficult to keep the storyline in mind. With the detailed explanations, the point was made clear, I understood and trust the claims, and got curious with even more questions to be discussed.

General comments

The methods and result sections are rather long. It would be nice to shorten it by deciding on the important points and moving less important points to the appendix. I give some suggestions in the detailed comments.

Methodology description: the way of providing spatial moments to analytic equations did not become clear to me (P17L11). I assume, you computed the quantities for sufficiently close time points from the distributed model, and provided a smoothing function depending on time as input to the solver for the analytic equation system.

An overview of the approach would be helpful: 1) Express each equation of state variable change of each individual location based on the spatial mean of the pool sizes and the deviations from it at local scale. And 2) Apply a spatial averaging over the obtained equations, resulting in an equation composed of terms of the mean pool sizes and the spatial covariance of the pools and heterogeneously distributed parameters.

I tried to check the math, but did not always come to the same results (see detailed comments, eq. A7).

In the discussion I would like to read about several points:

Slowdown of decomposition: To my opinion the slowdown of decomposition despite plenty of available substrate (Fig 7d) is a very important feature/insight of the model. A very simple model (albeit still required input of time series of heterogeneity variances) can explain why we can find very old potentially quickly decomposable SOM. The reasons should be explained in more detail (right skewed distribution of decomposition rate, low probability of co-occurrence of high substrate concentration, . . .)
Oscillations at multi-annual time scale: Observations of such a phenomenon are very rare. I once argued that we do not see such modelled oscillations with microbial explicit models because of superposition of dynamics across many pores. Here, such spatial heterogeneity is the cause of fluctuations.

Development of the heterogeneity / damping of oscillations (Fig 4): The systems develops to a steady state without any more oscillations. Is the initial heterogeneity developing in direction of homogeneity? Probably not because the simulated SOM stocks differ from the homogeneous system. What is the spatial distribution and covariance between substrate, biomass, and quality after 60 years? Is there a covariance pattern that is stable? I suggest putting another two panels to Fig. 2 showing microbial and substrate distribution at year 60.

Role of disturbances: What happens if you simulate a disturbance (homogenization) after the system is near steady state? Does this start the oscillating pattern again?

Magnitude of the heterogeneity effects: In Figure 4, the effects look large, because the axis ranges from 5 to 7, but aside from the initial disturbance, the effect is only about 1/10 of the steady state. Are there reasonable parameter combinations where the effect is larger? Or do we not need to care this much about heterogeneity at steady state?

2D system: Are the insights transferable to a 3D system. What would you expect to change? Since, there is currently no transport and interaction between the cells, I infer that aside from maybe slightly different development of the initial correlations, the dynamics should stay the same. The macro-scale equations are not affected, as I understood.

More complex systems: The analytical scale transition approach worked nicely with the basic simple model. With more complex models that include many more heterogeneous parameters it will be difficult to impossible to close the model with all the combination covariances (the factorial grows very fast). Can you describe a strategy

to determine which combinations are important and which combinations can be neglected? When have we sufficiently including more and more heterogeneities?

Time scale: I am especially interested in modelling decadel to longer-term SOM dynamics. Are the multi-annual oscillations important for the longer term dynamics? Do you expect heterogeneity to change with global change in the longer term? What is the advantage of describing the changed steady state with heterogeneity (Fig 7d) with heterogeneity inputs compared to effective model parameters? I see some advantages, but it would be nice to clarify them in the paper.

Specific comments:

eq. 4 ..6: Your simple basic model refers to the Schimel and Weintraub 2003, who actually used and suggested an inverse MM kinetics D = ks Cs Cb / (kM + Cb). It would be nice to amend your work by this decomposition equation.

P11L11: The sentence does not make sense to me. The variance itself is not always negative. Probably you ment: "This term is always negative because the variance of the spatial substrate distribution is a positive quantity and ..."

P12L15ff: May state that therefore the mean field approximation is exact and spatial variance of this parameter has no effect on the macro-scale dynamics.

P12L19: This paragraph comes a bit surprising without context. Why do you look at SGR?

P14L11ff: Potential for moving to appendix. Only the information starting from P15L5 is important.

P14L19: What is (fg)? I could not find the explanation. It is used several times in the text eq. 26, 27 and appendix figure and table captions.

P16L10ff: I suggest to give more meaningful names to the scenarios instead of numbers. E.g "Steady simulation" and "High Substrate Simulation" (also update Fig 3).

P17L11: It did not

P18L14: The equation is not understandable here (what is a and b). I suggest just referring to the appendix equation.

P19L1: I suggest to shortly inline write the definition of SGR or refer to Eq. 20.

P22L4ff: The details are hard to follow. I suggest moving this paragraph to the appendix near the referenced figures. However, I suggest keeping the conclusion that additional higher order terms cannot be ignored.

Fig5 is nice but actually does not add insight in addtion to the dynamics already shown in the other figure 4. Another possibility to shorten the manuscript and moving the Figure to the appendix.

P22L15: I suggest to remove the first sentence – its essentially a figure caption. Better start with the subject of the paragraph: the leation between SGR and Cs. Why its important to look at SGR?

Fig8: I assume this refers to the transient simulations. Please indicate this in the figure caption. Further, I suggest adding a third column and move SOT lines out of the second panel and add color redundant to linetype, because its really hard to distinguish the lines. First, I was confused by R_hom in plots of correlations. I assume it was added for comparison and should be the same across all rows (it looks shifted, but that may be an optical illusion). Please, clarify in the figure caption. I initially thought from your description in 2.6 that in the transient scenario decomposed a large initial SOM field but did not receive any further inputs. However, this contradicts the behaviour in Fig. 8. Please clarify in 2.6. The respiration seems to run into a steady state, that is independent of the heterogeneity. Initially I was puzzled by the same respiration despite very different substrate and biomass values, but if this in steady state with an input rate this is the expected behaviour. Would be nice to read a little discussion on this.

Fig9: May add an arrow to one line to indicate the development in time.

P27L12: "Early state of decomposition process": In my interpretation of your results its more a "Period after disturbance" - but this relates to the missing presentation of development of spatial heterogeneity.

P28L14: Cited literature. There is another group who did much work on pore-scale modelling with H.J. Vogel in Leipzig although in the hydrological domain. He also suggested a "scaleway" (e.g. 2003 Moving through scales). Would be nice to reference some of their work for following links in literature.

P28L23: typo: differences "are" (instead of "is")

P287L21: Here you discuss high frequency measurements of respiration in the range of minutes for moving forward to measure effects of heterogeneity. However, your simulated dynamics of heterogeneity works at multi-annual time scale. How does this relate to each other?

eq. A7: When checking the derivatives, I a get a different dF/dCSdCS with a different denominator of the second term: (kM+CS)^3. I assume its a typo in the manuscript, but please, check your code, or show that your derivation is indeed correct.

Fig A1: How does this figure differ from Fig 2 in the main text?

eq. A13: Where does the log_e10 in the logUniform distribution come from? Do you expect your results to change, if you use the log to base e instead of the log to base 10?

---

## Author Comment (AC1) · 25 Jul 2019

We would like to thank Thomas Wutzler for an encouraging and detailed review of our manuscript. This response is intended to be short and part of the full response which would be submitted after the other reviews become available as well. In the following, we address the most conceptually important points raised by Dr. Wutzler (denoted by italic font). Our responses are highlighted in blue (normal font).

1. *Methodology description: the way of providing spatial moments to analytic equations did not become clear to me (P17L11). I assume, you computed the quantities for sufficiently close time points from the distributed model, and provided a smoothing function depending on time as input to the solver for the analytic equation system*

   There are two ways to illustrate how heterogeneities affect soil organic matter kinetics. One is to solve directly the upscaled equations; the other is to numerically solve the micro-scale equations and aggregate (=average) results at the macro-scale. Solving the upscaled differential equation would require some mechanism of transferring second order moments calculated from distributed model to the upscaled differential equation – this issue of 'model closure' is presented in the Discussion. Here we followed the second approach and did not solve the upscaled differential equations (Eq. (9) and (10), P8L15), as explained in the manuscript (P13L4 and P30L22). We only used the averaged dynamics as simulated from distributed model and compared that with the homogeneous equivalent.

   This choice might seem to counter the purpose of this work – i.e., propose an analytical approach to the upscaling problem. However, the lack of model closure (admittedly the main limitation of our approach, as acknowledged in the Discussion) does not allow a full solution of the upscaled analytical model. Thus we used numerical solutions to illustrate the effect of heterogeneities, and the analytical equations to provide a theoretical framework for studying the problem.

   It should be pointed out that the numerical averaging approach yields exactly the same solution of the upscaled analytical equations. We tested this for the case of biophysical heterogeneity with multiplicative kinetics because the upscaled equation are exact and only the covariance of substrate and biomass is needed as additional information. As expected, results were exactly the same from the upscaled differential equation and the distributed model.

2. *I tried to check the math, but did not always come to the same results (see detailed comments, eq. A7)*

   It is a typing mistake. Eq. A7 should be $-\frac{2\,k_{S,mm}K_M\,\bar{C}_B}{(K_M+\bar{C}_S)^3}$. Numerical code and results are correct.

3.  *Development of the heterogeneity / damping of oscillations (Fig 4): The systems develops to a steady state without any more oscillations. Is the initial heterogeneity developing in direction of homogeneity? Probably not because the simulated SOM stocks differ from the homogeneous system. What is the spatial distribution and covariance between substrate, biomass, and quality after 60 years? Is there a covariance pattern that is stable? I suggest putting another two panels to Fig. 2 showing microbial and substrate distribution at year 60.*

Yes, the covariance pattern for biophysical heterogeneity and the sum of the higher order terms are stable in the long term, after new steady state has been reached. We ran the simulation for 100 years and a similar figure to Figure 4 is shown below which confirm that higher order terms are indeed stable.

Because C input is homogeneously distributed, the microbial and substrate concentration spatial distributions would be also homogeneous. However, they attained a different concentration value at steady state. This concentration value is shown in Figure 4. Adding panels showing a spatially homogeneous distribution might not add much information, but we can better explain this pattern in a revised manuscript.

[Figure]

**Figure R1: This figure is same as to Figure 4 in the manuscript but for 100 years simulation period.**

4.  *Role of disturbances: What happens if you simulate a disturbance (homogenization) after the system is near steady state? Does this start the oscillating pattern again?*

No, oscillations would not start again after the system has reached a steady state, because with spatially homogeneous inputs the steady state will also be spatially homogeneous. Thus, no oscillations would occur whether the system has only biophysical heterogeneities or is fully heterogeneous.

5. *Magnitude of the heterogeneity effects: In Figure 4, the effects look large, because the axis ranges from 5 to 7, but aside from the initial disturbance, the effect is only about 1/10 of the steady state. Are there reasonable parameter combinations where the effect is larger? Or do we not need to are this much about heterogeneity at steady state?*

To answer, we ran two scenarios in which we changed the kinetic constant parameter $k_{s,mult}$, 1) decreasing $k_{s,mult}$ in the biophysical heterogeneity (Figure R2 and Figure R3) and 2) increasing the heterogeneity of $k_{s,mult}$ (by increasing its standard deviation) in the full heterogeneity (Figure R4). From Fig. R3 and R4, it is clear that decreasing the rate constant increases the amplitude and wavelength of the oscillations. As shown in Figure R4, increasing the heterogeneity of the rate constant (right column) increases the amount of undecomposed substrate C compared to a lower degree of heterogeneity (middle column). This pattern can be explained using the analytical expression of the steady state substrate C (see Eq. (A13) in Appendix A2, P36L5). For the increased heterogeneity case shown in the right column, we used values of $a = -10.1$ and $b = -8.56$, where $a$ and $b$ have the same meaning as in Eq. (A13). The analytical expression for the steady state, evaluated with these values of $a$ and $b$, results in exactly the same steady state of substrate C as simulated by the distributed model (i.e. 15 mgC/gSoil).

These fluctuations are similar to those noted in earlier papers using spatially lumped models (Manzoni and Porporato, 2007; Sierra and Mueller, 2015). These papers showed that the occurrence and amplitude of the fluctuations depends on the kinetic parameter values, as is the case here.

[Figure]

**Figure R2 This figure is similar to Figure 4 left column in the manuscript. Different line colors represent varying levels of rate constant $k_{s,mult}$ with base case same as in Figure 4 of manuscript. Scenario 1 (initial condition at steady state): effect of biophysical on the macroscopic decomposition dynamics when the substrate is distributed randomly around the steady state: (a) mean substrate C ( $C_s$ ), (b) mean microbial C ( $C_b$ ), (c) mean respiration rate (R), and (d) sum of second and third order terms ($\sum$HOT).**

[Figure]

**Figure R3 Enlarged view of the time trajectories of the mean substrate C concentration extracted from figure R2.**

[Figure]

**Figure R4: This figure is similar to Figure 4 in the manuscript, except for the right column, where we added trajectories for the full heterogeneity case, with increased heterogeneity of the rate constant ($k_{s,mult}$).**

6.   *More complex systems: The analytical scale transition approach worked nicely with the basic simple model.*
     *With more complex models that include many more heterogeneous parameters it will be difficult to impossible*
     *to close the model with all the combination covariances (the factorial grows very fast). Can you describe a*

*strategy to determine which combinations are important and which combinations can be neglected? When have we sufficiently including more and more heterogeneities?*

This is an excellent point. We hope that this work will stimulate discussion precisely in this direction. At a large enough scale, most likely some higher order moments will not matter as much, possibly leading to model simplifications.

Which higher order moments can be neglected would depend upon the kinetics and type of heterogeneity studied. If the kinetics of decomposition at microscale is known, then one could use the upscaling procedure provided in the manuscript and get a second order approximation of upscaled decomposition rate. Afterwards, depending upon the nature of heterogeneity (i.e., biophysical, biochemical or a combination of the two), one could start eliminating the unnecessary covariances and variance terms to arrive at a manageable upscaled decomposition rate.

When have we included enough heterogeneities? This is more difficult question to answer. From a modeling perspective, the Taylor expansion could be truncated when results from distributed model and upscaled equation start to converge (or differ less than a preset error). To give an example, for a fully heterogeneous system simulated with multiplicative kinetics, $\bar{R} \neq (1-Y)\left(\bar{k}_{s,mult}\bar{C}_s\bar{C}_b + \bar{C}_s\overline{k'_{s.mult}C'_b} + \bar{C}_b\overline{k'_{s.mult}C'_s} + \bar{k}_{s,mult}\overline{C'_sC'_b}\right)$ because the third order moment $\overline{k'_{s.mult}C'_sC'_b}$ is missing in the summation. However, it is possible that the dynamics at the micro scale lead to low values of higher order moments, because substrate consumption, mortality of the microorganisms, and transport (not explicitly modelled here) contribute to smoothing spatial gradients.

In order to investigate if enough heterogeneities are included in the upscaled differential equation, a figure can be included in the revised manuscript exploring the effects of progressively adding higher order moments on the simulated dynamics of mean respiration rate. Such a figure would illustrate how errors are reduced when adding spatial moments to the mean field approximation.

7. *Time scale: I am especially interested in modelling decadal to longer-term SOM dynamics. Are the multi-annual oscillations important for the longer term dynamics? Do you expect heterogeneity to change with global change in the longer term? What is the advantage of describing the changed steady state with heterogeneity (Fig 7d) with heterogeneity inputs compared to effective model parameters? I see some advantages, but it would be nice to clarify them in the paper.*

Yes, multi-annual oscillations are important in our simulations. As our results suggest, in the fully heterogeneous system, organic carbon reaches a new steady state that is dependent upon the micro-scale features (see Eq. A13, P36L5); i.e., heterogeneity in the kinetic parameters.

The linkages between these predictions with global change are not clear at this stage. We do not know if climatic changes alter the chemical heterogeneity of organic substrates. However, land management does. Based on our results we could speculate that less heterogeneous litter input – as for example in agricultural

fields compared to a forest – could lead to less soil organic carbon in the long term. In a revised manuscript, we could elaborate on this possibility.

As explained above, these preliminary replies are meant to continue the discussion and offer readers a glimpse at how we could address Dr. Wutzler's comments. A complete rebuttal will be provided in due time and if deemed useful by the editor.

**References**

Manzoni, S., and Porporato, A.: Theoretical analysis of nonlinearities and feedbacks in soil carbon and nitrogen cycles, Soil Biology & Biochemistry, 39, 1542-1556, 2007.
Sierra, C. A., and Mueller, M.: A general mathematical framework for representing soil organic matter dynamics, Ecological Monographs, 85, 505-524, 10.1890/15-0361.1, 2015.

---

## Referee Comment (RC2) · Ali Ebrahimi (Referee) · 23 Aug 2019

Referee: Ali Ebrahimi (MIT, alieb@mit.edu)

Summary
The manuscript entitled "Dynamic upscaling of decomposition kinetics for carbon cycling models" by Chakrawal et al. deals with upscaling the role of soil micro-scale structure and heterogeneity in microbial and carbon distributions to macroscale carbon cycle models. The authors developed a novel mathematical model based on 2D mesh grid system to account for microscale heterogeneities and they used scale transition theory to provide an analytical framework for decomposition kinetics at macro-scale. The idea of upscaling is highly novel and the author's work on trying

to show that microscale heterogeneities could play a significant role in large scale carbon dynamics is interesting and informative. To extend, I could re- derive, the mathematical model is formulated correctly and it should be accessible to audience with average knowledge on mathematical modeling. While the model is advanced, I do have some major concerns on the relevance of the analytical model for large scale carbon dynamics and the ability of the model to represent real soil with empirically measurable quantities. The lack of systematic parameterizations based on such quantities, makes the current conclusions rather speculative and hard to translate to relevant environmental scenarios in natural ecosystems.

Detailed review
Major concerns:
I) it is unclear to what extend the parameterization proposed in the analytical kinetic model could be experimentally validated. My major problem is that some of the quantities do not have real biogeochemical or physical meanings in which could be experimentally measured. For instance there is an emphasis on the second order moments as state variables used to close the model; however it is hard to think how such variable could be experimentally measured.
II) The type of model and scenarios proposed in this study are relevant and could potentially address some of the inconsistency in our field measurements but it could only be possible if the model could establish a systematic link to relevant abiotic and biotic factors observed in the field. While in the discussion authors have tried to relate some of the scenarios in the study to soil aggregation or pore connectivity and an entire subsection is dedicated for that, I still find that the modeling framework is too abstract that makes the explanations quite speculative and hard to think to what extend the decomposition rate may vary under realistic settings.
III) The model could potentially describe some of the underlying abiotic and physio- logical mechanisms that shape the decomposition dynamics but in the current form of the manuscript this has not been explored. For instance, I was wondering to what

extend half saturation to substrate and decomposition rate constant (KM and Ks) are shaping the dynamics observed in the model. I would guess if lower KM or high Ks would have been chosen the heterogeneous scenarios would have converged faster to the homogenous one.

Minor comments:
- I was wondering to what extend the fluctuating environmental condition (for instance fluctuating in carbon distributions) could play a role in shaping the carbon decomposition dynamics. Do you expect to see faster convergence to homogenous scenario in high intensity fluctuations?
- The system size that is modeled in grid based network is rather small. The number of grids, or pores equal to 10000, is basically enough to model an aggregate with the size of 0.5mm. I was wondering if this size is sufficiently large to capture heterogeneities in the soil? For instance inter aggregate pores or macro-pores?
- Following up on the results for negative correlations, I was wondering how much physical inaccessibility of the carbon to microbes could be relevant for the soil systems? For instance, most of the carbon protections in soil are often driven by soil aggregation and creation of anoxic microsites. In a broader term, the counter gradients created by carbon and other necessary substrates for carbon degradation could lead to inaccessibility of the carbon for microbes and not necessary physical inaccessibility. This is a phenomenon that has been previously shown in soil aggregates that due to creation of anoxic zones, the carbon configuration does not play a role in carbon consumption (Ebrahimi and Or, 2018 GCB) and in other studies showing carbon protection by aggregation (e.g., Keiluweit et al., 2017 Nat. comm.).
Recommendations
The current study aims to take an important step to propose upscaling strategy of carbon decomposition rates from microscale. This study opens up a direction for follow up studies to provide more realistic parameterization of large scale carbon models. I think the current model could be used to study many aspects of soil microbial

processes at micro-scale and help to create hypothesis that could be addressed experimentally. This being said, the manuscript should use more focus on what the model could offer at the current form and the limitations and avoid speculations. The manuscript should also provide incremental steps toward better connecting the current model to measurable quantities and field observations.

Please also note the supplement to this comment:
https://www.geosci-model-dev-discuss.net/gmd-2019-133/gmd-2019-133-RC2-supplement.pdf

---

## Referee Comment (RC3) · Anonymous Referee #3 · 25 Aug 2019

As someone who has been thinking and working on biogeochemistry scaling for quite a while, I applaud the authors' willing to attack the problem of spatial upscaling. The authors applied a well-established theory to analyze how high-order microstructure would affect the emergent organic matter decomposition dynamics in soil under three assumed mathematical formulations. By analyzing several synthetic cases, they concluded that for non-linear formulations, the spatial mean equations generally fail to capture the true dynamics.

While the conclusions they drew are solid within their model configuration, I too share with others the concern that how this learned lesson could be translated into something universally applicable for other modelers. In particular, we in the soil biogeochemical modeling community have so far no unanimously accepted governing equation to solve

like that exist for geophysical fluid dynamics, or hydrodynamics in general, where resolving the microstructure effects can be achieved through the so called large-eddy simulation and sub-grid closure, and even field or laboratory experiments can be designed to derive parameterization schemes that are generally applicable for different situations. Personally, I am therefore wondering can the authors' approach become some tools that are easily accessible to others, e.g., like Markov chain Monte Carlo codes that are widely accessible through opensource software?

Second, I am a little bit disappointed that authors decided to ignore the interactions between different micro-grids. In physics, the successful upscaling is achieved only through the consideration of interactions. For instance, the scaling of Newton's law of momentum conservation, the derivation of center of gravity, or the scaling relationship between the Chapman-Enskog theory, lattice Boltzmann approach and the Nevier-Stokes equation, are all hinged on the interactions between their parts. Therefore, it is not surprising at all that the authors found that their mean-field-approximation deviated significantly from their so-called full model simulations. Further, from existing scaling theories in the literature, another key of success seems to maintain the essential invariants of the system when one transits from one scale to another, yet the Michaelis-Menten kinetics they use is a crude approximation and misses some important invariant that is included in its origin law of mass action (Tang and Riley, 2017), and is deemed to show the difference they found. In addition, there's no guarantee that the mean-field equation will possess the same form as the micro-scale equation. For this, a very good example can be found in geophysical fluids, where at different scales, their governing equations are different, e.g., Gill (Atmosphere-Ocean dynamics, 1982)). Another more relevant example on decomposition is in Wang and Allison (2019).

Third, I feel the authors have some misunderstanding about the mean-field theory and the meaning of well-mixed soil condition. In fact, the scaling problem we are facing here is very similar like the situation hydrologists encountered in upscaling the soil moisture
and soil matric potential relationship in the 1970s-1980s. Using statistical theory, they were able to derive closed analytical relationships (e.g., Mualem, 1976) to inform important soil water retention curve formulations to be derived from empirical data (e.g., van Genuchten, 1980). Therefore, whenever moisture-pressure relationships are included in soil biogeochemical models, some microstructure is included in the so-called mean-field-theory based model (although I should admit that the authors did not consider soil moisture in this study). Or put this straightforwardly, mean field theory does not rule out the inclusion of microstructure, as was demonstrated in the recent upscaling study of substrate affinity parameter (Tang and Riley, 2019), and the study of turbulence (e.g., Takahashi, 2017). In the same vein, a well-mixed soil can also have microstructure, and be properly parameterized. In fact, the latter is what motivated the dual-porosity or the multiple-Rates Mass Transfer models, which have enjoyed many successful applications (e.g., Haggerty and Gorelick, 1995).

Other comments

I agree with Dr. Wutzler that there might be some problems with a few of their equations, e.g. Eqs (A1) and (A2), and the authors should double check their derivations.

Reference

Haggerty, R., and Gorelick, S. M.: Multiple-Rate Mass-Transfer for Modeling Diffusion and Surface-Reactions in Media with Pore-Scale Heterogeneity, Water Resources Research, 31, 2383-2400, Doi 10.1029/95wr10583, 1995. Mualem, Y.: A New Model for Predicting the Hydraulic Conductivity of Unsaturated Porous Media, WRR, 1976 Tang, J. Y., and Riley, W. J.: SUPECA kinetics for scaling redox reactions in networks of mixed substrates and consumers and an example application to aerobic soil respiration, Geoscientific Model Development, 10, 3277-3295, 10.5194/gmd-10-3277-2017, 2017. Tang, J. Y., and Riley, W. J.: A Theory of Effective Microbial Substrate Affinity Parameters in Variably Saturated Soils and an Example Application to Aerobic Soil Heterotrophic Respiration, Journal of Geophysical Research-Biogeosciences,

124, 918-940, 10.1029/2018jg004779, 2019. Takahashi, K.: Mean-field theory of turbulence from the variational principle and its application to the rotation of a thin fluid disk, Prog Theor Exp Phys, ARTN 083J01 10.1093/ptep/ptx109, 2017. Van Genuchten M.T.: A closed-form equation for predicting the hydraulic conductivity of unsaturated soils, SSSAJ, 1980. Wang, B. and Allison S.: Emergent properties of organic matter decomposition by soil enzymes, SBB, 2019.

---

## Author Comment (AC2) · 25 Sep 2019

**Final response to reviewer's comments on "Dynamic upscaling of decomposition**

**kinetics for carbon cycling models"**

We would like to thank the three reviewers for their comments. In this brief summary, we highlight the comments that in our view are most important to address should a revision of our manuscript be encouraged. In general, the reviewers commented favourably regarding the potential interest of the proposed work, but raised concerns about its applicability. We agree with this general concern and had already openly acknowledged the limitations of our approach in the original manuscript. However, we also think that a theoretical approach to link different scales in soil carbon cycling models is missing and this contribution provides a way to start bridging this gap that complements ongoing efforts by other groups.

Reviewer 1: the main concerns regard the interpretation of results (oscillations, convergence to equilibrium, sensitivity to changes in parameter values), and the establishment of a closed-form solution that can be applicable in biogeochemical models. In our response, we provide additional analyses and explanations of the results that can be included in an extended Discussion in the revised manuscript. In particular, we extended our analysis to a fourth type of decomposition kinetics used in soil C cycling models (inverse Michaelis-Menten).

Reviewer 2: the main concerns regard the validation of the proposed approach, its high level of abstraction, and the lack of representation of some physical processes known to determine heterogeneous distributions of soil substrates and microorganisms. In our response, we argue in favour of a theoretical framework, while acknowledging its limitation. A 'standard' model calibration/validation is not possible due to lack of fine-scale data, but the theoretical insights provided by our approach can still be useful. It is correct that some physical processes had not been represented, but our goal is to establish a link between macro- and micro-scale dynamics starting from an idealized system. In a revised manuscript, we would further highlight approach limitations; moreover, also in response to reviewer 3, we can include a simple representation of mass transfer as a proxy for physical transport processes that we had initially neglected.

Reviewer 3: the main concerns regard the applicability of the approach, the assumption of negligible cell-to-cell connectivity, and our interpretation of averaging and mean-field approximations. As explained in the responses above, our approach is still admittedly far from being readily applicable and we acknowledge this limitation in the manuscript. We can, however, improve the model by including mass transfer, thus addressing the second concern (new results are presented in the detailed response). Finally, we clarify our interpretations of the terms 'mean field approximation' and 'well-mixed' conditions, which might have created some ambiguities.

Detailed responses are attached below.

**Response to reviewer 1 (Thomas Wutzler):**

We would like to thank Thomas Wutzler for an encouraging and detailed review of our manuscript. In the following, we address the points raised by Dr. Wutzler (denoted by italic font). Our responses are highlighted in blue font.

**General comments**

1. Methodology description: the way of providing spatial moments to analytic equations did not become clear to me (P17L11). I assume, you computed the quantities for sufficiently close time points from the distributed model, and provided a smoothing function depending on time as input to the solver for the analytic equation system.

There are two ways to illustrate how heterogeneities affect soil organic matter kinetics. One is to solve directly the upscaled equations (assuming the second order terms are known); the other is to numerically solve the micro-scale equations and aggregate (=average) results at the macro-scale. In the first approach, solving the upscaled differential equation would require transferring information on the second order moments (either via closure equations or using moments calculated by the distributed model) to the upscaled differential equation – this issue of 'model closure' is presented in the Discussion. Here we followed the second approach and instead of solving the upscaled differential equations (Eq. (9) and (10), P8L15), we used the averaged dynamics as simulated from the distributed model (explained in the manuscript in P13L4 and P30L22). The average dynamics are then compared to those obtained under homogeneous conditions.

This choice might seem to counter the purpose of this work - i.e., propose an analytical approach to the upscaling problem. However, the lack of model closure (admittedly the main limitation of our approach, as acknowledged in the Discussion) does not allow a full solution of the upscaled analytical model. Thus we used numerical solutions to illustrate the effect of heterogeneities, and the analytical equations to provide a theoretical framework for studying the problem.

It should be pointed out that the numerical averaging approach yields exactly the same solution of the upscaled analytical equations. We tested this for the case of biophysical heterogeneity with multiplicative kinetics because the upscaled equation are exact and only the covariance of substrate and biomass is needed as additional information (the second order moments only involve the covariance). As expected, results were exactly the same from the upscaled differential equation and the distributed model.

In a revised manuscript, we would further emphasize this rationale.

2. An overview of the approach would be helpful: 1) Express each equation of state variable change of each individual location based on the spatial mean of the pool sizes and the deviations from it at local scale. And 2) Apply a spatial averaging over the obtained equations, resulting in an equation composed of terms of the mean pool sizes and the spatial covariance of the pools and heterogeneously distributed parameters.

This is indeed our approach. We start with a micro-scale model i.e. Eq. (2) and (3) and apply spatial averaging to obtain the upscaled (or macro-scale) equation i.e. Eq. (9) and (10). However, instead of writing the averaged rates  $(\overline{D})$  explicitly in Eq. (9) and (10), we derive  $\overline{D}$  for different micro-scale kinetics. The upscaled expression of  $\overline{D}$  are shown in Table 2 of manuscript. We will provide a better roadmap as suggested at the beginning of the theory section.

3. I tried to check the math, but did not always come to the same results (see detailed comments, eq. A7)

There was a typing mistake. The incorrect term in Eq. A7 should read  $-\frac{2 k_{s,mm} K_M \bar{C}_B}{(K_M + \bar{C}_S)^3}$ . Numerical code and results are correct.

In the discussion I would like to read about several points:

4. Slowdown of decomposition: To my opinion the slowdown of decomposition despite plenty of available substrate (Fig 7d) is a very important feature/insight of the model. A very simple model (albeit still required input of time series of heterogeneity variances) can explain why we can find very old potentially quickly decomposable SOM. The reasons should be explained in more detail (right skewed distribution of decomposition rate, low probability of co-occurrence of high substrate concentration, ...)

A paragraph at the end of section 4.1 Predicted effects of spatial heterogeneity on decomposition will be added to address this comment:

"Our analysis suggests that the persistence of SOM in heterogeneous systems is a consequence of the microscale heterogeneity in soil carbon cycling. In the transient simulations with biophysical heterogeneity, persistence is a result of spatial disconnection between substrate and microorganism, captured in our framework by a low probability of co-location at the beginning of the simulation. In the transient simulations for the fully heterogeneous systems, persistence is a result of the combined effects of low probability of co-location and high probability of low decomposition rate constant at the beginning of the simulation. The heterogeneity in substrate quality explains the higher persistence of SOM in the fully heterogeneous system compared to the biophysically heterogeneous system."

5. Oscillations at multi-annual time scale: Observations of such a phenomenon are very rare. I once argued that we do not see such modelled oscillations with microbial explicit models because of superposition of dynamics across many pores. Here, such spatial heterogeneity is the cause of fluctuations.

In short, yes, we find that heterogeneous initial placement of substrates can trigger fluctuations around the steady state, and in the case of chemical heterogeneity can actually alter the steady state. Any perturbation from the steady state would lead to fluctuations, and not only a re-arrangement of substrates and microbes in space. When decomposition rate is a nonlinear function of substrate concentration in simple C cycling models like the one used here oscillations tend to appear in a wide region of the parameter space (Manzoni and Porporato, 2007; Sierra and Mueller, 2015). We elaborate on this in our reply to comment 8 below.

6. Development of the heterogeneity / damping of oscillations (Fig 4): The systems develops to a steady state without any more oscillations. Is the initial heterogeneity developing in direction of homogeneity? Probably not because the simulated SOM stocks differ from the homogeneous system. What is the spatial distribution and covariance between substrate, biomass, and quality after 60 years? Is there a covariance pattern that is stable? I suggest putting another two panels to Fig. 2 showing microbial and substrate distribution at year 60.

Yes, in the case of biophysical heterogeneity, the covariance and the sum of the higher order terms are stable in the long term, after the steady state has been reached. Also in the case of full heterogeneity these moments stabilize, but now the steady state is different. We ran the simulation for 100 years and a figure similar to our original Figure 4 is shown below, which confirm that higher order terms are indeed stable.

Because C input is homogeneously distributed, the microbial and substrate concentration spatial distributions are also homogeneous at steady state. However, these compartments attained a different concentration value at steady state compared to the homogeneous system (shown in Figure 4). Adding panels showing a spatially homogeneous distribution might not add much information, but we can better explain this pattern in a revised manuscript.

Figure R1: Substrate and microbial biomass C (top rows), respiration and higher order terms (bottom rows) as a function of time, for different scenarios of initial spatial heterogeneity. This figure is the same as Figure 4 in the manuscript but for 100 years of simulation period, aiming to highlight the long-term behaviors of C pools and fluxes.

7. Role of disturbances: What happens if you simulate a disturbance (homogenization) after the system is near steady state? Does this start the oscillating pattern again?

Homogenizing the system when it reaches the steady state will not change the spatial distribution of substrates and microbes because with spatially homogeneous inputs at steady state they are homogeneously distributed. Thus, oscillations would not start again after the system has reached a steady state, regardless of whether the system has only biophysical heterogeneities or is fully heterogeneous. 8. Magnitude of the heterogeneity effects: In Figure 4, the effects look large, because the axis ranges from 5 to 7, but aside from the initial disturbance, the effect is only about 1/10 of the steady state. Are there reasonable parameter combinations where the effect is larger? Or do we not need to are this much about heterogeneity at steady state?

To answer, we ran two scenarios in which we changed the kinetic constant parameter  $k_{s,mult}$ : 1) decreasing  $k_{s,mult}$  in the biophysical heterogeneity (Figure R2 and Figure R3) and 2) increasing the heterogeneity of  $k_{s,mult}$  (by increasing its standard deviation) in the full heterogeneity case (Figure R4). From Fig. R3 and R4, it is clear that decreasing the rate constant increases the amplitude and wavelength of the oscillations. As shown in Figure R4, increasing the heterogeneity of the rate constant (right column) increases the amount of undecomposed substrate C compared to a lower degree of heterogeneity (middle column). This pattern can be explained using the analytical expression of the steady state substrate C (see Eq. (A13) in Appendix A2, P36L5). For the increased heterogeneity case shown in the right column, we used values of a = -10.1 and b = -8.56, where *a* and *b* have the same meaning as in Eq. (A13). The analytical expression for the steady state, evaluated with these values of *a* and *b*, results in exactly the same steady state of substrate C as simulated by the distributed model (i.e. 15 mgC/gSoil).

These fluctuations are similar to those noted in earlier papers using spatially lumped models (Manzoni and Porporato, 2007; Sierra and Mueller, 2015). These papers showed that the occurrence and amplitude of the fluctuations depends on the kinetic parameter values, as is the case here.

Figure R2: (a) mean substrate C ( $\overline{C}_S$ ), (b) mean microbial C ( $\overline{C}_B$ ), (c) mean respiration rate ( $\overline{R}$ ), and (d) sum of second and third order terms ( $\Sigma$ HOT) are shown as a function of time, for different scenarios of initial spatial heterogeneity. This figure is similar to Figure 4 left column in the manuscript (initial substrate is distributed randomly around the steady state). Different line colors represent varying levels of rate constant  $k_{s,mult}$  with base case same as in Figure 4 of manuscript.

---

## Author Comment (AC3) · 25 Sep 2019

**Final response to reviewer's comments on "Dynamic upscaling of decomposition kinetics for carbon cycling models"**

We would like to thank the three reviewers for their comments. In this brief summary, we highlight the comments that in our view are most important to address should a revision of our manuscript be encouraged. In general, the reviewers commented favourably regarding the potential interest of the proposed work, but raised concerns about its applicability. We agree with this general concern and had already openly acknowledged the limitations of our approach in the original manuscript. However, we also think that a theoretical approach to link different scales in soil carbon cycling models is missing and this contribution provides a way to start bridging this gap that complements ongoing efforts by other groups.

Reviewer 1: the main concerns regard the interpretation of results (oscillations, convergence to equilibrium, sensitivity to changes in parameter values), and the establishment of a closed-form solution that can be applicable in biogeochemical models. In our response, we provide additional analyses and explanations of the results that can be included in an extended Discussion in the revised manuscript. In particular, we extended our analysis to a fourth type of decomposition kinetics used in soil C cycling models (inverse Michaelis-Menten).

Reviewer 2: the main concerns regard the validation of the proposed approach, its high level of abstraction, and the lack of representation of some physical processes known to determine heterogeneous distributions of soil substrates and microorganisms. In our response, we argue in favour of a theoretical framework, while acknowledging its limitation. A 'standard' model calibration/validation is not possible due to lack of fine-scale data, but the theoretical insights provided by our approach can still be useful. It is correct that some physical processes had not been represented, but our goal is to establish a link between macro- and micro-scale dynamics starting from an idealized system. In a revised manuscript, we would further highlight approach limitations; moreover, also in response to reviewer 3, we can include a simple representation of mass transfer as a proxy for physical transport processes that we had initially neglected.

Reviewer 3: the main concerns regard the applicability of the approach, the assumption of negligible cell-to-cell connectivity, and our interpretation of averaging and mean-field approximations. As explained in the responses above, our approach is still admittedly far from being readily applicable and we acknowledge this limitation in the manuscript. We can, however, improve the model by including mass transfer, thus addressing the second concern (new results are presented in the detailed response). Finally, we clarify our interpretations of the terms 'mean field approximation' and 'well-mixed' conditions, which might have created some ambiguities.

Detailed responses are attached below.

**Response to reviewer 2 (Ali Ebrahimi):**

We would like to thank Ali Ebrahimi for the review of our manuscript. Our responses are highlighted in blue font. Some of the comments point to the approach limitations – indeed, this is a theoretical study the findings of which are hard to validate because there is no data at a fine-enough resolution. We would like to emphasize that the value of the proposed approach is to provide a framework for studying heterogeneity effects on measurable C fluxes and stimulate discussion in this area.

Major concerns:

1. *It is unclear to what extend the parameterization proposed in the analytical kinetic model could be experimentally validated. My major problem is that some of the quantities do not have real biogeochemical or physical meanings in which could be experimentally measured. For instance there is an emphasis on the second*

*order moments as state variables used to close the model; however it is hard to think how such variable could be experimentally measured.*

It is correctly stated by the reviewer that the second order moments (SOTs) do not represent 'real' biogeochemical fluxes, but they do have a clear physical meaning – SOTs represent spatial variability and co-variation of the state variables. SOTs should be considered as corrections to the mean-field approximation of the micro-scale model. State transition theory states that if the macro-scale system is heterogeneous, then rates calculated using the mean state variables are not accurate, i.e. the mean-field approximation does not offer a good representation of the dynamics. These SOT can be estimated from the spatial moments of the state variables, as done in applications of state transition theory in population ecology (e.g., Englund and Leonardsson, 2008). The problem is that these measurements in soil environments are difficult, so we hope that this theoretical study (as others being proposed lately) will stimulate advances in empirical approaches to fill this gap.

Experimental validation should be thought in the following sense: observations obtained from SOC decomposition represent the averaged response of the system, and this averaged response is expected to differ between a truly homogeneous system and a heterogeneous one. Therefore, experimental validation should stem from designing a truly homogeneous soil system and comparing it with regular soil based experiments, which are expected to be influenced by spatial heterogeneities in the sample. If any difference is observed between the homogeneous and heterogeneous systems, then our framework suggests that this difference should be attributed to spatial variability at the micro-scale (second order terms).

Furthermore, in this contribution we do not propose a parametrization of the upscaled equations – the so-called 'closure problem'. We simply compare the effect of micro-scale heterogeneities on the averaged C dynamics. While the current formulations based upon scale transition theory provide a way for conceptually including the effect of micro-scale heterogeneities in upscaled equations, finding a parametrization for SOTs remains a challenge. This challenge was presented in the original manuscript (P29L27, P30L22-26, P31L19), but we will expand on this in a revised version.

2. *The type of model and scenarios proposed in this study are relevant and could potentially address some of the inconsistency in our field measurements but it could only be possible if the model could establish a systematic link to relevant abiotic and biotic factors observed in the field. While in the discussion authors have tried to relate some of the scenarios in the study to soil aggregation or pore connectivity and an entire subsection is dedicated for that, I still find that the **modeling framework is too abstract that makes the explanations quite speculative** and hard to think to what extend the decomposition rate may vary under realistic settings.*

The modeling framework described in the manuscript is motivated by its simplicity to describe C dynamics at the micro-scale and a tractable number of SOTs that results from scale transition theory. This simplicity (and high level 'abstraction' to quote the reviewer) allow for theoretical insights on how spatial heterogeneities affect the macro-scale fluxes. Several experiments are now exploring how substrate placement affect respiration (e.g.,

Don et al., 2013; Schnecker et al., 2019) – these results point to spatial placement as a key driver of C fluxes. While the theory presented here can be used as a framework for interpreting those results, as explained in the Discussion a direct theory validation is not possible at this stage. Other approaches to soil heterogeneities have focused on spatial gradients in aggregates (as in recent papers by the reviewer); here we propose a more general – albeit harder to validate – approach based on a statistical description of substrate and microbial placement in the soil.

While we already acknowledge the theoretical nature of this study and its limitation in the original manuscript (which the reviewer correctly points out in his comment), it should be noted that statistical approaches as the one proposed have not been applied to soil systems and thus might offer novel solutions to the limitations of current microbial-explicit models. Here we do not claim that we fully address these limitations, but we hope that this work can contribute to the discussion in this area.

3. *The model could potentially describe some of the underlying abiotic and physio-logical mechanisms that shape the decomposition dynamics but in the current form of the manuscript this has not been explored. For instance, I was wondering to what extend half saturation to substrate and decomposition rate constant (KM and Ks) are shaping the dynamics observed in the model. I would guess if lower KM or high Ks would have been chosen the heterogeneous scenarios would have converged faster to the homogenous one.*

Indeed the values of the kinetic parameters affect the speed of convergence to the steady state, the actual value of the steady state (this is a novel result as well), and the nature of the fluctuations towards the steady state. We explored how varying the decomposition rate constant and its variability on the macro-scale respiration rate in a set of figures prepared in response to a related comment by reviewer 1 (Figures R2, R3, and R4).

Minor concerns:

4. *I was wondering to what extend the fluctuating environmental condition (for instance fluctuating in carbon distributions) could play a role in shaping the carbon decomposition dynamics. Do you expect to see faster convergence to homogenous scenario in high intensity fluctuations?*

We hope to interpret correctly the meaning of the term 'high-intensity fluctuations' in this comment, as conditions of higher contrast between substrate and microbial biomass concentrations. If our interpretation is correct, this question is best answered by comparing the convergence to homogeneous conditions of systems with different degrees of correlation between substrates and microbial biomass. With negative correlation (highest contrast), the system converges more or less at the same speed as in the case of positive correlation. Convergence is slower when there is no correlation between substrate and microbes (Figure 4).

5. *The system size that is modeled in grid based network is rather small. The number of grids, or pores equal to 10000, is basically enough to model an aggregate with the size of 0.5mm. I was wondering if this size is sufficiently large to capture heterogeneities in the soil?* ***For instance inter aggregate pores or macro-pores****?*

For system with no spatial interaction, 10000 grid points are enough for statistically meaningful mean dynamics and dynamics of second order term do not change by increasing the number of grid points. We have not explored the role of variability among aggregates or of a pore structure – these are all good suggestions for future work, but are beyond our scope here.

6. *Following up on the results for negative correlations**, I was wondering how much physical inaccessibility of the carbon to microbes could be relevant for the soil systems?** For instance most of the carbon protections in soil are often driven by soil aggregation and creation of anoxic microsites. In a broader term, the counter gradients created by carbon and other necessary substrates for carbon degradation could lead to inaccessibility of the carbon for microbes and not necessary physical inaccessibility. This is a phenomenon that has been previously shown in soil aggregates that due to creation of anoxic zones, the carbon configuration does not play a role in carbon consumption (Ebrahimi and Or, 2018 GCB) and in other studies showing carbon protection by aggregation (e.g., Keiluweit et al., 2017 Nat. comm.).*

This is also an interesting point. Indeed, inaccessibility of substrate to microorganism can be caused by several bio-physio-chemical processes. Inaccessibility caused by any biophysical factor should decrease the mean respiration rate because inaccessible C is not contributing to the $CO_2$ emission. In the spatially explicit mass transfer based model (described in the response to reviewer 3), we explored the effect of different degrees of spatial redistribution of the decomposition products. This analysis shows that increasing the level of spatial redistribution would result in decomposition of all the substrate in the domain even if substrate and microorganism are initially isolated i.e. negative initial correlation. Therefore, it is the combined effect of initial inaccessibility and spatial redistribution mechanisms that is responsible for making substrates available to microbes.

Returning to the question of the reviewer: with the current model we cannot provide a clear answer as in some systems time-varying anoxic conditions will drive the spatial distribution of microbial activity (which we did not account for here), while in other systems physical protection or heterogeneous substrate 'quality' may matter more (which we did consider).

**References**

Don, A., Rödenbeck, C. and Gleixner, G.: Unexpected control of soil carbon turnover by soil carbon concentration, Environ Chem Lett, 11(4), 407–413, doi:10.1007/s10311-013-0433-3, 2013.

Englund, G., and Leonardsson, K.: Scaling up the functional response for spatially heterogeneous systems, Ecology Letters, 11, 440-449, 10.1111/j.1461-0248.2008.01159.x, 2008.

Schnecker, J., Bowles, T., Hobbie, E.A. et al. Biogeochemistry (2019) 144: 47. https://doi.org/10.1007/s10533-019-00571-8

---

## Author Comment (AC4) · 25 Sep 2019

**Final response to reviewer's comments on "Dynamic upscaling of decomposition kinetics for carbon cycling models"**

We would like to thank the three reviewers for their comments. In this brief summary, we highlight the comments that in our view are most important to address should a revision of our manuscript be encouraged. In general, the reviewers commented favourably regarding the potential interest of the proposed work, but raised concerns about its applicability. We agree with this general concern and had already openly acknowledged the limitations of our approach in the original manuscript. However, we also think that a theoretical approach to link different scales in soil carbon cycling models is missing and this contribution provides a way to start bridging this gap that complements ongoing efforts by other groups.

Reviewer 1: the main concerns regard the interpretation of results (oscillations, convergence to equilibrium, sensitivity to changes in parameter values), and the establishment of a closed-form solution that can be applicable in biogeochemical models. In our response, we provide additional analyses and explanations of the results that can be included in an extended Discussion in the revised manuscript. In particular, we extended our analysis to a fourth type of decomposition kinetics used in soil C cycling models (inverse Michaelis-Menten).

Reviewer 2: the main concerns regard the validation of the proposed approach, its high level of abstraction, and the lack of representation of some physical processes known to determine heterogeneous distributions of soil substrates and microorganisms. In our response, we argue in favour of a theoretical framework, while acknowledging its limitation. A 'standard' model calibration/validation is not possible due to lack of fine-scale data, but the theoretical insights provided by our approach can still be useful. It is correct that some physical processes had not been represented, but our goal is to establish a link between macro- and micro-scale dynamics starting from an idealized system. In a revised manuscript, we would further highlight approach limitations; moreover, also in response to reviewer 3, we can include a simple representation of mass transfer as a proxy for physical transport processes that we had initially neglected.

Reviewer 3: the main concerns regard the applicability of the approach, the assumption of negligible cell-to-cell connectivity, and our interpretation of averaging and mean-field approximations. As explained in the responses above, our approach is still admittedly far from being readily applicable and we acknowledge this limitation in the manuscript. We can, however, improve the model by including mass transfer, thus addressing the second concern (new results are presented in the detailed response). Finally, we clarify our interpretations of the terms 'mean field approximation' and 'well-mixed' conditions, which might have created some ambiguities.

Detailed responses are attached below.

**Response to reviewer 3**

We would like to thank reviewer 3 for the review of our manuscript. Our responses are highlighted in blue font.

1. *While the conclusions they drew are solid within their model configuration, I too share with others the concern that how this learned lesson could be translated into something universally applicable for other modelers. In particular, we in the soil biogeochemical modeling community have so far no unanimously accepted governing equation to solve like that exist for geophysical fluid dynamics, or hydrodynamics in general, where re-solving the microstructure effects can be achieved through the so called large-eddy simulation and sub-grid closure, and even field or laboratory experiments can be designed to derive parameterization schemes that are generally*

*applicable for different situations. Personally, I am therefore wondering can the authors' approach become some tools that are easily accessible to others, e.g., like Markov chain Monte Carlo codes that are widely accessible through open source software?*

Our methodology could be considered as a conceptual tool to understand the link of governing equations at two different scales. As mentioned by the reviewer, the lack of universally accepted C dynamics equations at different scales is one of the key issues in soil science. Use of scale transition theory is advantageous because it does not assume any predefined form of macro-scale equation, and depending upon the structure of the micro-scale model, the upscaling procedure results in macro-scale terms that define the variation across scales as a result of micro-scale properties (i.e. heterogeneities). The formulation of the micro-scale model is also uncertain, but we can test different variants of decomposition kinetics to test how the nonlinearities at the micro-scale translate into more or less nonlinear behavior at the large scales. In a revision, we would include an addition kinetic model to further expand the analysis, as described in our response to reviewer 1 and below.

The terms in the macro-scale equations that depend on micro-scale features are mathematically expressed by the second order spatial moments. Based on which second order moments appear in the upscaled equation and the kind of heterogeneity present in the system, one could start thinking about the relative importance of each term. This issue is partially covered in our answer to the comment 10 of reviewer 1 where we describe a possible strategy to obtain a second order approximation of upscaled decomposition rate. Such a second order approximation could be useful in operational models. In this contribution, however, we aim at providing a framework for upscaling that is complementary to other recent approaches, among which those referenced below (e.g., Tang and Riley, 2017).

2. *Second I am a little bit disappointed that authors decided to ignore the interactions between different micro-grids. In physics, the successful upscaling is achieved only through the consideration of interactions. For instance, the scaling of Newton's law of momentum conservation, the derivation of center of gravity, or the scaling relation-ship between the Chapman-Enskog theory, lattice Boltzmann approach and the Naiver-Stokes equation, are all hinged on the interactions between their parts. Therefore, it is not surprising at all that the authors found that their mean-field-approximation deviated significantly from their so-called full model simulations. Further, from existing scaling theories in the literature, another key of success seems to maintain the essential invariants of the system when one transits from one scale to another, yet the Michaelis-Menten kinetics they use is a crude approximation and misses some important invariant that is included in its origin law of mass action (Tang and Riley, 2017), and is deemed to show the difference they found. In addition, there's no guarantee that the mean-field equation will possess the same form as the micro-scale equation. For this, a very good example can be found in geophysical fluids, where at different scales, their governing equations are different, e.g., Gill (Atmosphere-Ocean dynamics, 1982)). Another more relevant example on decomposition is in Wang and Allison (2019).*

This comment includes two separate questions: the first is related to spatial redistribution and the second is regarding the choice of micro-scale kinetics. To address the first comment regarding the importance of spatial

interactions (or redistribution), we implemented two new versions of our model that include a redistribution mechanism – the first is based on diffusion and second is based on a generic mass transfer. The idea is to understand to what extent these spatial fluxes are important in the decomposition of soil organic carbon and their effect on the upscaling procedure we propose.

**Diffusion-based three pool model**

To demonstrate the effect of diffusive fluxes, we need to have an additional pool of C that is mobile, i.e. DOC. So the micro-scale model structure changes as follows,

$$\frac{dSOC}{dt} = a\,k_B\,B - V_m B\,\frac{SOC}{K_m + SOC}$$

$$\frac{\partial DOC}{\partial t} = (1-a)\,k_B\,B - V_u B\,\frac{DOC}{K_{DOC} + DOC} + V_m B\,\frac{SOC}{K_m + SOC} + D_{DOC}\left(\frac{\partial^2 DOC}{\partial x^2} + \frac{\partial^2 DOC}{\partial y^2}\right)$$

$$\frac{\partial B}{\partial t} = Y V_u B\,\frac{DOC}{K_{DOC} + DOC} - k_B\,B + D_B\left(\frac{\partial^2 B}{\partial x^2} + \frac{\partial^2 B}{\partial y^2}\right)$$

$$R = \frac{\partial CO_2}{\partial t} = (1-Y)\,\frac{K_{smm}\,DOC}{K_{DOC} + DOC}\,B$$

where $SOC$ is soil organic carbon and $B$ is microbial C (shorter notation, but conceptually the same as $C_s$ and $C_b$ in the manuscript). $DOC$ is dissolved organic carbon, an additional carbon pool that is prone to transport via advection (not considered here) and diffusion. Also biomass is assumed to be transported according to a gradient-driven process with a given diffusivity. $V_m$ and $K_m$ are the maximum decomposition rate constant and the half saturation constant for SOC and similarly $V_u$ and $K_{DOC}$ are the maximum uptake rate and the half saturation constant for DOC uptake. $D_{DOC} = 2.6\,E-12\,\frac{m^2}{s}$ and $D_B = 0.1 D_{DOC}$ are the DOC and biomass diffusivities. The coefficient $a$ partitions microbial cell turnover between SOC and DOC.

Diffusion at pore scale is affected by the pore geometry and at the Darcy scale it is approximated as a function of soil moisture in unsaturated conditions. Here, we used the bulk diffusivity to simulate diffusion of DOC, and assumed that the diffusivity for biomass is 10 times smaller than that of DOC. Experiments studying the DOC pool suggest that DOC is a small fraction of soil organic matter and is quickly taken up by microbes (Schnecker et al., 2019). Therefore, it is reasonable to expect that the DOC pool will be at quasi-steady and the DOC three-pool model would be well-approximated by the two-pool model (same as in the manuscript). We obtained the parameters of the three pool model by changing the kinetics of DOC production and uptake so that the dynamics of SOC and B are comparable in both models. Figure R7 shows the time evolution of SOC, DOC and the microbial C pool in a heterogeneous system, along with the dynamics of the same variables obtained in a homogeneous system based on three-pool and two-pool models.

[Figure]

**Figure R7: Temporal evolution of the mean SOC, DOC and microbial C pools when C substrates and biomass are initially negatively (left panel), positively (center) and not correlated (right). Note that the right y-axes and the black and red curves refer to DOC in all panels.**

[Figure]

**Figure R8: Mean respiration rate when C substrates and biomass are initially negatively, positively and not correlated in two- and three-pool models.**

First, this analysis shows that the two- and three-pool models behave similarly when the system is homogeneous, which is expected because we constrained DOC to be turning over quickly (as suggested by empirical evidence).

Second, the three-pool model responds to the initial heterogeneous distribution of substrate and biomass showing a delayed respiration peak when the substrates are initially separated from the microbial cells (negative correlation). This result is consistent with that shown in Figure 6 of the manuscript. However, when no correlation occurs between substrates and microbes, an even longer delay emerges, in contrast to our original results in Figure 6. This suggests that diffusion does not alter the qualitative responses to micro-scale heterogeneity if microbes are in close vicinity or separated from the substrate, but could play a role in the intermediate case of no correlation. This difference between the model including diffusion and the original version without diffusion is caused by the specific initial placement of substrates and microbes. In the original

model, the initial placement did not matter, but only its statistical properties determined the dynamics (i.e., the sample size was sufficiently large). In contrast, when diffusion is included, a larger sample size becomes necessary to ensure that a specific initial configuration does not influence the dynamics. The need for a larger sample size is due to the occurrence of spatial interactions that create clusters spanning a large fraction of the domain and whose dynamics control the dynamics of the mean values of state variables and fluxes.

**Mass transfer-based spatially explicit model**

To explicitly include spatial fluxes across grid cells without changing the structure of the two-pool model used in the original manuscript, we implemented an alternative mass transfer mechanism. In this model, SOC is decomposed at rate $D$ from which $\alpha D$ is transferred in equal amounts to the four neighboring grid cells. Hence, in each cell microbes take up C from neighboring cells at a rate $\frac{\alpha}{4}\left(D_{i-1,j} + D_{i+1,j} + D_{i,j-1} + D_{i,j+1}\right)$. This choice is motivated by the observation that the products of de-polymerization are more soluble than stable organic matter and thus are more likely to be transported away from the site of decomposition. So instead of modelling DOC explicitly, we assumed that a fraction of the C flux that represents the source of soluble C is transported in other cells. This mass transfer mechanism can be interpreted as a consequence of any type of redistribution process in soils, including diffusion, dispersion or bioturbation.

[Figure]

If $\alpha$ is zero, no mass transfer occurs and the model becomes equivalent to our original two pool model. If $\alpha$ is greater than zero, then there is mass transfer among the grid cells. In this way, by changing the value of $\alpha$, we can study the effect of spatial mass transfer flux on mean carbon dynamics. Micro-scale equations at one grid cell (control volume) take the following form,

$$\frac{dCs_{i,j}}{dt} = I - D_{i,j} + T_{i,j}$$

$$\frac{dB_{i,j}}{dt} = Y\left((1-\alpha)D_{i,j} + \frac{\alpha}{4}\left(D_{i-1,j} + D_{i+1,j} + D_{i,j-1} + D_{i,j+1}\right)\right) - T_{i,j}$$

$$\frac{dCO2_{i,j}}{dt} = (1-Y)\left((1-\alpha)D_{i,j} + \frac{\alpha}{4}\left(D_{i-1,j} + D_{i+1,j} + D_{i,j-1} + D_{i,j+1}\right)\right)$$

We used this form of micro-scale model to simulate the effect of $\alpha$ on the averaged dynamics of decomposition for positive, negative and uncorrelated system in the same way as it was done in the original manuscript. In Fig. R9, we show the specific growth rate as a function of substrate for an uncorrelated initial distribution of substrates and microbes, and for all three kinetics– multiplicative (mult), Michaelis-Menten (MM) and inverse Michaelis-Menten (inv. MM). When $\alpha = 0$, result in Fig. R9 are same as in Fig. R6 for the uncorrelated case. When $\alpha > 0$, microbes that were initially deprived of substrate can receive it from neighboring grid cells. As a consequence of improved accessibility, given enough time microbes would consume all the substrate, whereas without mass transfer some C remains undecomposed. In other words, the persistence of substrate C we have highlighted in the manuscript in the fully heterogeneous system is lost when mass transfer provides food for microbial growth in all cells of the domain (provided enough time for transport to occur).

An interesting result emerges from Fig. R9: the inverse MM kinetics captures the effect of heterogeneity better than MM or mult, i.e. the shape of the kinetics is relatively similar to homogeneous conditions. This result might reflect some degree of scale-invariance of inverse MM kinetics.

We plan to implement this modified model in a revised manuscript, including the new figures presented here and the corresponding discussion.

[Figure]

**Figure R9: Effect of biophysical (left column) and full (right column) heterogeneity on the mean specific growth rate (SGR) as a function of mean substrate C ($\overline{C}_S$) for an uncorrelated initial distribution of substrates and microbes. The tree horizontal panels are for (a,b) multiplicative, (c,d) Michaelis-Menten and (e,f) Inverse Michaelis-Menten kinetics. Different colors represent varying values of $\alpha$. Time progresses from right to left, as substrate C is depleted.**

3. *Third, I feel the authors have some misunderstanding about the mean-field theory and the meaning of well-mixed soil condition. In fact, the scaling problem we are facing here is very similar like the situation hydrologists encountered in upscaling the soil moisture and soil matric potential relationship in the 1970s-1980s. Using statistical theory, they were able to derive closed analytical relationships (e.g., Mualem, 1976) to inform important soil water retention curve formulations to be derived from empirical data (e.g.,van Genuchten, 1980). Therefore, whenever moisture-pressure relationships are included in soil biogeochemical models, some microstructure is included in the so-called mean-field-theory based model (although I should admit that the authors did not consider soil moisture in this study). **Or put this straightforwardly, mean field theory does not rule out the inclusion of microstructure, as was demonstrated in the recent up-scaling study of substrate affinity parameter (Tang and Riley, 2019), and the study of turbulence (e.g., Takahashi, 2017).** In the same vein, a well-mixed soil can also have microstructure, and be properly parameterized. In fact, the latter is what motivated the dual-porosity or the multiple-Rates Mass Transfer models, which have enjoyed many successful applications (e.g., Haggerty and Gorelick, 1995).*

We do not argue against macro-scale models that account for micro-scale heterogeneities in a lumped way – in fact, we hope that scale transition theory can help moving in that direction (complementing other approaches as in the cited paper by Tang and Riley). However, admittedly we are not quite to the point of proposing a closure approach that would provide closed-form equations such as the cited water retention curves.

By "mean field approximation", we indicate the C fluxes calculated with the mean values of the state variables (e.g., as in Melbourne and Chesson, 2006), and not a mean-field theory that accounts for micro-scale processes. By "well-mixed" conditions, we refer to conditions where diffusion is faster than reaction, providing spatially uniform concentrations. This is different from assuming that e.g., random pores are 'well-mixed' in a soil – an assumption used to derive the cited water retention curves. We can clarify the use of these terms in a revised version to avoid any confusion.

In Tang and Riley papers, heterogeneity is introduced by the reaction network adopted for the decomposition of substrate; in other words one or more substrates can be decomposed via one or more enzymes. The question of interest in that case was: what would be the form of the decomposition function in order to account for the reaction network complexities? In this sense, we agree that a well-mixed system can have different types of substrates and enzymes, and one could apply the method developed by tang and Riley to calculate the overall decomposition function. The same philosophy was adopted by Michaelis and Menten to derive their approximated enzymatic reaction equation.

However, this question is different from what we ask here. Our contribution investigates what would happen to the decomposition function if substrate and microbes (or enzymes) are physically co-located or isolated. Spatial heterogeneity is the subject of investigation rather than the heterogeneity of the reaction network in a multi-substrate and multi enzyme system. For our purposes, the assumption of Michaelis-Menten kinetics serves the purpose. In case of chemical heterogeneity where we actually define different substrate qualities by having spatially varying kinetics parameters, the substrate decomposition in a grid cell (control volume) is performed according to the kinetic parameters associated with that cell. In other words, there is no transport of microbes (or enzymes) across grid cells that would create a multi-species system. If there was transport of chemically-different compounds, kinetic laws accounting for complex reaction networks should be used (e.g., SUPECA kinetics in Tang and Riley, 2017). Even for the spatially explicit model based on mass transfer presented earlier, simple forward or inverse MM kinetics would be enough because only a fraction of decomposition flux is transferred across the grid cells and we can assume that the compounds being transported are similar, not requiring more sophisticated kinetic laws.

While we did not implement SUPECA or other complex kinetics (note that we do not model enzymes explicitly), in a revision we can include inverse Michaelis-Menten kinetics in addition to the other kinetics. This would allow exploring the behavior of another type of micro-scale decomposition model. We are also motivated to analyze MM and inv. MM kinetics because they are commonly used and in our view modelers tend to underestimate possible scaling issues with these nonlinear functions.

Preliminary results obtained with the inv. MM kinetics are shown above.

**References**

Melbourne, B. A., and Chesson, P.: The scale transition: Scaling up population dynamics with field data, Ecology, 87, 1478-1488, 10.1890/0012-9658(2006)87[1478:tstsup]2.0.co;2, 2006.

Schnecker, J., Bowles, T., Hobbie, E.A. et al. Biogeochemistry (2019) 144: 47. https://doi.org/10.1007/s10533-019-00571-8

Tang, J. Y., and Riley, W. J.: SUPECA kinetics for scaling redox reactions in networks of mixed substrates and consumers and an example application to aerobic soil respiration, Geoscientific Model Development, 10, 3277-3295, 10.5194/gmd-10-3277-2017, 2017.

---

## Author Response (AR1)

**Revised manuscript "Dynamic upscaling of decomposition kinetics for carbon cycling models"**

We have previously uploaded the point-by-point response to the reviewer comments as author comments AC1-4. This document lists the modifications made in the revised manuscript to reflect reviewers' comments and our responses. Our arguments in support to the responses and additional analyses that are not included in the revised manuscript (but that are relevant to address the comments) are found in the author comments AC1-4.

Part of mathematical text in the marked-up manuscript did not render well because of latexdiff, for clarity of these texts please refer to the revised manuscript. Page and line number references correspond to the revised manuscript.

**Reviewer 1 comments (Thomas Wutzler):**

We would like to thank Thomas Wutzler for an encouraging and detailed review of our manuscript. In the following, we list the changes (highlighted in blue normal font) made in the revised manuscript in response to Dr. Wutzler (denoted by italic font).

General comments

1.  *Methodology description: the way of providing spatial moments to analytic equations did not become clear to me (P17L11). I assume, you computed the quantities for sufficiently close time points from the distributed model, and provided a smoothing function depending on time as input to the solver for the analytic equation system.*

    Please see our response in the interactive discussion.

2.  *An overview of the approach would be helpful: 1) Express each equation of state variable change of each individual location based on the spatial mean of the pool sizes and the deviations from it at local scale. And 2) Apply a spatial averaging over the obtained equations, resulting in an equation composed of terms of the mean pool sizes and the spatial covariance of the pools and heterogeneously distributed parameters.*

    The schematic Figure 1 has been improved to provide a better graphical overview of the approach, and at the beginning of the theory section the rationale was outlined in clearer terms as suggested (see, P5 L18-23):

    "To obtain the macro-scale dynamics we employ two approaches: (i) a numerical approach based on grid-scale simulations followed by spatial averaging (upper panel Fig. 1) and (ii) an analytical approach based on scale transition theory (lower panel Fig. 1). The first, computationally demanding approach requires solving the micro-scale equations at each cell of the domain grid. Spatial averages and variances are thus calculated numerically over the domain at each time point in the simulation. With the analytical approach, the dynamic equations are first averaged and then solved directly for the mean state variables. The obtained analytical expressions are used to interpret the results of the numerical simulations."

3.  *I tried to check the math, but did not always come to the same results (see detailed comments, eq. A7)*

    This was a typing mistake. To address this comment, the expression has been corrected in Table 2 (P14) and Eq. A7 (P36).

The corrected expression is $-\frac{2\,k_{s,mm}K_M\,\bar{C}_B}{(K_M+\bar{C}_S)^3}$.

*In the discussion I would like to read about several points:*

4. *Slowdown of decomposition: To my opinion the slowdown of decomposition despite plenty of available substrate (Fig 7d) is a very important feature/insight of the model. A very simple model (albeit still required input of time series of heterogeneity variances) can explain why we can find very old potentially quickly decomposable SOM. The reasons should be explained in more detail (right skewed distribution of decomposition rate, low probability of co-occurrence of high substrate concentration, …)*

   To address this comment, a paragraph at the end of section 4.1: 'Predicted effects of spatial heterogeneity on decomposition' has been added at P29L16-22,

   "Our analysis suggests that the persistence of SOM in heterogeneous systems may be a consequence of the micro-scale heterogeneity in soil carbon cycling. In the transient simulations with biophysical heterogeneity, persistence is a result of spatial disconnection between substrate and microorganism, captured in our framework by a low probability of co-location at the beginning of the simulation. In the transient simulations for the fully heterogeneous systems, persistence is a result of the combined effects of low probability of co-location and high probability of low decomposition rate constant at the beginning of the simulation. The heterogeneity in substrate quality thus explains the higher persistence of SOM in the fully heterogeneous system compared to the biophysically heterogeneous system."

5. *Oscillations at multi-annual time scale: Observations of such a phenomenon are very rare. I once argued that we do not see such modelled oscillations with microbial explicit models because of superposition of dynamics across many pores. Here, such spatial heterogeneity is the cause of fluctuations.*

   Please see our response in the interactive discussion. We now highlight the occurrence of these fluctuations in Section A4 (see response below).

6. *Development of the heterogeneity / damping of oscillations (Fig 4): The systems develops to a steady state without any more oscillations. Is the initial heterogeneity developing in direction of homogeneity? Probably not because the simulated SOM stocks differ from the homogeneous system. What is the spatial distribution and covariance between substrate, biomass, and quality after 60 years? Is there a covariance pattern that is stable? I suggest putting another two panels to Fig. 2 showing microbial and substrate distribution at year 60.*

   To address this comment, the following text has been added at P19L14-16,

   'In systems including both biophysical and full heterogeneity, the sums of the HOT are stable in the long term, once the steady state has been reached. This was confirmed by running the model for 100 years.'

7. *Role of disturbances: What happens if you simulate a disturbance (homogenization) after the system is near steady state? Does this start the oscillating pattern again?*

   This comment was previously misinterpreted. If disturbances are re-introduced after 'new' steady state has been reached in a previously heterogeneous system, then yes, oscillations would reappear because this new

steady state would be spatially homogeneous (recall that C input is homogeneous on the domain). Any perturbation near this 'new' steady state would cause fluctuations to reappear.

To address this comment, the following text has been added at P19L17,

'Furthermore, any additional perturbation of the new steady state caused by an external factor will re-introduce the fluctuations.'

8. *Magnitude of the heterogeneity effects: In Figure 4, the effects look large, because the axis ranges from 5 to 7, but aside from the initial disturbance, the effect is only about 1/10 of the steady state. Are there reasonable parameter combinations where the effect is larger? Or do we not need to are this much about heterogeneity at steady state?*

To address this comment, new analyses have been performed. The respectiveresults have been added to the manuscript as a part of the appendix – Appendix A4: 'Sensitivity of fluctuations to changes in $k_M$ in scenario 1', at P40.

'We performed two sensitivity analyses in which we altered the kinetic constant parameter for the multiplicative decomposition model kM: 1) decreasing kM in the biophysical heterogeneity–positively correlated Cs and Cb (Fig. A1) and increasing the heterogeneity of kM (by increasing its standard deviation) in the full heterogeneity case (Fig. A2). From Fig. A1, it is clear that decreasing the rate constant increases the amplitude and wavelength of the oscillations. As shown in Fig. A2, increasing the heterogeneity of the rate constant increases the amount of undecomposed substrate C compared to a lower degree of heterogeneity (Fig. 4). This pattern can be explained using the analytical expression of the steady state substrate C (see Eq.(A16) in Appendix A3). For the increased heterogeneity case shown in Fig. 4, we used values of a and b as listed in TableA3 for biochemical heterogeneity 1 and multiplicative kinetics, where a and b have the same meaning as in Eq. (A16). The analytical expression for the steady state, evaluated with these values of a and b, results in exactly the same steady state of substrate C as simulated by the distributed model (i.e., 15mgC/gSoil).

These fluctuations are similar to those noted in earlier papers using spatially lumped models (Manzoni and Porporato, 2007; Sierra and Muller, 2015). These papers showed that the occurrence and amplitude of the fluctuations depend on the kinetic parameter values, as is the case here.'

9. *2D system: Are the insights transferable to a 3D system. What would you expect to change? Since, there is currently no transport and interaction between the cells, I infer that aside from maybe slightly different development of the initial correlations, the dynamics should stay the same. The macro-scale equations are not affected, as I understood.*

Please see our response in the interactive discussion.

10. *More complex systems: The analytical scale transition approach worked nicely with the basic simple model. With more complex models that include many more heterogeneous parameters it will be difficult to impossible to close the model with all the combination covariances (the factorial grows very fast). Can you*

*describe a strategy to determine which combinations are important and which combinations can be neglected? When have we sufficiently including more and more heterogeneities?*

To address this comment, the following text has been added at P33L9-12,

'Along similar lines, how many terms in the Taylor expansion should be retained at each level of this hierarchy remains an open question. It is also possible that the dynamics at the micro scale in combination with C redistribution lead to low values of higher order moments, thus allowing us to neglect higher order terms-- because substrate consumption, mortality of the microorganisms, and transport contribute to smoothing spatial gradients.'

Admittedly, we do not have a clear answer to this comment.

11. *Time scale: I am especially interested in modelling decadal to longer-term SOM dynamics. Are the multi-annual oscillations important for the longer term dynamics? Do you expect heterogeneity to change with global change in the longer term? What is the advantage of describing the changed steady state with heterogeneity (Fig 7d) with heterogeneity inputs compared to effective model parameters? I see some advantages, but it would be nice to clarify them in the paper.*

Please see our response in the interactive discussion. Despite the interest in these discussion points, we opted for keeping the revised Discussion streamlined and did not cover as initially suggested these topics for the sake of space (the manuscript is already long and discussions along the lines suggested would be rather speculative with the type of model we are using).

Specific comments:

12. *eq. 4 ..6: Your simple basic model refers to the Schimel and Weintraub 2003, who actually used and suggested an inverse MM kinetics $D = ks\, Cs\, Cb\, /\, (kM + Cb)$. It would be nice to amend your work by this decomposition equation.*

To address this comment, new analyses based on inverse MM kinetics were performed. Results and modification to existing figures are listed as following:

- Figures 1 and 3 now include inverse MM kinetics

- Table 2 now lists the macroscopic rate of decomposition for inverse MM kinetics in all three heterogeneous cases

- An additional figure, Fig. 7 has been added to represent the time evolution of state variables in scenario 2

- Figure 8 has been modified by adding two new panels for inverse MM kinetics

- The respective results are now discussed at several locations – in particular: P21, L13-29.

- Derivation of mean rate of decomposition for inverse MM kinetics has now been presented in appendix on P38, L5

- An additional figure, Fig. A6 has been added to represent the time evolution of SOTs for biophysical heterogeneity in scenario 2

13. *P11L11: The sentence does not make sense to me. The variance itself is not always negative. Probably you meant: "This term is always negative because the variance of the spatial substrate distribution is a positive quantity and ..."*

To address this comment, the text has been modified as suggested at P12L6,

'The spatial variance term is always negative because the variance of the spatial substrate distribution is a positive quantity and the partial derivatives multiplying the variances are negative in all decomposition functions that saturate at high substrate concentration.'

14. *P12L15ff: May state that therefore the mean field approximation is exact and spatial variance of this parameter has no effect on the macro-scale dynamics.*

To address this comment, the text has been modified as suggested at P13L7-9,

'Similar derivations can be done for the microbial mortality rate (F = T). The Taylor expansion of microbial mortality is simpler because we assume T to follow first order kinetics implying that all the second order terms are equal to zero. Therefore, the mean field approximation is exact and spatial variance of this parameter has no effect on the macro-scale dynamics'

15. *P12L19: This paragraph comes a bit surprising without context. Why do you look at SGR?*

To address this comment, the text has been modified at P13L11,

'To illustrate how macro-scale decomposition kinetics are affected by spatial heterogeneity, we define a macro-scale specific growth rate (SGR), which is calculated by diving the mean respiration rate by mean microbial C in the system'

16. *P14L11ff: Potential for moving to appendix. Only the information starting from P15L5 is important*

To address this comment, a part of section 2.4: Initial 2D random fields of SOM and kinetic parameters has been moved to appendix– Appendix A2 at P38, L6. Please see the revised manuscript for clarity.

The starting paragraph of section 2.4 Initial 2D random fields of SOM and kinetic parameters will be changed as follows:

'Two-dimensional spatially heterogeneous distributions of substrates and microbial C were generated to run the distributed model. The obtained distributions were based on following constraints: i) the total amount of organic C is set, ii) the total amount of microbial C is 1% of total organic C, iii) the maximum amount of C in a cell is set (Eq. (A12)), and iv) some grid cells have no microbial biomass. For details to the field generation procedure, see Appendix A2.'

17. *P14L19: What is (fg)? I could not find the explanation. It is used several times in the text eq. 26, 27 and appendix figure and table captions.*

Please see our response in the interactive discussion.

18. *P16L10ff: I suggest to give more meaningful names to the scenarios instead of numbers. E.g "Steady simulation" and "High Substrate Simulation" (also update Fig 3).*

The names of the scenarios have been changed as suggested. Now they read as, Steady state simulation (SS) and High substrate simulation (HS). Figure 3 and the text throughout the manuscript have been modified to reflect the change.

**Response to reviewer 2 (Ali Ebrahimi):**

We would like to thank Ali Ebrahimi for the review of our manuscript. Our responses are highlighted in blue (normal font). Some of the comments point to the approach limitations – indeed, this is a theoretical study the findings of which are hard to validate because there is no data at a fine-enough resolution. We would like to emphasize that the value of the proposed approach is to provide a framework for studying heterogeneity effects on measurable C fluxes and stimulate discussion in this area.

Major concerns:

1. *It is unclear to what extend the parameterization proposed in the analytical kinetic model could be experimentally validated. My major problem is that some of the quantities do not have real biogeochemical or physical meanings in which could be experimentally measured. For instance there is an emphasis on the second order moments as state variables used to close the model; however it is hard to think how such variable could be experimentally measured.*

In addition to our response in the interactive discussion, this comment has been addressed by adding the following text at P30L19-26.

'Further, an experimental validation of the present work should stem from designing a microscale experiments using artificial porous media with different degrees of heterogeneity. Recent application of microfluidics in soil science (Stanley et al., 2016; Aleklett et al., 2018) could allow isolating the effect of spatial heterogeneity. If any difference is observed among heterogeneous systems, then our framework could be used to attribute these differences to spatial heterogeneity at the micro-scale. While the proposed mathematical framework is conceptually useful, it is thus challenging to test. Nevertheless, the prediction that co-location of microorganisms and substrates promotes decomposition is consistent with and explains theoretically the results of recent experiments (Don et al., 2013; Schnecker et al., 2019).'

2. *The type of model and scenarios proposed in this study are relevant and could potentially address some of the inconsistency in our field measurements but it could only be possible if the model could establish a systematic link to relevant abiotic and biotic factors observed in the field. While in the discussion authors have tried to relate some of the scenarios in the study to soil aggregation or pore connectivity and an entire subsection is dedicated for that, I still find that the **modeling framework is too abstract that makes the explanations quite speculative** and hard to think to what extend the decomposition rate may vary under realistic settings.*

Please see our response in the interactive discussion.

3. *The model could potentially describe some of the underlying abiotic and physio-logical mechanisms that shape the decomposition dynamics but in the current form of the manuscript this has not been explored. For instance, I was wondering to what extend half saturation to substrate and decomposition rate constant (KM and Ks) are shaping the dynamics observed in the model. I would guess if lower KM or high Ks would have been chosen the heterogeneous scenarios would have converged faster to the homogenous one.*

In addition to our response in the interactive discussion, to address this comment (and in combination to our response to Reviewer 1), we have included a new sensitivity analyses on the kinetic rate constant in Appendix A4. The added text is reported in our response to Reviewer 1 above; new figures A1 and A2 have also been included to illustrate the findings.

Minor concerns:

4. *I was wondering to what extend the fluctuating environmental condition (for instance fluctuating in carbon distributions) could play a role in shaping the carbon decomposition dynamics. Do you expect to see faster convergence to homogenous scenario in high intensity fluctuations?*

Please see our response in the interactive discussion.

5. *The system size that is modeled in grid based network is rather small. The number of grids, or pores equal to 10000, is basically enough to model an aggregate with the size of 0.5mm. I was wondering if this size is sufficiently large to capture heterogeneities in the soil?* **For instance inter aggregate pores or macro-pores***?*

Please see our response in the interactive discussion.

6. *Following up on the results for negative correlations**, I was wondering how much physical inaccessibility of the carbon to microbes could be relevant for the soil systems?** For instance most of the carbon protections in soil are often driven by soil aggregation and creation of anoxic microsites. In a broader term, the counter gradients created by carbon and other necessary substrates for carbon degradation could lead to inaccessibility of the carbon for microbes and not necessary physical inaccessibility. This is a phenomenon that has been previously shown in soil aggregates that due to creation of anoxic zones, the carbon configuration does not play a role in carbon consumption (Ebrahimi and Or, 2018 GCB) and in other studies showing carbon protection by aggregation (e.g., Keiluweitet al., 2017 Nat. comm.).*

Please see our response in the interactive discussion. Moreover, we added the following short paragraph in the Discussion to address this comment (see, P32 L18-21):

"Including C redistribution as a simple mass transfer process does not allow studying how soil structure affects macro-scale dynamics by creating and maintaining heterogeneous distributions of resources and oxygen, such as in soil aggregates (Keiluweit et al., 2017; Ebrahimi and Or, 2018). These patterns result from the interaction of transport and reaction processes that the proposed idealized models cannot capture."

**Response to reviewer 3**

We would like to thank reviewer 3 for the review of our manuscript. In the following, we list the changes (highlighted in blue normal font) made in the revised manuscript in response to Reviewer 3 (denoted by italic font).

1. *While the conclusions they drew are solid within their model configuration, I too share with others the concern that how this learned lesson could be translated into something universally applicable for other modelers. In particular, we in the soil biogeochemical modeling community have so far no unanimously accepted governing equation to solve like that exist for geophysical fluid dynamics, or hydrodynamics in general, where re-solving the microstructure effects can be achieved through the so called large-eddy simulation and sub-grid closure, and even field or laboratory experiments can be designed to derive parameterization schemes that are generally applicable for different situations. Personally, I am therefore wondering can the authors' approach become some tools that are easily accessible to others, e.g., like Markov chain Monte Carlo codes that are widely accessible through open source software?*

   Please see our response in the interactive discussion.

2. *Second I am a little bit disappointed that authors decided to ignore the interactions between different micro-grids. In physics, the successful upscaling is achieved only through the consideration of interactions. For instance, the scaling of Newton's law of momentum conservation, the derivation of center of gravity, or the scaling relation-ship between the Chapman-Enskog theory, lattice Boltzmann approach and the Naiver-Stokes equation, are all hinged on the interactions between their parts. Therefore, it is not surprising at all that the authors found that their mean-field-approximation deviated significantly from their so-called full model simulations. Further, from existing scaling theories in the literature, another key of success seems to maintain the essential invariants of the system when one transits from one scale to another, yet the Michaelis-Menten kinetics they use is a crude approximation and misses some important invariant that is included in its origin law of mass action (Tang and Riley, 2017), and is deemed to show the difference they found. In addition, there's no guarantee that the mean-field equation will possess the same form as the micro-scale equation. For this, a very good example can be found in geophysical fluids, where at different scales, their governing equations are different, e.g., Gill (Atmosphere-Ocean dynamics, 1982)). Another more relevant example on decomposition is in Wang and Allison (2019).*

   To address this comment, new analyses based on mass-transfer model have been performed. New results and modifications to existing figures are listed below:

   - The micro-scale model in Fig. 1 has been modified to account for C redistribution via a simplified mass-transfer approach

   - Equations 2-4 have been added to represent a generic mass-transfer model on P8, L2-4

   - An additional figure, Fig. 9, has been s added to represent the effect of C redistribution among grid cells

   - The new results are discussed on P23, L9-14

- Discussion points based on the new mass-transfer model have been added at several locations, specifically on P29, L11

'Increasing local connectivity among grid cells moderately reduces the effect of spatial heterogeneity on the macro-scale variables and fluxes'

3. *Third, I feel the authors have some misunderstanding about the mean-field theory and the meaning of well-mixed soil condition. In fact, the scaling problem we are facing here is very similar like the situation hydrologists encountered in upscaling the soil moisture and soil matric potential relationship in the 1970s-1980s. Using statistical theory, they were able to derive closed analytical relationships (e.g., Mualem, 1976) to inform important soil water retention curve formulations to be derived from empirical data (e.g.,van Genuchten, 1980). Therefore, whenever moisture-pressure relationships are included in soil biogeochemical models, some microstructure is included in the so-called mean-field-theory based model (although I should admit that the authors did not consider soil moisture in this study). **Or put this straightforwardly, mean field theory does not rule out the inclusion of microstructure, as was demonstrated in the recent up-scaling study of substrate affinity parameter (Tang and Riley, 2019), and the study of turbulence (e.g., Takahashi, 2017).** In the same vein, a well-mixed soil can also have microstructure, and be properly parameterized. In fact, the latter is what motivated the dual-porosity or the multiple-Rates Mass Transfer models, which have enjoyed many successful applications (e.g., Haggerty and Gorelick, 1995).*

In addition to our response in the interactive discussion this comment has been addressed by new results based on inverse MM kinetics, as also mentioned above. This compromise allows maintaining analytical tractability, while avoiding the inclusion of enzyme dynamics in the model, which would alter significantly the model structure. Moreover, we now refer in several points to the work by Tang and Riley (2013, 2017), which is indeed a good example of upscaling, though with a focus on reaction network rather than space per se. For example, the following paragraph has been added to the Discussion (see, P31 L9-12) :

[revised manuscript text omitted]
 inverse MM kinetics (Fig. 7 right panel) show similar dynamics as in the case of biophysical heterogeneity, but with reduced peak magnitude. The smaller mean fluxes are due to the left skewed probability distribution of the kinetic parameters ($k_M$ and $k_{MM}$), which causes slower decay despite the mean values of the kinetic parameters being the same. Mathematically, this behavior is caused by the additional covariances in the fully heterogeneous system as explained in the following paragraph.

~~Figure A4 presents all the higher order spatial moments in the analytical expression of the macroscopic mean respiration rate, for the multiplicative decomposition model. The left (respectively right) vertical column shows the results for biophysically (fully) heterogeneous system with horizontal rows corresponding to the cases of positive (i.e. Fig. A4a and d), negative (i.e. Fig. A4b and e) and uncorrelated (i.e. Fig. A4c 
[revised manuscript text omitted]

Ebrahimi, A. and Or, D.: On Upscaling of Soil Microbial Processes and Biogeochemical Fluxes From Aggregates to Landscapes, Journal of Geophysical Research: Biogeosciences, https://doi.org/10.1029/2017JG004347, 2018.

Ekschmitt, K., Kandeler, E., Poll, C., Brune, A., Buscot, F., Friedrich, M., Gleixner, G., Hartmann, A., Kästner, M., Marhan, S., Miltner, A., Scheu, S., and Wolters, V.: Soil-carbon preservation through habitat constraints and biological limitations on decomposer activity, https://doi.org/10.1002/jpln.200700051, 2008.

Englund, G. and Leonardsson, K.: Scaling up the functional response for spatially heterogeneous systems, Ecology Letters, 11, 440–449, https://doi.org/10.1111/j.1461-0248.2008.01159.x, 2008.

Falconer, R. E., Battaia, G., Schmidt, S., Baveye, P., Chenu, C., and Otten, W.: Microscale Heterogeneity Explains Experimental Variability and Non-Linearity in Soil Organic Matter Mineralisation, PLOS ONE, 10, e0123 774, https://doi.org/10.1371/journal.pone.0123774, 2015.

Fatichi, S., Katul, G. G., Ivanov, V. Y., Pappas, C., Paschalis, A., Consolo, A., Kim, J., and Burlando, P.: Abiotic and biotic controls of soil moisture spatiotemporal variability and the occurrence of hysteresis, Water Resources Research, 51, 3505–3524, https://doi.org/10.1002/2014WR016102, 2015.

Forney, D. C. and Rothman, D. H.: Common structure in the heterogeneity of plant-matter decay, Journal of the Royal Society Interface, 9, 2255–2267, https://doi.org/10.1098/rsif.2012.0122, 2012.

Fraser, F. C., Todman, L. C., Corstanje, R., Deeks, L. K., Harris, J. A., Pawlett, M., Whitmore, A. P., and Ritz, K.: Distinct respiratory responses of soils to complex organic substrate are governed predominantly by soil architecture and its microbial community, Soil Biology and Biochemistry, 103, 493–501, https://doi.org/10.1016/j.soilbio.2016.09.015, 2016.

Georgiou, K., Abramoff, R. Z., Harte, J., Riley, W. J., and Torn, M. S.: Microbial community-level regulation explains soil carbon responses to long-term litter manipulations, Nature Communications, 8, 1–10, https://doi.org/10.1038/s41467-017-01116-z, 2017.

German, D. P., Marcelo, K. R. B., Stone, M. M., and Allison, S. D.: The Michaelis-Menten kinetics of soil extracellular enzymes in response to temperature: A cross-latitudinal study, Global Change Biology, 18, 1468–1479, https://doi.org/10.1111/j.1365-2486.2011.02615.x, 2012.

Ginovart, M. and Valls, J.: Individual Based Modelling of Microbial Activity to Study Mineralization and Nitrification Process in Soil, AICME II abstracts, 6, 1996, https://doi.org/10.1016/j.nonrwa.2004.12.005, 1996.

Herbst, M., Tappe, W., Kummer, S., and Vereecken, H.: The impact of sieving on heterotrophic respiration response to water content in loamy and sandy topsoils, Geoderma, 272, 73–82, https://doi.org/10.1016/j.geoderma.2016.03.002, 2016.

Hunt, A. G. and Manzoni, S.: Networks on Networks, 2053-2571, Morgan & Claypool Publishers, https://doi.org/10.1088/978-1-6817-4159-8, 2015.

Jenkinson, D. and Rayner, J.: The turnover of soil organic matter in some of the Rothamsted classical experiments, Soil science, 123, 298–305, 1977.

Jenny, H., Gessel, S., and Bingham, F.: Comparative study of decomposition rates of organic matter in temperate and tropical regions, Soil Science, 68, 419–432, 1949.

Juarez, S., Nunan, N., Duday, A. C., Pouteau, V., Schmidt, S., Hapca, S., Falconer, R., Otten, W., and Chenu, C.: Effects of different soil structures on the decomposition of native andadded organic carbon, European Journal of Soil Biology, 58, 81–90, https://doi.org/10.1016/j.ejsobi.2013.06.005, 2013.

Kaiser, C., Franklin, O., Dieckmann, U., and Richter, A.: Microbial community dynamics alleviate stoichiometric constraints during litter decay, Ecology Letters, 17, 680–690, https://doi.org/10.1111/ele.12269, 2014.

Keeling, M. J. J., Wilson, H. B. B., and Pacala, S. W. W.: Deterministic Limits to Stochastic Spatial Models of Natural Enemies, The American naturalist, 159, 57–80, https://doi.org/10.1086/324119, 2002.

Keiluweit, M., Wanzek, T., Kleber, M., Nico, P., and Fendorf, S.: Anaerobic microsites have an unaccounted role in soil carbon stabilization, Nature communications, 8, 1771, 2017.

[revised manuscript text omitted]

---

## Author Response (AR2)

**Revised manuscript "Dynamic upscaling of decomposition kinetics for carbon cycling models"**

This document lists the point-by-point response to the reviewer comments and the modifications made in the revised manuscript to reflect reviewers' comments. Additionally, we have updated the model code on Zenodo repository, which can be now accesses at https://doi.org/10.5281/zenodo.3576613.

Part of mathematical text in the marked-up manuscript did not render well because of latexdiff. Please refer to the revised manuscript for the correct mathematical expressions. Page and line number references correspond to the revised manuscript.

**Reviewer 1 comments (Thomas Wutzler):**

We would like to thank Thomas Wutzler for a positive assessment of our revised manuscript. In the following, we list the changes (highlighted in blue normal font) made in the revised manuscript in response to Dr. Wutzler (denoted by italic font).

1. *One thing I still missing is some discussion of the stable pattern of heterogeneities formed at the final states: From Fig. A3f I get that the second order terms are zero except the negative covariance between decomposition rate and substrate concentration for the multiplicative decomposition model. I interpret this as a shift in substrate quality (higher concentrations of low quality) than the original loguniform distribution of substrate qualities. And this to me explains the higher new steady-state stocks for the same given substrate input. For the other decomposition models, I do not find such information, but there seems to be a fundamental difference with the MM decomposition, which has positive sum of higher order terms for a longer time.*

We agree with the reviewer's interpretation of stable pattern of heterogeneities formed at the final states. As stated by the reviewer, the negative covariance between decomposition rate ($k_M$) and substrate concentration for the multiplicative decomposition model can be interpreted as the occurrence of relatively high concentrations of low-quality substrate in at least some grid cells, which lowers decomposition in those cells. It should be noted that the effect of strongly negative covariance causing low decomposition leading to accumulation of C and a new steady state, occurs in the first years of simulation. However, in the long term, the negative covariance balances the larger value of the MFA compared to a homogenous system, allowing the overall decomposition rate to match the input rate.

The steady state values at each gird cell $\left(i.e., \frac{k_B}{Y\,k_M}\right)$ are inversely proportional to $k_M$. As a result, at steady state the grid cells with low values of $k_M$ (whose likelihood depends on the initial probability density function of $k_M$) attain higher concentrations of substrate C. Since steady state substrate is also inversely proportional to the rate constant when using Michaelis-Menten or inverse Michaelis-Menten kinetics, we expect this behavior of sustained higher substrate C in chemically heterogeneous systems to occur regardless of the chosen kinetics law.

We address this comment by adding a paragraph at the end of Appendix A3:

"Equation (A15) shows that $\bar{C}_s^*$ in the heterogeneous system deviates from the value attained in a homogeneous system because in general $k_M$ differs from $(b-a)\ln(10)/(10^{-a}-10^{-b})$. Similar derivations can be made for the other formulations of decomposition kinetics."

**Minor comments:**

1. *Abstract sentence at line 15 is hard to parse. After my first reading I got – "It was not possible". Hence I suggest reordering to: "The homogenous model assumption was not able to capture ... Only the inclusion of second order moments ..."*

To address this comment, P1L15-17 was streamlined and changed to,

"The model assuming homogeneous conditions was not able to capture the mean behavior of the heterogeneous system because the second order moments cause $\bar{R}$ in the heterogeneous system to be higher or lower than in the homogeneous system, depending on the sign of the second order spatial moments."

2. *page 7: symbol D is used both for diffusion coefficient and for decomposition. I suggest to distinguish the diffusion coefficient by an subscript.*

We apologize for the ambiguity. The diffusion coefficient $D$ is now changed to $D_{diff}$.

3. *What are the implications or complications of dropping the assumption of spatially invariant Y and I?*

Spatially varying $Y$ and $I$ do not pose any major numerical averaging challenges; however, interpreting the results using the analytical equations becomes more challenging when a large number of second order terms is involved. For example, in the case of multiplicative model with $D = k\, C_s\, C_b$;

$$\frac{d\bar{C}_s}{dt} = \bar{I} - \bar{D} + \bar{T},$$

$$\frac{d\bar{C}_b}{dt} = \overline{YD} - \bar{T} = \bar{G} - \bar{T},$$

where $\bar{D}$ and $\bar{G}$ for the biophysically heterogeneous and biochemically homogeneous system are given by,

$$\bar{D} = k_M\, \bar{C}_s\, \bar{C}_b + k_M \overline{C_s' C_b'}\,,$$

$$\bar{G} = \overline{YD} = k_M\left[\bar{Y}\, \bar{C}_s\, \bar{C}_b + \bar{Y}\,\overline{C_s' C_b'} + \bar{C}_s\,\overline{Y' C_b'} + \bar{C}_b\,\overline{Y' C_s'}\right] + \overline{k_M' C_s' C_b'}\,,$$

$$\bar{R} = \bar{D} - \overline{YD} = \bar{D} - \bar{G}.$$

The steady state values of $\bar{C}_s$ and $\bar{C}_b$ would also be changed because of their dependence on $I$ and $Y$. For example, if only $Y$ is spatially varying (keeping $k_M$ and $I$ spatially invariant), then the average steady state value of $C_b$ increases and $C_s$ decreases for a distribution of $Y$ skewed toward higher values. With this logic, we expect the $\overline{Y' C_b'}$ to be positive and $\overline{Y' C_s'}$ to be negative near steady state.

We added a comment at P13L15 to explain that adding variability in $I$ and $Y$ would change the macroscopic equations and potentially affect C dynamics as well:

"The same rationale used to derive Eq. (21) can be applied in systems where the C input rate $I$ or the microbial C-use efficiency $Y$ are not homogeneous. This would cause additional second and higher order terms to appear in the macroscale equations, with consequences on the overall C balances."

4. *P12L10: Why do Cb and k need the averaging bar here? Better call Cb an initial condition rather than model parameter.*

To address this comment, P12L17 has been changed to,

"In this case, model parameters $k$ vary spatially but the initial values of the state variables $C_s$ and $C_b$ are constant everywhere in the domain."

5. *P20 Fig4: By looking at the y-axis scales I noticed that you have quite high microbial biomass per organic matter (Cs+Cb) of about 15% which you report being still low compared to half-saturation Km. Did you have any incubation experiment in mind, i.e. of which soil is your simulation representative for?*

The parameters have not been calibrated from a specific incubation experiment. The high value of steady state $C_b$ is caused by the choice of parameters $Y$, $I$ and $k_b$. In MM kinetics, the half-saturation constant, $K_{MM} = 25$ mgC/gSoil is the same for the transient and the steady state cases. The initial mean $C_s$ in the high substrate scenario (121 mgC/gSoil) is higher than $K_{MM}$ and in the steady state scenario (5.9 mgC/gSoil).

The issue of relatively high value of microbial C can be addressed only by changing the value of $k_B$. When assuming multiplicative kinetics, the steady state for substrate and microbial C are $\frac{k_B}{Y\,k_M}$ and $\frac{Y\,I}{(1-Y)k_B}$, respectively. If we increase the value of $k_B$ three times then the ratio of $C_b/C_s$ is approximately 0.018 or 1.8% - a value more in line with commonly measured microbial biomass to total SOC ratios. This choice of a new $k_B$ does not drastically change the results in the high substrate simulation scenario. Below is figure R1, which shows the dynamics of substrate and microbial C, as also presented in Fig 4 in the manuscript. The initial and final steady state values of substrate and microbial C are now different (in the fully heterogeneous system).

However, we would prefer to keep the current parameterization, which is reasonable when considering that one model is simple and includes a single substrate C pool. This substrate compartment does not include all soil organic carbon, but only the fraction that turns over relatively quickly. To explain this, the following sentences were added at P7L8,

"It should be noted that conceptually we include in $C_s$ only organic C that is available for de-polymerization and not stabilized; in other words, we focus on decomposition time scales in the order of weeks to months. Modeling all the processes (and associated heterogeneity effects) leading to C stabilization is beyond our scope."

In addition, the following sentences at P18L7,

"With the current parameter choice, the ratio of microbial biomass to substrate C attains at steady state values that are larger than would be expected for biomass-to-total SOC ratios (Xu et al., 2013). This is due to our interpretation of substrate C as a relatively active fraction of total SOC."

Biophysical heterogeneity

[Figure]

**Figure R1: Substrate and microbial biomass C (top rows), respiration and higher order terms (bottom rows) as a function of time, for different scenarios of initial spatial heterogeneity. This figure is similar to Figure 4 in the manuscript, except for a value of $k_{b,new} = \frac{k_b}{3}$, where $k_{b,new}$ is the altered microbial mortality constant.**

6. *P22L13 and Fig.6. I missed some discussion why the sumSOT is behaving so differently for the MM kinetics in the full heterogeneity compared to the other kinetics.*

Figure R2 below shows the SOTs for MM kinetic for uncorrelated substrate and microbial biomass in a fully heterogeneous system. SOT1, SOT2 and so on are the SOTs of Eq. A9 starting from first to last (see equation below).

$$\overline{F}(C_s, C_b, [k_{MM}, K_{MM}]) = \qquad \textcolor{red}{1}$$

$$F(\overline{C}_s, \overline{C}_b, \overline{k}_{MM}, \overline{K}_{MM}) + \frac{1}{2}\frac{\partial^2 F}{\partial C_s^2}\bigg|_{\overline{C}_s, \overline{C}_b, \overline{k}_{MM}, \overline{K}_{MM}} \sigma^2_{C_s} +$$

$$\frac{1}{2}\frac{\partial^2 F}{\partial K_M^2}\bigg|_{\overline{C}_s, \overline{C}_b, \overline{k}_{MM}, \overline{K}_{MM}} \textcolor{red}{2} \sigma^2_{K_M} + \frac{\partial^2 F}{\partial k_{MM}\partial K_{MM}}\bigg|_{\overline{C}_s, \overline{C}_b, \overline{k}_{MM}, \overline{K}_{MM}} \textcolor{red}{3} \overline{k'_{MM} K'_{MM}} +$$

$$\frac{\partial^2 F}{\partial C_s\partial C_b}\bigg|_{\overline{C}_s, \overline{C}_b, \overline{k}_{MM}, \overline{K}_{MM}} \textcolor{red}{4} \overline{C'_s C'_b} + \frac{\partial^2 F}{\partial k_{MM}\partial C_s}\bigg|_{\overline{C}_s, \overline{C}_b, \overline{k}_{MM}, \overline{K}_{MM}} \textcolor{red}{5} \overline{C'_s k'_{MM}} +$$

$$\frac{\partial^2 F}{\partial k_{MM}\partial C_b}\bigg|_{\overline{C}_s, \overline{C}_b, \overline{k}_{MM}, \overline{K}_{MM}} \textcolor{red}{6} \overline{C'_b k'_{MM}} + \frac{\partial^2 F}{\partial K_{MM}\partial C_s}\bigg|_{\overline{C}_s, \overline{C}_b, \overline{k}_{MM}, \overline{K}_{MM}} \textcolor{red}{7} \overline{C'_s K'_{MM}} +$$

$$\frac{\partial^2 F}{\partial K_{MM}\partial C_b}\bigg|_{\overline{C}_s, \overline{C}_b, \overline{k}_{MM}, \overline{K}_{MM}} \textcolor{red}{8} \overline{C'_b K'_{MM}}. \qquad\qquad\qquad\qquad (A9)$$

From the figure, it is clear that the sum of SOT+MFA (green dashed line) is not close to the simulated mean respiration (purple dashed line), this means that SOTs are not enough for closing the system in the full heterogeneous case. The sum of SOT (red dashed line) is initially positive, but later becomes negative. The positive $\sum SOT$ is driven by the $SOT6 = \frac{\partial^2 F}{\partial k_{MM} C_b}\overline{C'_b k'_{MM}} = \frac{\bar{c}_S}{\bar{c}_S + \overline{K}_{MM}}\overline{C'_b k'_{MM}}$ (light cyan). In other words, grid cells with high amount of microbial C and high rate constant cause the positive covariance $(\overline{C'_b k'_{MM}})$ and this covariance becomes negative only after microbial C $(C_b)$ nears the steady state.

[Figure]

**Figure R2. Second order terms (SOT; see Eq. A9) and their contribution to the modeled mean respiration rate in a fully heterogeneous system with Michaelis-Menten decomposition kinetics, as a function of time.**

We address this comment by adding the following sentences at the end of section 3.2.2.

"This initial positive $\sum$ SOT is driven by the sixth SOT in Eq. (A9) i.e. $\frac{\partial^2 F}{\partial k_{MM} C_b}\overline{C'_b k'_{MM}}$, because grid cells with high amount of microbial C and high rate constant cause the covariance $\overline{C'_b k'_{MM}}$ to be positive. This covariance becomes negative only after microbial C nears the steady state."

**Reviewer 3 comments:**

We would like to thank reviewer 3 for a positive assessment of our revised manuscript. In the following, we list the changes (highlighted in blue normal font) made in the revised manuscript in response to Dr. Wutzler (denoted by italic font).

1. *My only suggestion is that the authors may consider a better use of Table 2 when interpreting the results (particularly section 3). At some places, linking the discussion with the exact term in Table 2 may help readers keeping their line of thought. As it is currently written, I sometimes have to find the equation when I go through the description and discussion of the results.*

Thank you for the suggestion. To address this comment, P19L18-20 has been changed to:

"Figure 4g and 4h show the sum of all higher order terms ($\sum$HOT, see Table 2 for multiplicative kinetics). For a biophysically heterogeneous system, the $\sum$ HOT only includes the spatial covariance term (Eq. (18)), but for a fully heterogeneous system it includes the last three terms of the Eq. (21) as well as the third order term $\overline{k_M' C_s' C_b'}$."

P21L16 has been changed to:

"Similar to Fig. 4, Fig. 5g and 5h show the sum of all higher order terms (see Table 2, multiplicative kinetics)."

2. *Finally, the authors may do one more careful check of the English. Though I am not an expert, I think some places missed a comma (e.g., line 4, page 12), or misused a word (e.g., The use of "aligned" in Line 13 page 19).*

Thank you for pointing it out. In the revised manuscript, we have proof-read the manuscript and corrected language issues. We also streamlined the abstract, which was fairly long.

**References:**

[revised manuscript text omitted]